# The astrocyte-produced growth factor HB-EGF limits autoimmune CNS pathology

Central nervous system (CNS)-resident cells such as microglia, oligodendrocytes and astrocytes are gaining increasing attention in respect to their contribution to CNS pathologies including multiple sclerosis (MS). Several studies have demonstrated the involvement of pro-inflammatory glial subsets in the pathogenesis and propagation of inflammatory events in MS and its animal models. However, it has only recently become clear that the underlying heterogeneity of astrocytes and microglia can not only drive inflammation, but also lead to its resolution through direct and indirect mechanisms. Failure of these tissue-protective mechanisms may potentiate disease and increase the risk of conversion to progressive stages of MS, for which currently available therapies are limited. Using proteomic analyses of cerebrospinal fluid specimens from patients with MS in combination with experimental studies, we here identify Heparin-binding EGF-like growth factor (HB-EGF) as a central mediator of tissue-protective and anti-inflammatory effects important for the recovery from acute inflammatory lesions in CNS autoimmunity. Hypoxic conditions drive the rapid upregulation of HB-EGF by astrocytes during early CNS inflammation, while pro-inflammatory conditions suppress trophic HB-EGF signaling through epigenetic modifications. Finally, we demonstrate both anti-inflammatory and tissue-protective effects of HB-EGF in a broad variety of cell types in vitro and use intranasal administration of HB-EGF in acute and post-acute stages of autoimmune neuroinflammation to attenuate disease in a preclinical mouse model of MS. Altogether, we identify astrocyte-derived HB-EGF and its epigenetic regulation as a modulator of autoimmune CNS inflammation and potential therapeutic target in MS.

MS is a demyelinating disease of the CNS, characterized by relapsing focal inflammation and formation of lesions in the white and gray matter[1]. MS initially manifests with a first demyelinating episode, often termed clinically isolated syndrome (CIS) or relapsing–remitting MS (RRMS), depending on the diagnostic criteria met[2,3]. In most cases, this first episode is followed by a temporally and spatially separated second neurological event, confirming the diagnosis of MS in CIS patients[4]. Both in CIS and in RRMS, the risk of a second relapse increases with magnetic resonance imaging (MRI) lesion load

and the presence of oligoclonal bands at diagnosis, among others[5–7]. While these observations improve clinical decision making and allow early therapeutic intervention with disease-modifying therapies (DMTs)[8–10] to delay disease progression, pathophysiological differences between the initial and consecutive demyelinating episodes are not entirely clear. Here, we identify HB-EGF as a protective factor produced by astrocytes during the initial episodes of CNS inflammation, where it controls recovery from acute autoimmune inflammation, but is suppressed in later stages of CNS inflammation.

✉e-mail: veit.rothhammer@fau.de

**Fig. 1 | Regulation of HB-EGF during autoimmune CNS inflammation.**
**a**, Multiplex analysis of CSF of controls ($n = 20$) and CIS ($n = 21$) and RRMS
($n = 54$) patients. **b**, PCA of the CSF abundance of the measured analytes in
controls ($n = 20$) and CIS ($n = 21$) and RRMS ($n = 54$) patients. **c**, Absolute CSF
concentrations of the measured analytes in the CSF of controls ($n = 20$) and
CIS ($n = 21$) and RRMS ($n = 54$) patients clustered by Euclidian distance.
**d**, Volcano plot depicting the $\log_2$ fold change in analyte abundance in the CSF
of CIS ($n = 21$) versus RRMS ($n = 54$) patients. **e**, CSF concentration of HB-EGF in

controls ($n = 20$) and CIS ($n = 21$) and RRMS ($n = 54$) patients. **f**, Receiver operating
curve describing CSF HB-EGF concentration as classifier for CIS versus non-CIS
diagnosis. **g**, Linear regression analysis with 95% confidence intervals of HB-EGF
concentration in the CSF of CIS patients ($n = 21$) and the number of cerebral
lesions. **h**, Patient-specific ratio between HB-EGF in the CSF (HB-EGF$_{CSF}$) and
HB-EGF in the serum (HB-EGF$_{Serum}$) in controls ($n = 20$) and CIS ($n = 21$) and RRMS
($n = 54$) patients. Data are shown as mean ± s.d. Ordinary one-way ANOVA with
Tukey's multiple comparisons test in **e** and **h**. PC, Principal Component.

## Regulation of HB-EGF during autoimmune CNS inflammation

Alterations in tissue- and neuroprotective factors following inflammatory insult of the CNS have recently been attributed with important roles in the pathogenesis and progression of neuroinflammatory events[11,12]. Therefore, we speculated that the loss of specific tissue-protective signals may facilitate a second relapse after a first demyelinating event. To identify disease-stage-specific protective signals altered between patients that experienced a single or multiple demyelinating events, we analyzed the abundance of 28 tissue-protective factors in the cerebrospinal fluid (CSF) of patients with only one inflammatory relapse (further termed CIS, $n = 21$), multiple relapses (further termed RRMS, $n = 54$) and noninflammatory controls that presented with primary headache ($n = 20$) (Fig. 1a). The set of analytes was composed of factors that have previously been described in the context of MS (GFAP, YKL-40, CD44), but also other neurological (Aβ$_{1-42}$, NSE) and non-neurological disorders (VEGF-A), while their exact regulation in the CSF of patients with CIS and RRMS largely remained undefined. Of note, patients with CIS were included in the study during their first inflammatory episode, while patients with RRMS had experienced at least one previous

inflammatory event and were also sampled during relapse. Following filtering, we performed dimensionality reduction based on the absolute concentration of the remaining factors and clinical parameters, including disease duration, disability measures, treatment type and CSF cell counts, among others (Supplementary Table 1 and Extended Data Fig. 1a). Principal component analysis (PCA) segregated patients by disease stage (Extended Data Fig. 1a). Next, we evaluated how well the abundance of the measured tissue-protective factors differentiated patient groups without the consideration of additional clinical measures. Dimensionality reduction showed a separation of patients with MS and controls, but also between CIS and RRMS patients (Fig. 1b). This separation was confirmed by clustering based on Euclidean distance (Fig. 1c) and was largely driven by the HB-EGF, which was significantly elevated in the CSF of CIS patients compared with RRMS and controls (Fig. 1d,e and Extended Data Fig. 1b).

HB-EGF is a member of the epidermal growth factor (EGF) family of proteins with critical roles in development, tissue regeneration and cancer in a variety of organs including kidney, liver, heart, bladder and skin[13]. Additionally, HB-EGF has been associated with neuronal

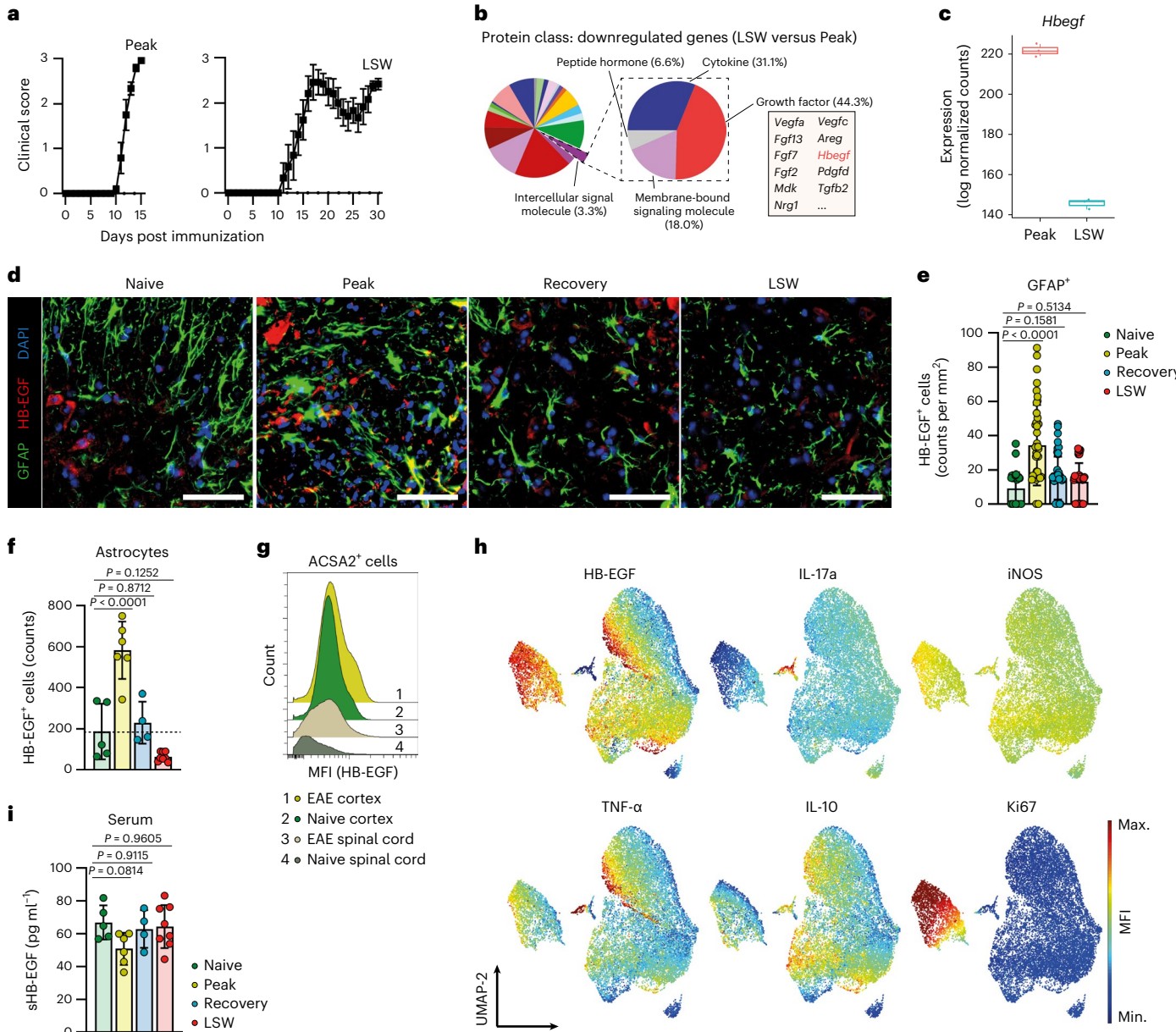

**Fig. 2 | Expression of HB-EGF by reactive astrocytes. a**, EAE development and timepoints (peak, LSW) used for RNA-seq analysis of ACSA2⁺ cortical astrocytes (*n* = 5 per group). **b**, PANTHER Protein class analysis of genes downregulated in LSW compared with peak astrocytes. **c**, Normalized RNA expression of *Hbegf* by LSW and peak astrocytes (*n* = 3 per group). Data are shown as mean with the 25th and 75th percentiles. **d**,**e**, Immunostaining (**d**) and quantification (**e**) of HB-EGF⁺GFAP⁺ cells in spinal cords of EAE (peak, recovery, LSW; *n* = 4 per timepoint) and naive mice (*n* = 3). Scale bar, 50 μm. **f**, Flow cytometric analysis of HB-EGF⁺ astrocytes during peak (*n* = 6), recovery (*n* = 4) and LSW

(*n* = 7) and in naive mice (*n* = 5). **g**, Representative histogram depicting median fluorescence intensity (MFI) of HB-EGF in cortex and spinal cord astrocytes of EAE and naive mice (*n* = 3/5 per group). **h**, UMAP plot of ACSA2⁺ astrocytes analyzed by multidimensional intracellular flow cytometry; colors indicate MFI. **i**, Serum levels of sHB-EGF in EAE mice over the course of disease. Peak *n* = 6, recovery *n* = 4, LSW *n* = 8, naive *n* = 5. Data are shown as mean ± s.d. Data are shown as mean ± s.e.m. in **a**. Ordinary one-way ANOVA with Dunnett's multiple comparisons test (tested against naive) in **e**, **f** and **i**.

survival and tissue regeneration during development and ischemic diseases of the CNS, where it promotes protective mechanisms and induces neuronal and oligodendrocyte survival and differentiation[13–15]. However, its role in CNS autoimmunity is yet unknown. The distinct increase in HB-EGF in CSF of CIS patients was confirmed by a logistic regression model, demonstrating that HB-EGF concentration in the CSF of patients with MS is an effective discriminator between CIS versus non-CIS patients (Fig. 1f and Extended Data Fig. 1c).

Along these lines, we observed a negative correlation between HB-EGF levels in the CSF and the number of CNS lesions in CIS patients,

which has been described as an important risk factor for the conversion to RRMS[16] (Fig. 1g). The analysis of a patient with RRMS with available longitudinal data further demonstrated a continuous decrease of HB-EGF in the CSF with the number of relapses, while other factors remained constant or increased over time (Extended Data Fig. 1d). In contrast, we did not observe a correlation of HB-EGF CSF levels with sex, age or disease duration in CIS and RRMS patients (Extended Data Fig. 1e). Furthermore, HB-EGF serum levels were slightly reduced in CIS patients compared with controls, and the CSF HB-EGF/serum HB-EGF ratio was increased in CIS samples, suggesting a CNS-specific

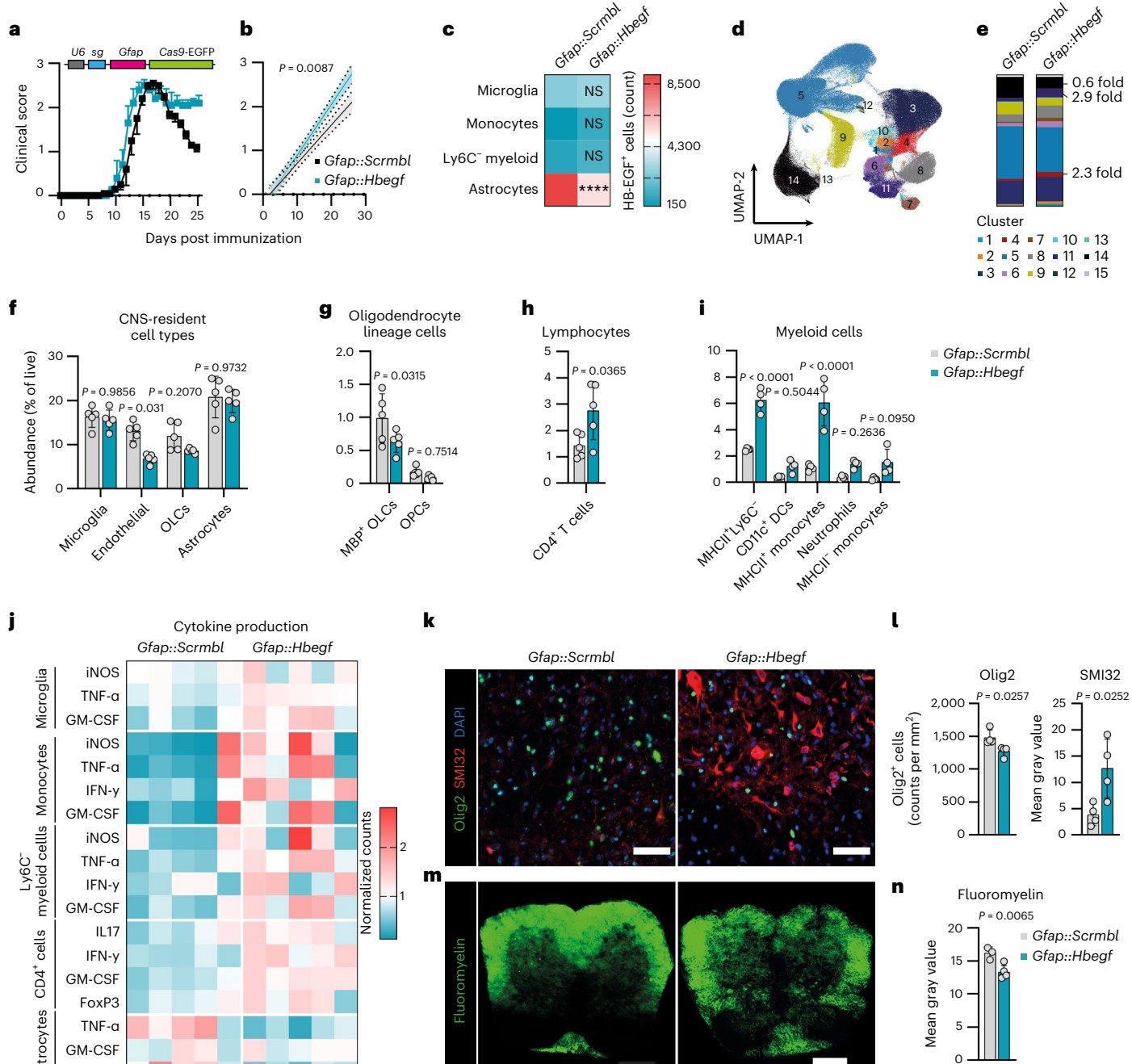

**Fig. 3 | Astrocyte-derived HB-EGF is important for recovery from EAE. a,b,** EAE development (**a**) and regression analysis from EAE start (**b**) in mice transduced with *Gfap::Scrmbl* (*n* = 5) and *Gfap::Hbegf* (*n* = 5). The experiment was repeated twice. **c,** Absolute counts of HB-EGF⁺ cells in the CNS of *Gfap::Scrmbl* (*n* = 4) and *Gfap::Hbegf* (*n* = 4) mice analyzed by intracellular flow cytometry. **d,e,** UMAP plot of CNS cells (**d**) with FlowSOM clusters (**e**) analyzed by high-dimensional flow cytometry in *Gfap::Scrmbl* (*n* = 5) and *Gfap::Hbegf* (*n* = 5) mice. **f–i,** Abundance of microglia, endothelial cells and OLCs (**f**), myelinating OLCs and OPCs (**g**), CD4⁺ T cells (**h**) and infiltrating myeloid cells (**i**) in the CNS of *Gfap::Scrmbl* (*n* = 4/5) and *Gfap::Hbegf* (*n* = 4/5) mice. A detailed gating strategy is provided in Extended Data Fig. 3b. **j,** Cytokine production by microglia, monocytes,

Ly6C⁻ myeloid cells, CD4⁺ T cells and astrocytes in the CNS of *Gfap::Scrmbl* (*n* = 5) and *Gfap::Hbegf* (*n* = 5) mice analyzed by intracellular flow cytometry. Counts were normalized to vehicle-treated mice. Additional statistics and gating strategy are provided in Extended Data Figs. 3b and 4e. **k,l,** Immunostaining (**k**) and analyses (**l**) of Olig2 and SMI32 in spinal cords of *Gfap::Scrmbl* (*n* = 4) and *Gfap::Hbegf* (*n* = 4) mice. Scale bar, 50 μm. **m,n,** Immunostaining (**m**) and analysis (**n**) of fluoromyelin in spinal cords of *Gfap::Scrmbl* (*n* = 3) and *Gfap::Hbegf* (*n* = 5). Scale bar, 200 μm. All data were collected at day 25 post immunization. Data are shown as mean ± s.d. if not indicated otherwise. Data are shown as mean ± s.e.m. in **a**. Two-way ANOVA with Sidak's multiple comparisons test in **c, f, g, h** and **i**; unpaired *t*-test in **l** and **n**. ****p$P$ < 0.0001. DC, dendritic cell; NS, not significant.

regulation in early stages of autoimmune CNS inflammation (Fig. 1h and Extended Data Fig. 1f). Lastly, we did not observe changes in the CSF abundance of HB-EGF in patients with progressive MS, suggesting that its elevation is specific to acute stages of the disease (Extended Data Fig. 1g and Supplementary Table 2).

Next, we sought to identify the cellular source of HB-EGF in the brain. Since we observed a CSF-specific increase in HB-EGF, we speculated that CNS-resident cells may produce the tissue-protective factor in response to acute inflammation. In this context, astrocytes have gained increasing attention with respect to their roles in autoimmune

CNS pathologies such as MS. In addition to their contribution to the pathogenesis and propagation of neuroinflammatory events[17], it has recently become clear that astrocytes can not only drive inflammation but also lead to its resolution through direct and indirect mechanisms associated to specific functional subsets[11,18].

To investigate the spatiotemporal expression of HB-EGF by astrocytes during early and late stages of autoimmune neuroinflammation, we induced experimental autoimmune encephalomyelitis (EAE) by immunization of wild-type C57Bl/6 mice with MOG$_{35-55}$ peptide and performed bulk RNA sequencing (RNA-seq) of ACSA2-sorted astrocytes during early acute and late stages of EAE, 15 d and 30 d after disease induction, respectively (Fig. 2a). While this preclinical model of MS is classically described as monophasic and only recapitulates the complexity of the disease to a limited extent, it is important to note that late stages of CNS inflammation were characterized by increased clinical scores and differed in the cellular composition in the CNS from recovery, but also peak stages of EAE (Extended Data Fig. 2a–c), pointing to differential mechanisms driving disease in acute versus later stages of clinical deficit, an observation that has been described before[19]. These differences were also recapitulated in astrocytes during late-stage worsening of EAE (further defined as LSW) and early acute stages (peak) on the transcriptomic and protein level (Extended Data Fig. 2d–g), pointing to a specific regulation of the CNS micromilieu during late stages of EAE as compared with the recovery and peak phases. Indeed, Kyoto Encyclopedia of Genes and Genomes (KEGG) Pathway analysis revealed an enrichment in pathways associated to neuroprotective functions in astrocytes during LSW compared with peak stages (Extended Data Fig. 2h). Among other tissue-protective growth factors (*Vegfa*, *Tgfb2*, *Fgf2*) reduced during LSW compared with peak EAE, we identified *Hbegf* downregulation in astrocytes (Fig. 2b,c). Immunohistochemical and flow cytometric analyses of HB-EGF production by cortical and spinal cord astrocytes throughout the course of EAE confirmed these observations, collectively demonstrating that both cortical and spinal cord astrocytes upregulate HB-EGF during peak stages and lose its expression during LSW (Fig. 2d–g and Extended Data Fig. 2i,j), aligning with our findings in patients with CIS and RRMS. Furthermore, dimensionality reduction following high-parameter intracellular flow cytometry revealed that HB-EGF was highly expressed by a proliferative subset of astrocytes, predominantly of cortical origin (Fig. 2g,h and Extended Data Fig. 2j). Finally, in line with our observations in patients with CIS and RRMS (Fig. 2i and Extended Data Fig. 1f), we observed no major differences in the seroabundance of HB-EGF

during EAE, further supporting its CNS-specific regulation. Overall, these data suggest that HB-EGF is produced by reactive astrocytes in response to acute autoimmune CNS inflammation, but decreases during its later stages.

## Astrocyte-derived HB-EGF promotes recovery from EAE

To investigate the role of astrocyte-derived HB-EGF in the pathogenesis and progression of autoimmune CNS inflammation, we inactivated *Hbegf* using a lentiviral vector that co-expresses *Gfap*-driven CRISPR–Cas9 and a targeting single guide RNA (sgRNA) and induced EAE as previously described[20]. When compared with mice targeted with an *Scrmbl* control sgRNA, astrocyte-specific inactivation of *Hbegf* interfered with the recovery from acute EAE (Fig. 3a–c and Extended Data Fig. 3a–e).

High-parameter flow cytometry followed by dimensionality reduction revealed substantial differences in the cellular abundance of both CNS-resident and -infiltrating cells (Fig. 3d,e and Extended Data Fig. 3f). By using significance analysis of microarray (SAM)[21], a non-parametric statistical method used for determining significant features from input data, we found particularly clusters 4, 6, 7, 8 and 11 significantly expanded, and clusters 5 and 14 decreased, in *Gfap::Hbegf* mice (Fig. 3d,e and Extended Data Fig. 4a,b). These changes in cluster abundance corresponded to a decrease in oligodendrocyte lineage cells (OLCs) and endothelial cells, as well as an increase in CD4$^+$ T cells and multiple myeloid lineage cells, which was confirmed by manual gating (Fig. 3f–i and Extended Data Fig. 4c,d). This was in line with an increase in pro-inflammatory cytokines produced by microglia, monocytes, Ly6C$^-$ myeloid cells and CD4$^+$ T cells in *Gfap::Hbegf* mice (Fig. 3j and Extended Data Fig. 4e,f), suggesting that astrocyte-derived HB-EGF is critical for the regulation of inflammatory functions of both CNS-resident, but also -infiltrating, cell types. The increase in pathogenicity observed in *Gfap::Hbegf* mice was also in accordance with a loss of Olig2$^+$ cells, overall myelin, and an increase in axonal damage in the spinal cord (Fig. 3k–n). Collectively, these data demonstrate that astrocyte-derived HB-EGF is an important regulator of tissue-damage and inflammatory functions of CNS-resident and -infiltrating cell types during late stages of neuroinflammation.

## AhR and HIF1α control HB-EGF

To identify the inflammatory mechanisms that trigger the upregulation of HB-EGF during acute phases of autoimmune CNS inflammation, we challenged naive astrocytes in vivo by intracerebroventricular (i.c.v.) injection of TNF-α and IL-1β, two factors that have been linked to MS

**Fig. 4 | HIF1α and AhR oppositely control HB-EGF in astrocytes. a**, RT–qPCR analysis of *Hbegf* expression by ACSA2$^+$ astrocytes following i.c.v. injection of TNF-α and IL-1β or vehicle. *n* = 3 per group. **b**, RT–qPCR analysis of *Hbegf* expression by primary mouse astrocytes over 24 h following stimulation with TNF-α and IL-1β or vehicle. Unstimulated controls (timepoint 0) were used as reference. *n* = 4 per timepoint. **c**, Relative expression (% of parent) of HB-EGF by primary mouse astrocytes following stimulation with TNF-α and IL-1β or vehicle, quantified by intracellular flow cytometry. *n* = 3 per group. **d**, Representative example of a predicted Arnt::Hif1a-binding site in the *Hbegf* promoter identified by JASPAR[51]. **e**, RT–qPCR analysis of *Hbegf* expression by primary mouse astrocytes stimulated with CoCl$_2$ over 24 h. *n* = 4 per timepoint. **f**, Intracellular flow cytometric analysis of HB-EGF expression by primary mouse astrocytes under pseudohypoxic conditions (CoCl$_2$) for 24 h. *n* = 3 per group. **g**, Schematic depicting the *HBEGF* luciferase promoter–reporter construct, where activation of the *HBEGF* promoter drives the expression of a Gaussia luciferase. **h**, *HBEGF* promoter activity in HEK293T cells co-transfected with an *HBEGF* promoter activity reporter construct and a stably active HIF1α (pHIF1a, *n* = 6) or control vector (*n* = 2). **i**, *HBEGF* promoter activity in HEK293T cells following stimulation with TNF-α and IL-1β, or CoCl$_2$, over 24 h. *n* = 5 per group and timepoint. **j**, RT–qPCR analysis of *Hbegf* expression in primary mouse astrocytes following stimulation with CoCl$_2$, I3S, CH-223191 or a combination. *n* = 3–7 per group. **k**, Flow cytometric quantification of HB-EGF expression in primary mouse astrocytes (ACSA2$^+$GFP$^+$) transduced with a control

(*Gfap::Scrmbl*), AhR (*Gfap::Ahr*), HIF1α (*Gfap::Hif1*) or HB-EGF (*Gfap::Hbegf*) targeting CRISPR–Cas9 vector, quantified by intracellular flow cytometry. *n* = 3 per group. **l**, Relative expression (% of parent) of HB-EGF by astrocytes in *Gfap::Scrmbl* (*n* = 5), *Gfap::Hif1a* (*n* = 3) and *Gfap::Ahr* (*n* = 3) mice quantified by intracellular flow cytometry. **m**, UMAP plot of CNS cells from mice with EAE; data were obtained from Wheeler et al.[20]. MG, microglia; Mac, macrophages; Endo, endothelial cells; Oligo, oligodendrocytes. **n**, NicheNet[28] circle plot depicting ligand–target genes in CNS cells from EAE mice. **o**, GO enrichment analysis (Biological Process) of *Hbegf* target genes. **p**, PANTHER Protein class analysis of *Hbegf* target genes. **q**, Enrichment analysis (Descartes Cell Types and Tissue 2021) of *Hbegf* target genes. **r,s**, Schematic (**r**) and RT–qPCR analysis (**s**) of *Nos2*, *Csf2* and *Bdnf* expression in microglia stimulated with ACM from primary mouse astrocytes activated with TNF-α, IL-1β ± HB-EGF. *n* = 3 per group. Astrocytes were stimulated for 24 h and extensively washed before the addition of fresh medium and collection of ACM 24 h later. **t,u**, Schematic (**t**) and RT–qPCR analysis (**u**) of *Nos2*, *Ccl2* and *Bdnf* expression in astrocytes stimulated with microglia-conditioned medium (MGCM) from primary mouse microglia activated with LPS ± HB-EGF. *n* = 3 per group. Microglia were stimulated for 24 h and extensively washed before the addition of fresh medium and collection of MGCM 24 h later. Data are shown as mean ± s.d. if not indicated otherwise. Unpaired *t*-test in **a**, **c**, **f** and **h**; one-way ANOVA with Dunnett's multiple comparisons test (tested against control) in **b**, **e**, **j**, **k**, **l**, **s** and **u**; two-way ANOVA with Sidak's multiple comparisons test in **i**.

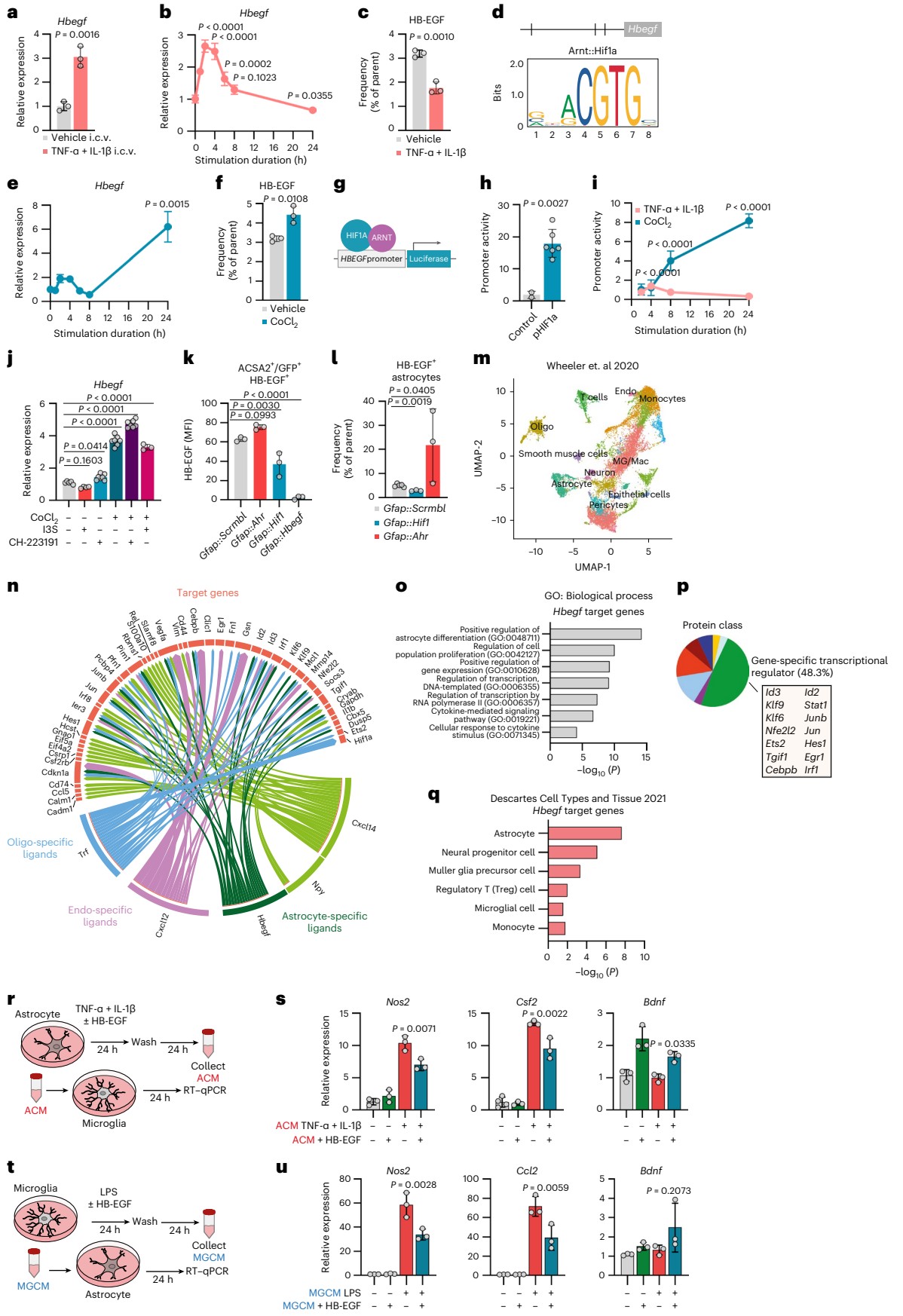

pathology and are potent inducers of astrocyte reactivity[20,22,23]. Indeed, inflammatory challenge resulted in an increase in *Hbegf* expression by astrocytes, supporting our previous observations in the context of EAE (Fig. 4a). This was in contrast to inflammatory challenge of primary astrocytes with the same set of cytokines in vitro, which induced a rapid upregulation of *Hbegf*, followed by a downregulation below baseline over 24 h (Fig. 4b), which was also reflected in reduced protein levels 24 h after stimulation (Fig. 4c). More specifically, we found that both the cytoplasmic and membrane-bound parts of HB-EGF were downregulated in response to stimulation with TNF-α and IL-1β (Extended Data Fig. 5a). This was also seen for the soluble form of HB-EGF, which was rapidly secreted in the supernatant of TNF-α and IL-1β by activated astrocytes within 8 h, but not further produced after prolonged stimulation (Extended Data Fig. 5b–e).

We thus speculated that HB-EGF may be part of a rapid response mechanism regulated by signals provided by the cellular microenvironment in vivo. Bioinformatic analysis of the *Hbegf* promoter identified multiple conserved binding sites of the Hypoxia Inducible Factor 1 Subunit Alpha (HIF1α), the transcriptional master regulator responsible for cellular responses to hypoxic conditions (Fig. 4d and Supplementary Table 3). Indeed, we detected *Hif1a* expression upregulated in astrocytes challenged with TNF-α and IL-1β by i.c.v. injection in vivo (Extended Data Fig. 4f). Furthermore, pseudohypoxia induced by cobalt chloride ($CoCl_2$) significantly increased the production of membrane-bound and soluble HB-EGF (sHB-EGF) in primary human and mouse astrocytes on both messenger RNA and protein levels (Fig. 4e,f and Extended Data Fig. 5g–i). These findings were confirmed by a luciferase-based *HBEGF* promoter–reporter assay (Fig. 4g), in which the activity of the *HBEGF* promoter drives the expression of a Gaussia luciferase. Co-transfection of HEK293T cells with the *HBEGF* luciferase promoter construct and a stable form of HIF1α significantly increased the *HBEGF* promoter activity (Fig. 4h), indicating the direct control of HB-EGF by HIF1α. Moreover, stimulation of transfected HEK293T cells with TNF-α and IL-1β, or $CoCl_2$, recapitulated previous observations and demonstrated a rapid induction, followed by a decrease in *HBEGF* promoter activity in response to TNF-α and IL-1β, while pseudohypoxia resulted in a continuous activation of the *HBEGF* promoter (Fig. 4i).

HIF1α requires dimerization with the aryl hydrocarbon receptor nuclear translocator (ARNT) to exert its functions as a transcription factor[24,25]. Since ARNT is also required by the ligand-activated transcription factor aryl hydrocarbon receptor (AhR)[25,26], and AhR harbors multiple conserved binding sites in the HB-EGF promoter region (Supplementary Table 3), we hypothesized that competition between HIF1α and AhR for ARNT may oppositely control *Hbegf* expression (Extended Data Fig. 5j). Combinatorial stimulation of primary mouse astrocytes with the AhR ligand indoxyl-3-sulfate (I3S), the chemical AhR antagonist CH-223191, the pseudohypoxic reagent $CoCl_2$ or a combination of these supported this notion and revealed synergistic effects on *Hbegf* expression in astrocytes (Fig. 4j). Moreover, lentiviral transduction of primary mouse astrocytes with an AhR (*Gfap::Ahr*) or HIF1α (*Gfap::Hif1*) targeting CRISPR–Cas9 vector confirmed the opposing regulation of HB-EGF by both transcriptional regulators (Fig. 4k and Extended Data Fig. 5k). Finally, in support of these findings, we observed an increase in astrocytic HB-EGF during late-stage EAE when *Ahr* was abrogated (*Gfap::Ahr*), while the HB-EGF expression by astrocytes was reduced in mice with defunct HIF1α signaling (*Gfap::Hif1a*) compared with *Gfap::Scrmbl* controls (Fig. 4l and Extended Data Fig. 5l,m). Overall, these data suggest that HB-EGF production by astrocytes is driven in an HIF1a-dependent manner by hypoxic conditions during CNS inflammation, and is suppressed by AhR signaling.

## HB-EGF has anti-inflammatory properties

Astrocytes are strategically located throughout the CNS and at bordering areas such as the blood–brain barrier (BBB) to communicate with a broad variety of cell types during health and disease[27]. To identify cellular targets and downstream signals of astrocyte-derived HB-EGF, we leveraged a public single-cell RNA-seq (scRNA-seq) dataset of CNS cells at peak of EAE and performed NicheNet analysis, a bioinformatic tool to model cellular interactions by linking ligands to target genes[28] (Fig. 4m,n and Extended Data Fig. 6a–c). *Hbegf* target genes were enriched for pathways associated to astrocyte differentiation, as reported previously[29,30], as well as pathways related to proliferation, cytokine-mediated signaling and transcriptional regulation of astrocytes and other cells in the CNS (Fig. 4o). Indeed, HB-EGF-regulated genes included transcriptional regulators *Stat1* and *Irf1* (Fig. 4p), indicating that HB-EGF may control inflammatory responses. Among potential target cell types, we identified multiple CNS-resident (astrocytes, neuronal progenitor cells, Mueller glia cells, microglia cells) but also CNS-infiltrating cell types (T cells, monocytes) (Fig. 4q).

Based on these in situ observations, we next sought to dissect the effects of HB-EGF on various cell types in vitro. First, we tested the effects of HB-EGF on the inflammatory functions of glial cells. HB-EGF treatment reduced pro-inflammatory gene expression (*Nos2*, *Tnf*, *Ccl2*) and elevated the expression of tissue-protective *Lif* in primary mouse astrocytes activated with TNF-α and IL-1β (Extended Data Fig. 7a). The functional relevance of this anti-inflammatory astrocyte polarization was highlighted by the effects of astrocyte-conditioned medium (ACM) on microglia (Fig. 4r). ACM from stimulated astrocytes that were treated with HB-EGF and washed extensively after stimulation and before collection of ACM suppressed pro-inflammatory gene expression in microglia and induced tissue-protective transcriptional responses (Fig. 4s).

Similarly, direct treatment of LPS-activated microglia with HB-EGF reduced pro-inflammatory signaling (Extended Data Fig. 7a) and the capacity to activate astrocytes via secreted factors (Fig. 4t,u).

**Fig. 5 | HB-EGF exerts anti-inflammatory and tissue-protective effects on CNS-resident and -infiltrating cell types. a**, RT–qPCR analysis of *Il1b*, *Cd68*, *Tnf* and *Lif* expression in microglia pre-activated with LPS for 8 h and stimulated with ACM derived from pseudohypoxic ($CoCl_2$) astrocytes, where astrocyte-derived HB-EGF was blocked by an anti-HB-EGF (αHB-EGF) antibody for 24 h. *n* = 3/4 per group. **b**, RT–qPCR analysis of *Icam1* expressed by primary mouse BMVECs stimulated with TNF-α, IFN-γ ± HB-EGF or vehicle. *n* = 3 per group. **c**, Schematic of i.c.v. injection of TNF-α and IL-1β ± HB-EGF or vehicle, followed by intracellular flow cytometry after 24 h. **d**, MFI of GM-CSF production by CD45$^{int}$CD11b$^+$ microglia (left) and CD45$^{hi}$CD11b$^+$Ly6C$^+$ monocytes (right) following i.c.v. injection of TNF-α and IL-1β ± HB-EGF or vehicle. *n* = 3 per group. **e**, Quantification of survival and the expression (% of parent) of differentiation markers by primary mouse oligodendrocytes at day 5 of culture in the presence of HB-EGF, T3, PDGF/FGF or vehicle quantified by flow cytometry. *n* = 11 per group. **f**, Representative scatter plots depicting PDGFRα and O4 expression by primary mouse oligodendrocytes at day 5 of culture in the presence of HB-EGF, T3, PDGF/FGF or vehicle. **g**, RT–qPCR analysis of *Ptprz1*, *Pdgfra*, *Plp* and *Mbp* expression by O4$^+$ oligodendrocytes following i.c.v. injection of TNF-α, IL-1β ± HB-EGF. *n* = 3 per group. **h,i**, Immunostaining (**h**) and quantification (**i**) of Olig2$^+$ cells in optic nerves stimulated with IFN-γ ± HB-EGF or vehicle. *n* = 5–7. Scale bar, 50 μm. **j–l**, Schematic (**j**), fluoromyelin staining (**k**) and quantification (**l**) of LPC-induced demyelination in the corpus callosum. *n* = 28. Scale bar, 400 μm. Data are shown as median with the 25th and 75th percentiles. **m**, Representative scatter plots of neuronal cells stained with Annexin V (A-V) and propidium iodide (PI) following stimulation with TNF-α ± HB-EGF or vehicle. **n**, Quantification of early apoptotic (A-V$^+$PI$^-$), late apoptotic (A-V$^+$PI$^+$) and necrotic (A-V$^-$PI$^+$) neuronal cells following stimulation with TNF-α ± HB-EGF, or vehicle. *n* = 4 per group. **o,p**, Immunostaining (**o**) and quantification (**p**) of RBPMS$^+$ retinal ganglion cells in retinae stimulated with IFN-γ ± HB-EGF or vehicle. *n* = 11–19 per group. Scale bar, 50 μm. Data are shown as mean ± s.d. One-way ANOVA with Dunnett's (tested against control) or Tukey's multiple comparisons test if not indicated otherwise. Unpaired *t*-test in **l**.

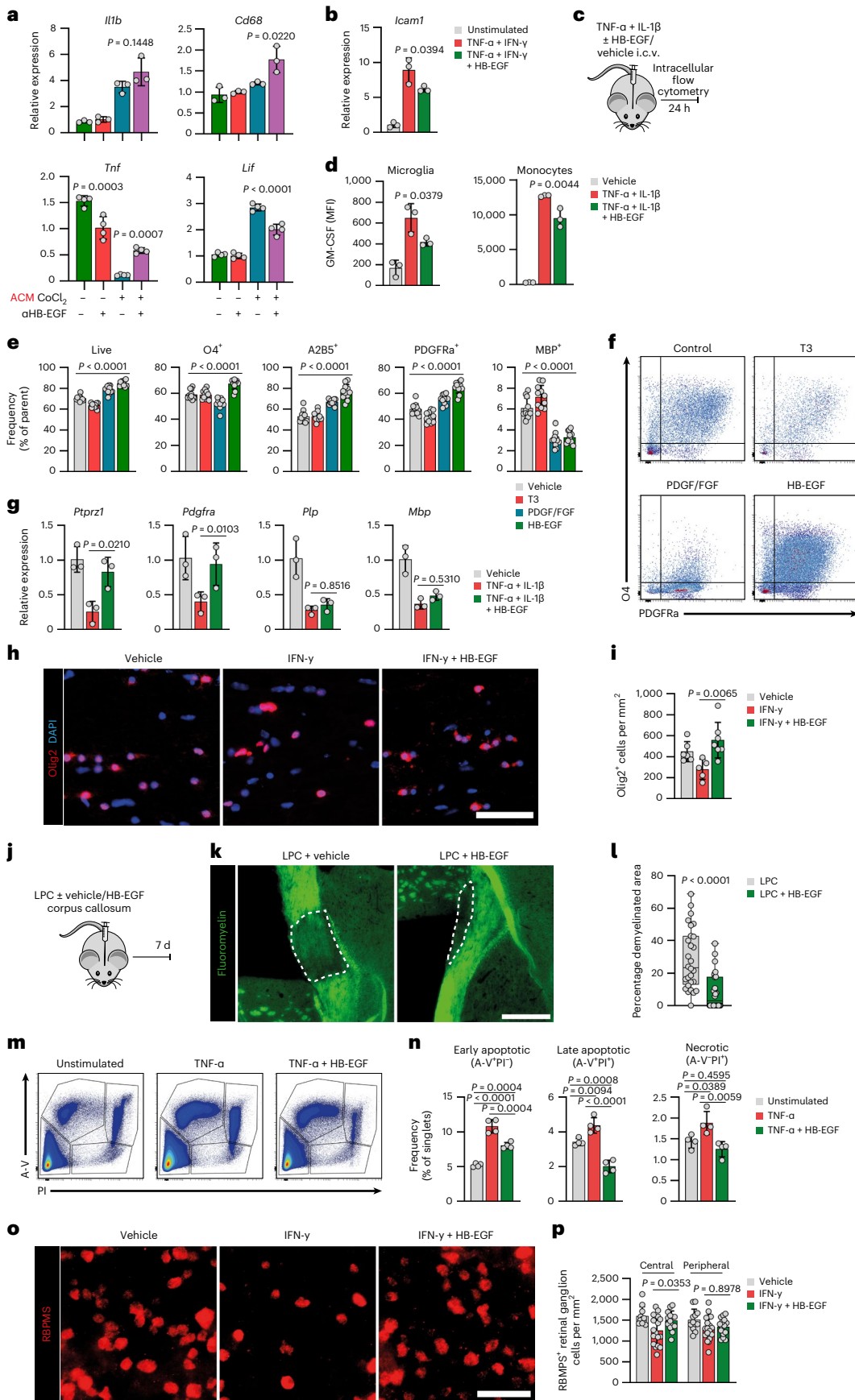

To confirm the relevance of HB-EGF in astrocyte–microglia crosstalk, we stimulated LPS-activated microglia with ACM derived from hypoxic astrocytes in the presence of an anti-HB-EGF (αHB-EGF) blocking antibody or isotype control. Indeed, blockade of HB-EGF increased the expression of the pro-inflammatory genes *Il1b*, *Cd68* and *Tnf*, while reducing the expression of the trophic factor *Lif* by microglia, overall supporting the idea that astrocyte-derived HB-EGF directly controls the inflammatory potential of microglia (Fig. 5a).

Based on our observation that decreased HB-EGF production by astrocytes during late EAE reduced endothelial cell numbers and led to an increase in myeloid and CD4 T cell infiltration, we hypothesized that astrocyte-derived HB-EGF may affect BBB properties by modulating the infiltration of peripheral immune cells at endothelial borders. Indeed, while HB-EGF had no direct effect on the polarization of CD4 T cells into $T_H17$, $T_H1$ or $T_{Reg}$ subtypes (Extended Data Fig. 7b), stimulation of primary brain microvascular endothelial cells (BMVCs) with HB-EGF following inflammatory challenge with TNF-α and IFN-γ decreased the expression of *Icam1*, an adhesion molecule that facilitates leukocyte endothelial transmigration (Fig. 5b).

Furthermore, HB-EGF reduced the costimulatory capacity of monocytes, while it had no effect on their migratory potential (Extended Data Fig. 7c), supporting the notion that HB-EGF may indirectly limit T cell-driven CNS pathology through co-regulatory pathways and BBB properties. To corroborate these observations and to investigate the effect of HB-EGF on glial and myeloid cells in vivo, we co-administered TNF-α, IL-1β and HB-EGF into the lateral ventricle of naive mice and analyzed the production of pro-inflammatory cytokines by intracellular flow cytometry (Fig. 5c). Indeed, HB-EGF significantly reduced the production of GM-CSF by both microglia and infiltrating myeloid cells (Fig. 5d and Extended Data Fig. 7d), altogether suggesting that HB-EGF exerts anti-inflammatory functions through multiple CNS-resident and -nonresident cell types.

### HB-EGF has tissue-protective properties

Signaling through members of the ErbB family modulates oligodendrocyte differentiation and myelinating capacities[31]. Based on our observations of increased myelin loss and changes in OLC subtypes in *Gfap::Hbegf* mice, we speculated that HB-EGF may affect the differentiation of oligodendrocyte precursor cells (OPCs) through its binding partners ErbB1 and ErbB4. To investigate the effects of HB-EGF on OLC differentiation in noninflammatory conditions, we differentiated primary mouse OPCs in the presence of HB-EGF, thyroid hormone (3,3′,5-triiodo-l-thyronine (T3)), PDGF/FGF or vehicle and analyzed their maturation status by flow cytometry at day 5 of the culture. The presence of HB-EGF during differentiation significantly increased the number of live cells after 5 d compared with all control conditions (Fig. 5e,f). Moreover, HB-EGF boosted the expression of the OPC markers A2B5, PDGFRα and O4, while the expression of the mature oligodendrocyte marker MBP was reduced, overall suggesting that HB-EGF favors OPC and premyelinating OLC states (Fig. 5e,f).

Next, we tested the effect of HB-EGF on oligodendrocyte maturation in the context of CNS inflammation following i.c.v. injection of TNF-α and IL-1β. Co-administration of HB-EGF increased the expression of OPC markers *Ptprz1* and *Pdgfra* by oligodendrocytes, supporting the notion that HB-EGF promotes early maturation stages (Fig. 5g). Furthermore, we detected increased Olig2+ cell numbers in an ex vivo optic nerve explant model following stimulation with HB-EGF and IFN-γ (Fig. 5h,i). This increase was particularly prominent in the rostral part of the optic nerve, which is in close proximity to the retinal ganglion cell layer (Extended Data Fig. 7e). Finally, to further confirm the tissue-protective effects of HB-EGF on OLCs in a noninflammatory model of CNS insult, we used the lysolecithin (LPC)-induced model of cortical demyelination. Indeed, co-administration of HB-EGF reduced LPC-induced myelin loss in the corpus callosum 7 d post injection (Fig. 5j–l), collectively demonstrating that HB-EGF supports OPC

survival and myelination in the context of neuroinflammation and demyelination.

Recent data indicate that the loss of neurotrophic support is a defining feature of early stages of neurodegeneration, closely associated to disease progression in the context of neuroinflammation[32]. Since we identified neural progenitor cells among the potential responder cell types of astrocyte-derived HB-EGF, we next evaluated the effect of HB-EGF on neuronal survival. Stimulation with HB-EGF rescued the pro-apoptotic effects of TNF-α and decreased the number of early apoptotic, late apoptotic and necrotic neuronal cells in an in vitro neurotoxicity assay (Fig. 5m,n and Extended Data Fig. 7f), matching similar observations in hippocampal cultures following kainite toxicity[33]. This was confirmed in a retinal explant model, where the addition of HB-EGF increased the number of RBPMS+ retinal ganglion cells following inflammatory challenge with IFN-γ (Fig. 5o,p).

Overall, these data show that astrocyte-produced HB-EGF exerts tissue-protective functions on a variety of CNS-resident and -nonresident cell types, matching our observations in CRISPR-inactivated mice. These functions include the anti-inflammatory regulation of astrocytes, microglia, monocytes and endothelial cells; effects on oligodendrocyte maturation and myelinating capacities; and trophic effects on neurons.

### Epigenetic regulation of astrocyte-derived HB-EGF

Based on our observations of decreased *Hbegf* expression in astrocytes during LSW of EAE, as well as in patients with a second inflammatory relapse, we speculated that activation of astrocytes under inflammatory conditions may induce hypermethylation of the *Hbegf* promoter, interfering with transcription factor binding and ultimately leading to a suppression of *Hbegf* expression. In line with this concept, quantitative PCR (qPCR) with reverse transcription (RT–qPCR) of sorted astrocytes revealed an upregulation of DNA-methyltransferases (DNMTs) during acute stages of EAE (Fig. 6a,b). Thus, we performed whole genome bisulfite sequencing (WGBS) of naive and EAE astrocytes at peak of disease. Concomitant with observations from a previous study[20] (Extended Data Fig. 7g), astrocytes displayed increased promoter and exon DNA methylation during the peak of EAE (Extended Data Fig. 7h). Moreover, WGBS identified increased promoter methylation at Hif1a:Arnt binding sites in the *Hbegf* locus of peak EAE astrocytes (Fig. 6c and Extended Data Fig. 7i), supporting the notion that the combination of reduced hypoxic conditions and increased promoter methylation at HIF1α binding sites may drive the decrease in *Hbegf* expression during progressive stages of autoimmune CNS inflammation. This was confirmed by high-resolution melt analysis (HRM) of bisulfite-converted genomic DNA (gDNA) from fluorescence-activated cell sorting (FACS)-sorted astrocytes obtained from EAE mice using bisulfite-specific primers that span HIF1α binding sites in the *Hbegf* promoter region (Extended Data Fig. 7j). Methylation analysis revealed that HIF1α binding regions of the *Hbegf* promoter are hypermethylated in astrocytes during both peak and recovery stages of EAE, possibly explaining the reduced expression of the protective growth factor during late stages (Fig. 6d).

Next, we tested whether this hypermethylation in the *Hbegf* promoter is dependent on the pro-inflammatory activation of astrocytes. In line with the downregulation of *Hbegf* on mRNA and protein levels after a single stimulation of primary mouse astrocytes with TNF-α and IL-1β (Fig. 4), we observed an increase in *Hbegf* promoter methylation following pro-inflammatory activation by HRM and 5-methyl-cytosine (5-mC) chromatin immunoprecipitation (ChIP) analysis (Fig. 6e–g). This increase in hypermethylation around HIF1α binding sites in the *Hbegf* promoter was reversed following pre-stimulation with 5-Azacytidine (5-Aza), a clinically approved chemotherapeutic agent and inhibitor of DNMTs (Fig. 6e,f). Indeed, pre-treatment of activated primary mouse astrocytes with 5-Aza boosted the expression of HB-EGF expression (Fig. 6h), indicating that therapeutic hypomethylation has the potential to elevate endogenous HB-EGF expression by astrocytes.

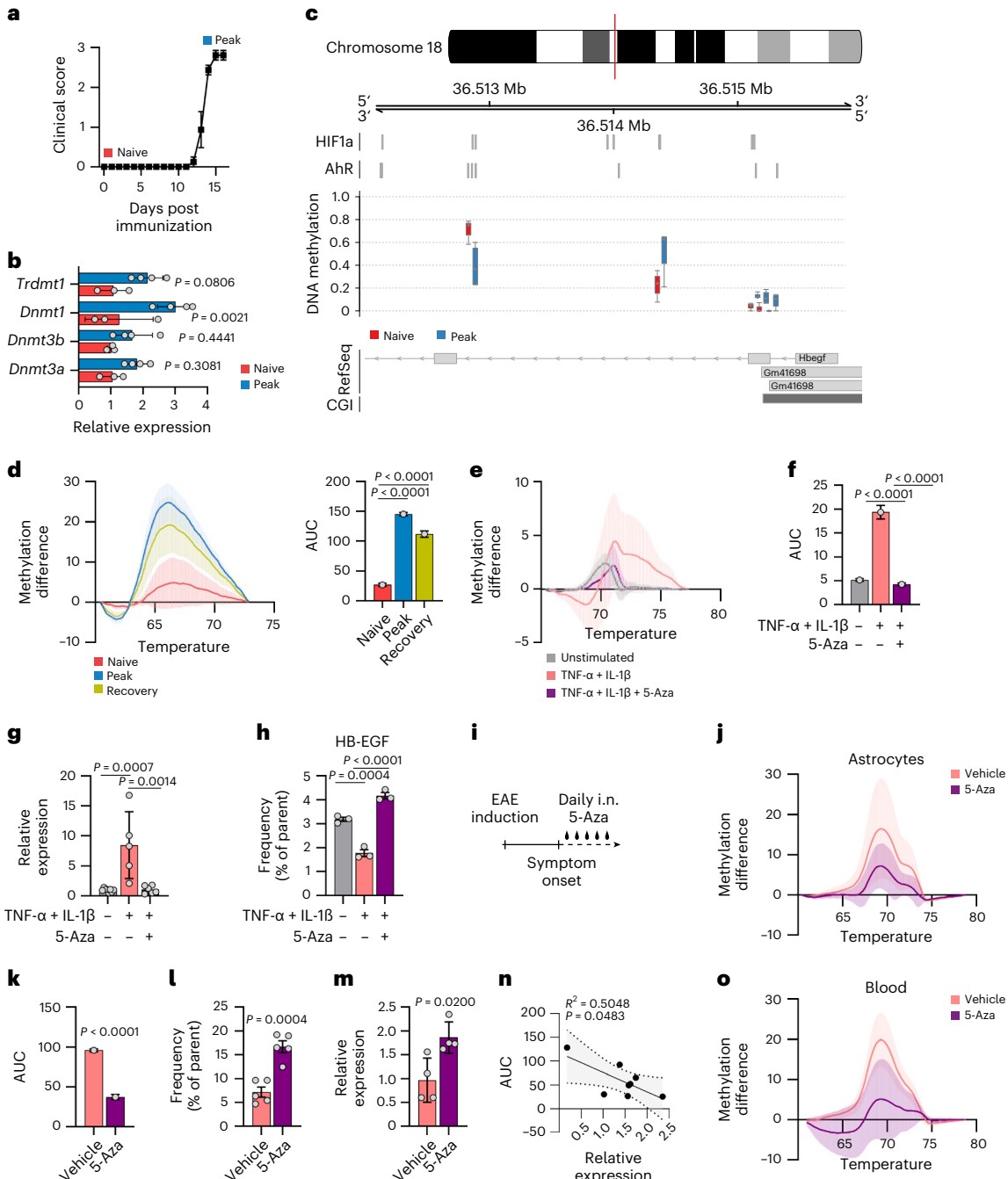

**Fig. 6 | Epigenetic mechanisms govern HB-EGF expression during autoimmune CNS inflammation. a**, EAE development and timepoints (peak, naive) used for WGBS of astrocytes isolated by FACS. *n* = 4 per timepoint. **b**, RT–qPCR analysis of DNA-methyltransferases (DNMT) expressed in FACS-isolated astrocytes during peak EAE (*n* = 4) and naive (*n* = 3) stages. **c**, Quantification of DNA methylation at HIF1α and AhR binding sites in the *Hbegf* locus in astrocytes during peak EAE (*n* = 4) and naive (*n* = 4) stages. **d**, HRM with difference plot and area under the curve (AUC) of bisulfite-converted gDNA of the *Hbegf* promoter region from FACS-sorted astrocytes obtained from the CNS of naive or EAE mice at peak or recovery stages. *n* = 4 (3 replicates per sample) per timepoint. Group means were used for AUC analysis. **e,f**, HRM with difference plot (**e**) and AUC analysis (**f**) of methylation at HIF1α binding sites in the *Hbegf* promoter in mouse following stimulation with TNF-α and IL-1β ± 5-Aza. *n* = 4 per group. Group means were used for AUC analysis. **g**, 5-mC ChIP of the *Hbegf* promoter region in primary mouse astrocytes following stimulation with TNF-α and IL-1β ± 5-Aza. *n* = 5–8 per group. **h**, Intracellular flow cytometric analysis of

HB-EGF expression by primary mouse astrocytes following stimulation with TNF-α and IL-1β ± the hypomethylating agent 5-Aza. *n* = 3 per group. **i**, Schematic of daily intranasal treatment with 5-Aza during EAE. **j**–**m**, HRM with difference plot (**j**), AUC analysis (**k**), relative levels (% of parent) of HB-EGF by flow cytometry (**l**) and relative mRNA expression of *Hbegf* by RT–qPCR in ACSA2⁺ astrocytes (**m**) from vehicle- or 5-Aza-treated mice during late-stage EAE. *n* = 4/5 per group. Group means were used for AUC analysis. **n**, Linear regression analysis with 95% confidence intervals of methylation in *Hbegf* promoter regions by AUC and relative *Hbegf* expression in ACSA2⁺ astrocytes obtained from vehicle- or 5-Aza-treated mice during late-stage EAE. *n* = 8. **o**, HRM analysis of methylation in the *Hbegf* promoter of cells obtained from whole blood of vehicle- or 5-Aza-treated mice during late-stage EAE. *n* = 4 per group (3 replicates per sample). Data are shown as mean ± s.d. if not indicated otherwise. Data are shown as mean ± s.e.m. in **a**. Two-way ANOVA with Sidak's multiple comparisons test in **b**; one-way ANOVA with Dunnett's multiple comparisons test (tested against control) in **d**, **f**, **g** and **h**; unpaired *t*-test in **k**, **l** and **m**. i.n., intranasal; Mb, megabases.

To test the therapeutic applicability of 5-Aza treatment in the context of autoimmune neuroinflammation, we induced EAE in wild-type mice and intranasally administered 5-Aza on a daily basis starting at symptom onset (Fig. 6i). Intranasal 5-Aza treatment successfully reduced *Hbegf* promoter hypermethylation and increased the expression of HB-EGF by astrocytes on protein and mRNA levels (Fig. 6j–m and Extended Data Fig. 7k,l). This was supported by a negative correlation between the extent of hypermethylation and the relative expression of *Hbegf* by astrocytes in vehicle- and 5-Aza-treated mice (Fig. 6n), supporting the idea of epigenetically mediated suppression of *Hbegf* expression during late stages of CNS inflammation. In addition to the hypomethylating effects of 5-Aza treatment in astrocytes, we observed a decrease in *Hbegf* promoter methylation in cells obtained from circulating cells, indicating that the epigenetic regulation of HB-EGF in response to autoimmune inflammation may not be specific to the CNS (Fig. 6o and Extended Data Fig. 7m). Of note, no effect was seen in naive mice treated with 5-Aza, suggesting that the effect is dependent on the inflammatory activation of astrocytes (Extended Data Fig. 7n,o). Collectively, these data suggest that inflammation may suppress HB-EGF production by astrocytes through epigenetic mechanisms in the context of EAE.

## Intranasal delivery of HB-EGF limits neuroinflammation

Based on our previous observations of combined anti-inflammatory and tissue-protective properties of HB-EGF on CNS-resident and -nonresident cell types, we next sought to validate these effects in vivo. Therefore, we induced EAE in wild-type mice and intranasally administered recombinant mouse HB-EGF (rmHB-EGF) on a daily basis starting after onset of symptoms (Fig. 7a). Notably, intranasal application has previously been shown to bypass the BBB in rodents and primates for high-molecular proteins by ourselves and others[34,35]. Indeed, intranasal application of rmHB-EGF attenuated disease and improved recovery, along with a reduction in peripheral immune cell infiltration in the spinal cord following peak of disease (Fig. 7b,c). This was further supported by high-dimensional flow cytometric analysis, which revealed a reduction of CD4[+] T cells, CD11c[+] dendritic cells and MHCII[+]Ly6C[+] monocytes particularly in the spinal cords of rmHB-EGF-treated mice compared with controls (Fig. 7d). These findings align with the prevalence of spinal cord pathology in C57Bl/6 mice during acute stages of EAE[36], which in turn directly correspond to the motor dysfunctions observed in the mouse model. In addition to a reduction in pro-inflammatory infiltrates in spinal cords of rmHB-EGF-treated mice, we also observed an increase in O4[+] OLCs in spinal cords and an increase in CD24[+] neuronal lineage cells in brains of rmHB-EGF-treated mice (Fig. 7d), indicating trophic effects of exogenous HB-EGF supplementation on CNS-resident cell types that are important for recovery and regeneration from autoimmune CNS inflammation.

Moreover, rmHB-EGF-treated mice featured a reduction in inflammatory cytokines produced by infiltrating- and CNS-resident cell types in both the spinal cord and brain at day 23 post immunization (Fig. 7e and Extended Data Fig. 8a). To investigate whether intranasal rmHB-EGF administration during acute autoimmune CNS inflammation affects the trafficking of peripheral immune cells into the CNS, we analyzed the immune cell composition of draining axillary lymph nodes. High-dimensional flow cytometry revealed no differences in the cellular composition of axillary lymph nodes in rmHB-EGF- or vehicle-treated mice, indicating that intranasal HB-EGF supplementation primarily affects inflammatory processes within the CNS (Extended Data Fig. 8b).

Since a reduction in endogenous HB-EGF expression by astrocytes was particularly prominent during late stages of EAE, we next performed intranasal rmHB-EGF supplementation throughout recovery stages, starting at symptom onset (Fig. 7f). In line with a decrease in disease severity, MRI of spinal cords demonstrated a reduction in lesion volume during peak and recovery stages in rmHB-EGF-treated mice compared with controls (Fig. 7f–h). Furthermore, corroborating our previous observations at day 23 post immunization, intranasal rmHB-EGF supplementation throughout late stages of EAE increased the number of OLCs, endothelial cells and astrocytes, while the number of infiltrating monocytes, neutrophils, B cells and T cells and their pro-inflammatory cytokine production in the CNS of rmHB-EGF-treated mice was reduced compared with vehicle-treated controls (Extended Data Fig. 8c,d). This was supported by immunostaining of EAE spinal cords, revealing an increase in Olig2[+] cell numbers and myelin in mice treated with rmHB-EGF (Fig. 7i–l), altogether suggesting that intranasal rmHB-EGF exerts dual roles by attenuating CNS inflammation and promoting tissue recovery during acute and late stages of EAE. In line with this notion, ACSA2[+] astrocytes obtained from rmHB-EGF-treated mice were defined by a reduced expression of pro-inflammatory transcripts (*Ccl2*, *Nos2*), and an increased expression of tissue-trophic factors (*Lif*, *Bdnf*), while we detected no changes in the HB-EGF serum levels or the inflammatory phenotype of peripheral immune cells in spleens of mice that received intranasal rmHB-EGF (Fig. 7m and Extended Data Fig. 8e–h). Finally, daily intranasal administration of rmHB-EGF starting at peak of disease attenuated disease severity and promoted recovery (Fig. 7n and Extended Data Fig. 8i), supporting the protective potential of rmHB-EGF during late stages of EAE. Collectively, these data support our previous observations of combined anti-inflammatory and tissue-protective effects of HB-EGF in vivo and highlight the potential of the growth factor in the regulation of autoimmune CNS inflammation.

## Astrocyte-derived HB-EGF in MS

To validate our observations in human pathology, we conducted immunostaining on stereotactic biopsies obtained from patients presenting with early, acute and progressive autoimmune CNS inflammation.

**Fig. 7 | Intranasal administration of recombinant HB-EGF attenuates neuroinflammation. a**, EAE development and linear regression analysis (from treatment start) of mice treated with rmHB-EGF or vehicle starting at symptom onset (day 10 post immunization). Vehicle *n* = 7, rmHB-EGF *n* = 7. Experiments were repeated three times. **b**,**c**, Representative scatter plots (**b**) and quantification (% of live cells; **c**) of CNS-resident cells (gate 1; CD45[-]CD11b[-]), infiltrating lymphocytes (gate 2; CD45[+]CD11b[-]), infiltrating myeloid cells (gate 3; CD45[hi]CD11b[+]) and microglia (gate 4; CD45[int]CD11b[+]) in the spinal cord and brain of EAE mice treated with rmHB-EGF or vehicle analyzed by flow cytometry after peak of disease. Vehicle *n* = 6, rmHB-EGF *n* = 5. **d**, Representation (% of live) of cell populations in the spinal cord (upper) and brain (lower) of vehicle- or rmHB-EGF-treated mice quantified by high-dimensional flow cytometry. NLCs, neuronal lineage cells; Macro, macrophages; Neutro, neutrophils. Vehicle *n* = 6, rmHB-EF *n* = 5. A detailed gating strategy is provided in Extended Data Fig. 3b. **e**, Heatmap of min. max. scaled cytokine expression by astrocytes (Astro), microglia (MG), monocytes (Mono), myeloid cells (Myeloid), CD4[+] T cells and CD8[+] T cells in the spinal cord and brains of vehicle- or rmHB-EGF-treated mice. *n* = 4 per group. A detailed gating strategy is provided in Extended Data Fig. 4e. **f**, EAE development and linear regression analysis (from treatment start) of mice treated with rmHB-EGF or vehicle starting at symptom onset (day 10 post immunization). Vehicle *n* = 5, rmHB-EGF *n* = 6. **g**, Representative MRI images (T1) of spinal cords from vehicle- and rmHB-EGF-treated EAE mice. Scale bar, 2 mm. **h**, Lesion volume in spinal cords quantified by MRI in vehicle- or rmHB-EGF-treated mice during peak (left) or recovery (right) stages of EAE. *n* = 8 per group. **i**,**j**, Immunostaining (**i**) and quantification (**j**) of Olig2[+] cells in the spinal cord of vehicle- or rmHB-EGF-treated mice. *n* = 4 per group. Scale bar, 50 μm. **k**,**l**, Immunostaining (**k**) and quantification (**l**) of fluoromyelin in the spinal cord of vehicle- or rmHB-EGF-treated mice. *n* = 4 per group. Scale bar, 200 μm. **m**, RT–qPCR analysis of *Ccl2*, *Nos2*, *Lif* and *Bdnf* expression by ACSA2[+] astrocytes derived from vehicle- or rmHB-EGF-treated mice during late-stage EAE. *n* = 5/6 per group. **n**, EAE development (left) and linear regression analysis (right, from treatment start) of mice treated with rmHB-EGF or vehicle starting at peak of disease (day 16 post immunization). Vehicle *n* = 9, rmHB-EGF *n* = 6. Data are shown as mean ± s.d. if not indicated otherwise. Data are shown as mean ± s.e.m. in **a**, **f** and **n**. Unpaired *t*-test in **h**, **j**, **l** and **m**; two-way ANOVA with Sidak's multiple comparisons test in **c** and **d**. Details for linear regression analyses are provided in the Methods section.

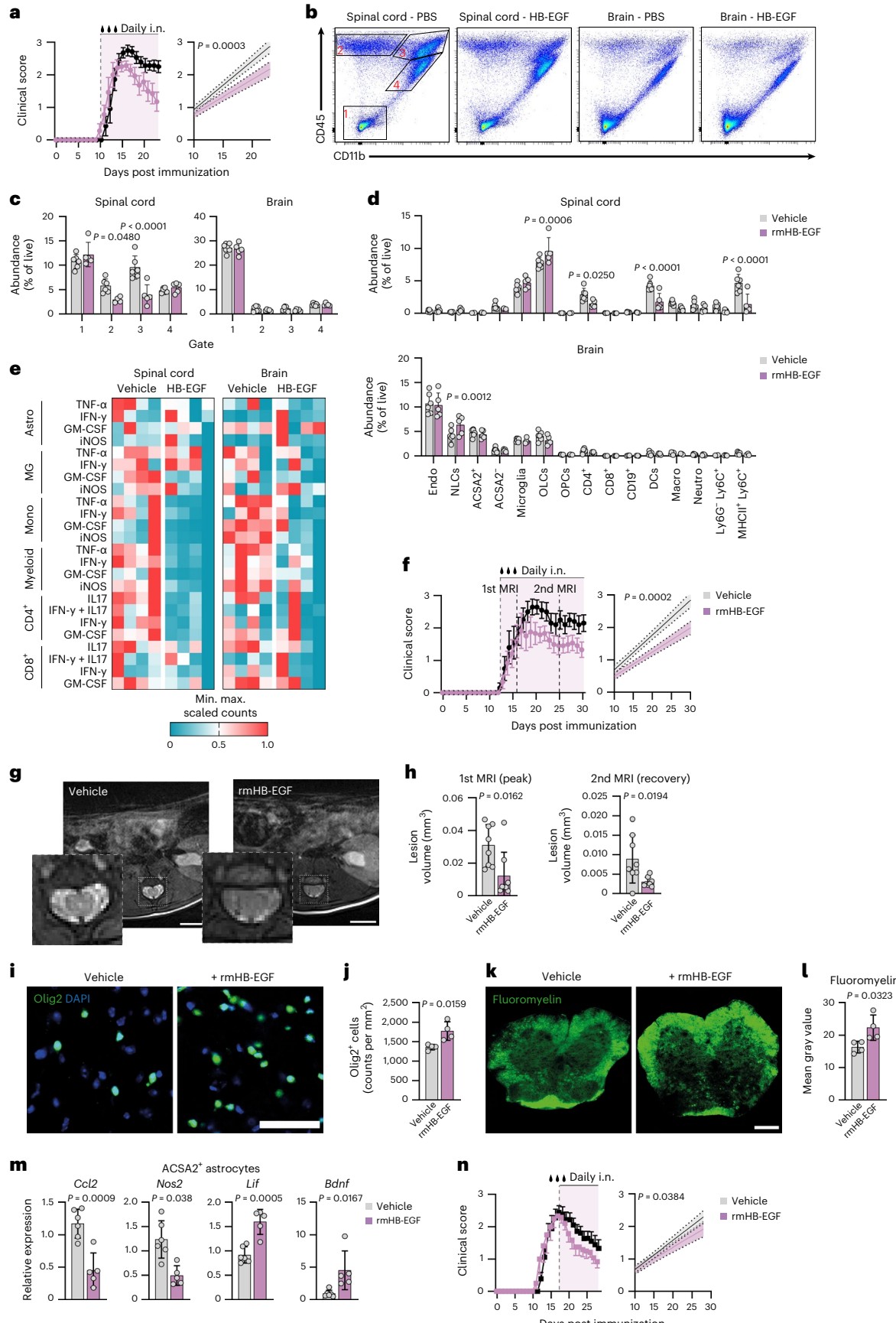

First, we analyzed the expression of HB-EGF by astrocytes in CNS tissue from a patient with a rapidly progressing form of MS (Supplementary Table 4). Despite a relatively short disease duration of 7 months, we observed significant infiltrates of CD3[+] T cells, but no detectable expression of HB-EGF by GFAP[+] astrocytes (Fig. 8a). Similarly, we detected no expression of HB-EGF by GFAP[+] astrocytes in CNS tissue obtained from a patient diagnosed with MS 6 yr ago (Fig. 8a and Supplementary Table 4), matching our observations of decreased HB-EGF expression by astrocytes during late-stage EAE. While these findings were consistent with the reduced levels of sHB-EGF in the CSF of patients with established disease, they did not provide evidence regarding the upregulation of HB-EGF by astrocytes in human pathology. To address this, we aimed to investigate HB-EGF expression during an acute autoinflammatory episode in individuals with no previous relapses. We therefore conducted immunohistochemical analyses on a patient diagnosed with acute disseminated encephalomyelitis (ADEM), an autoimmune neurological disorder characterized by widespread autoimmune neuroinflammation and demyelination following viral or bacterial infections or vaccinations (Supplementary Table 4). Indeed, immunostaining revealed significant infiltration of CD3[+] T cells around lesioned tissue, alongside robust HB-EGF immunoreactivity with a high degree of overlap with GFAP[+] astrocytes (Fig. 8a and Extended Data Fig. 9a).

Next, we sought to investigate whether *HBEGF* promoter hypermethylation is also present in MS. We therefore leveraged two publicly available epigenome datasets of normal-appearing white matter (NAWM) and white matter specimens obtained post-mortem from patients with progressive MS and non-neurological controls[37,38], and analyzed the extent of methylation in glial (NeuN[−]) or bulk nuclei (Supplementary Table 5). Indeed, we detected glia-specific hypermethylation of the *HBEGF* promoter in patients with MS compared with non-neurological controls in proximity to HIF1α binding sites, suggesting that the epigenetic suppression of trophic HB-EGF signaling may be of particular relevance in glial cells during progressive stages of MS (Fig. 8b, Extended Data Fig. 9b and Supplementary Table 5). Nevertheless, similar to our findings in whole blood samples from EAE mice, we detected robust *HBEGF* promoter methylation in patients with RRMS and secondary progressive MS (SPMS), regardless of whether they were undergoing an acute relapse or were in a remission phase (Fig. 8c–f, Extended Data Fig. 9c,d and Supplementary Table 6). Overall, these findings validate the relevance of astrocytic HB-EGF and its epigenetic modulation in human pathology.

## Discussion

Currently available DMTs show limited efficacy to halt neurodegeneration and CNS-intrinsic inflammatory processes in MS[39]. Thus, a better understanding of mechanisms driven by CNS-resident glial cells is of utmost importance to identify novel potential therapeutic strategies[40]. Here, we combined targeted analyses of CSF samples from patients with MS with preclinical mouse models to identify astrocyte-derived HB-EGF as an important mediator of tissue regeneration and anti-inflammatory functions in autoimmune CNS inflammation.

Failure of trophic and anti-inflammatory support during early stages of neurological insult increases the risk of disease progression and may be pivotal for therapeutic intervention[32,41]. Thus, targeting the failure of combined tissue-protective and anti-inflammatory mechanisms early in disease represents a potential strategy to improve clinical outcome. On these lines, here we demonstrate the upregulation of the trophic factor HB-EGF in the CSF of patients with their first demyelinating event (traditionally defined as CIS), potentially representing a protective response mechanism following acute inflammatory CNS insult. This protective mechanism, however, was absent in patients with multiple consecutive demyelinating events (RRMS) and decreased over time in patients with RRMS. The negative correlation between HB-EGF levels in the CSF and the number of CNS lesions in patients with CIS, which has previously been associated as a major risk factor for disease

progression[16], may furthermore support the notion that the combined anti-inflammatory and tissue-supportive effects of HB-EGF during early stages of autoimmune CNS inflammation directly contribute to disease progression and severity. It is important to note however, that the approach used in this study relied on pre-defined factors with already described functions in tissue protection. To overcome this limitation and to identify novel tissue-protective signals that are involved in MS progression, untargeted approaches, such as mass spectrometry, would be necessary, but technically challenging and lacking clinical applicability on larger scales[42].

In agreement with previous reports[43,44], we here identify astrocytes as a cellular source of HB-EGF in the context of CNS inflammation and demonstrate that their spatiotemporal expression of HB-EGF aligns with our observations in patients with MS. More specifically, we show that a highly proliferative astrocyte subtype produces HB-EGF in response to early stages of CNS inflammation, while their expression during later stages decreases despite clinical worsening. This loss of protective functions in astrocytes during late-stage CNS inflammation was not limited to HB-EGF, but also observed for other soluble mediators with protective functions in the context of CNS inflammation, overall suggesting that these mechanisms may depend on similar regulating cues. Albeit the animal model in this study is commonly used as a monophasic model for acute CNS inflammation, the continuing chronic paralysis and the associated loss of protective mediators in late stages may, to a limited extent, recapitulate the alterations observed in patients with CIS and RRMS during relapse[19], a notion supported by our analyses on cellular composition and astrocyte polarization during LSW in comparison with peak or recovery in EAE. Furthermore, we demonstrate that CRISPR–Cas9-mediated genetic perturbation of astrocyte-derived HB-EGF, and therefore their potential to produce the trophic factor during early stages of CNS inflammation, drastically increases inflammation, neurodegeneration and demyelination, ultimately resulting in reduced recovery from acute neuroinflammation, overall supporting the idea that HB-EGF is critical for tissue regeneration.

The rapid upregulation of astrocyte-derived HB-EGF during early stages of autoimmune CNS inflammation and its key role in disease progression can furthermore be explained by its transcriptional regulators. Using combined in vitro and in vivo studies, we demonstrate that HB-EGF in astrocytes is induced by hypoxia master regulator HIF1α, which competes with the ligand-activated AhR for transcriptional control[45]. It is thus conceivable that the primary infiltration by peripheral immune cells and their associated oxygen consumption is accompanied by hypoxic damage in active lesions[46]. This early hypoxic microenvironment may drive, depending on the strength of the hypoxic signal, tissue-supportive mechanisms by astrocytes[47]. In addition to changes in oxygen availability during progressive CNS inflammation, epigenetic changes induced by the activation of astrocytes may alter their tissue-supportive capacities mediated by HB-EGF. Indeed, we here show that inflammatory challenge of astrocytes induces HB-EGF promoter hypermethylation and suppresses *Hbegf* expression in both EAE and MS. We have observed increased methylation of *HB-EGF* promoter CpG sites in proximity to HIF1α binding sites in glial cells of patients with MS. This was in line with increased *HBEGF* promoter methylation in patients with RRMS and SPMS, indicating that the initial inflammatory episode in patients with CIS may epigenetically suppress tissue-supportive mechanisms and ultimately result in disease progression. Indeed, previous studies have reported disease-ameliorating effects of the hypomethylating agent 5-Aza (ref. 48). Therefore, hypomethylation by 5-Aza or targeted demethylation strategies could potentially sustain a tissue-protective and anti-inflammatory environment in patients with MS[49]. Overall, it is conceivable that the epigenetic control of tissue-protective programs and their subsequent failure is a common determinant of disease progression in a variety of autoimmune disorders.

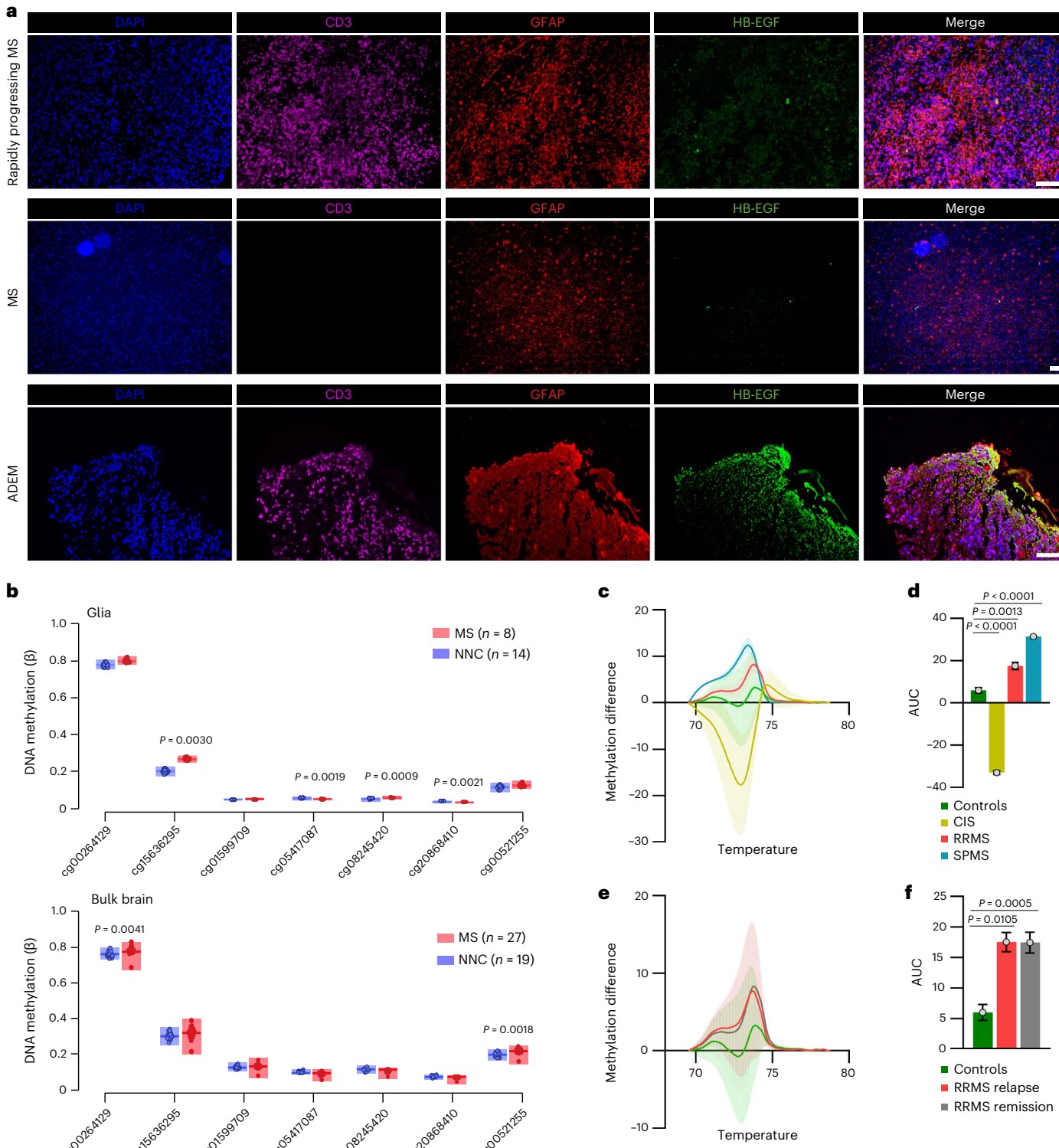

**Fig. 8 | Expression and epigenetic modulation of HB-EGF in the context of MS. a**, Immunostaining of a stereotactic biopsy from a patient with rapidly progressing MS (upper), established, stable MS (middle) and ADEM (lower). DAPI staining (blue) of nuclei, CD3 staining (purple) of T cells, GFAP staining (red) of astrocytes and HB-EGF staining (green). Scale bar, 100 µm. Patient characteristics are provided in Supplementary Table 4. **b**, DNA methylation in the *HBEGF* locus in NeuN⁻ glial cells (upper) and bulk tissue (lower) from patients with MS or non-neurological controls (NNCs). $n$ = 8 MS patients, $n$ = 14 NNCs for the methylation analysis in NeuN⁻ glial cells; $n$ = 27 MS patients, $n$ = 19 NNCs for the methylation analysis in bulk tissue. **c,d**, HRM with difference plot (**c**) and

AUC analysis (**d**) of methylation at HIF1α binding sites in the *HBEGF* promoter of whole blood samples derived from noninflammatory controls ($n$ = 18), CIS patients ($n$ = 3), RRMS patients ($n$ = 79) or SPMS patients ($n$ = 14). Group means were used for AUC analysis. **e,f**, HRM with difference plot (**e**) and AUC analysis (**f**) of methylation at HIF1α binding sites in the *HBEGF* promoter of whole blood samples derived from noninflammatory controls ($n$ = 18), RRMS patients during relapse ($n$ = 14) and RRMS patients during remission ($n$ = 64). Group means were used for AUC analysis. Data are shown as mean ± s.d. Limma moderated $t$-test was used in **b**; one-way ANOVA with Dunnett's multiple comparisons test (tested against controls) was used in **d** and **f**.

In addition to therapeutic hypomethylation of the *HB-EGF* promoter, we report that the exogenous supplementation of HB-EGF at clinically relevant timepoints may effectively counteract the failure of tissue-protective programs in the context of autoimmune CNS inflammation. Intranasal administration of HB-EGF attenuated neuroinflammation and improved recovery in a preclinical mouse model of MS, concomitant with anti-inflammatory and tissue-protective effects on a broad variety of cell types. This is in line with reports of protective effects mediated by HB-EGF in the context of development and ischemic diseases of the CNS, where it functions as a trophic factor, supporting neuronal and oligodendrocyte survival and differentiation[13–15], while its role in CNS inflammation has not been investigated before. The newly discovered relevance of HB-EGF as an anti-inflammatory and tissue-protective factor in the context of autoimmune CNS inflammation may therefore not only drive the development of novel therapeutic strategies for MS, but also for other types of CNS insults, as demonstrated in a study examining neonatal white matter damage[50]. Particularly, the beneficial effects on a broad variety of cell types, including anti-inflammatory effects on astrocytes, microglia, but also myeloid cells, in combination with the trophic effects on oligodendrocytes and neurons, make astrocyte-derived HB-EGF a candidate therapeutic target for a variety of neurological diseases.

Finally, we demonstrate that HB-EGF is upregulated by astrocytes in response to acute inflammatory insult, such as in ADEM, and absent in later stages of MS, recapitulating our findings of altered sHB-EGF CSF concentrations in patients with CIS and RRMS. Together with HBEGF promoter hypermethylation observed in glial and circulating cells of patients with MS, these data further substantiate the significance of our findings and highlight the relevance of HB-EGF in the context of human pathology. Collectively, our results document the relevance of astrocyte-derived HB-EGF for the pathogenesis of MS and provide therapeutic approaches for the treatment of autoimmune CNS inflammation.

## Online content

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

**Mathias Linnerbauer** [1], **Lena Lößlein** [1], **Oliver Vandrey**[1], **Anne Peter**[1], **Yanan Han**[2], **Thanos Tsaktanis**[1], **Emile Wogram**[3], **Maria Needhamsen**[2], **Lara Kular** [2], **Lisa Nagel**[4], **Julia Zissler**[1], **Marie Andert** [5], **Lisa Meszaros**[5], **Jannis Hanspach**[4], **Finnja Zuber**[1], **Ulrike J. Naumann**[1], **Martin Diebold** [3], **Michael A. Wheeler** [6,7], **Tobias Beyer**[8], **Lucy Nirschl**[8], **Ana Cirac**[8], **Frederik B. Laun**[4], **Claudia Günther**[9], **Jürgen Winkler**[3], **Tobias Bäuerle**[4], **Maja Jagodic** [2], **Bernhard Hemmer** [8,10], **Marco Prinz** [3,11,12], **Francisco J. Quintana**[6,7] & **Veit Rothhammer** [1,13] ✉

[1]Department of Neurology, University Hospital Erlangen, Friedrich-Alexander University Erlangen Nuremberg, Erlangen, Germany. [2]Department of Clinical Neuroscience, Karolinska Institutet, Karolinska University Hospital, Center for Molecular Medicine, Stockholm, Sweden. [3]Institute of Neuropathology, Faculty of Medicine, University of Freiburg, Freiburg, Germany. [4]Institute of Radiology, University Hospital Erlangen, Friedrich-Alexander University Erlangen Nuremberg, Erlangen, Germany. [5]Department of Molecular Neurology, University Hospital Erlangen, Friedrich-Alexander University Erlangen Nuremberg, Erlangen, Germany. [6]Ann Romney Center for Neurologic Diseases, Brigham and Women's Hospital, Harvard Medical School, Boston, MA, USA. [7]The Broad Institute of Harvard and MIT, Cambridge, MA, USA. [8]Department of Neurology, Klinikum rechts der Isar, Technical University of Munich, Munich, Germany. [9]Department of Medicine 1, University Hospital Erlangen, Friedrich-Alexander University Erlangen Nuremberg, Erlangen, Germany. [10]Munich Cluster for Systems Neurology (SyNergy), Munich, Germany. [11]Center for Basics in NeuroModulation (NeuroModulBasics), Faculty of Medicine, University of Freiburg, Freiburg, Germany. [12]Signalling Research Centres BIOSS and CIBSS, University of Freiburg, Freiburg, Germany. [13]Deutsches Zentrum Immuntherapie (DZI), University Hospital Erlangen, Erlangen, Germany. ✉e-mail: veit.rothhammer@fau.de

## Methods

### Mice

Mice were housed as previously described[52]. In brief, two to five animals per cage under a standard light cycle (12 h/12 h light/dark) (lights on from 7:00 to 19:00) at 20–23 °C and humidity ~50% with ad libitum access to water and food. Adult female mice 8–12 weeks old and postnatal stage (P) P0–P3 pups were used on a C57Bl/6J background (Jackson Laboratory, no. 000664).

### Human samples

CSF and serum were obtained from the Joint Biobank Munich in the framework of the German Biobank Node. Whole blood was obtained from the Biobank at the University Hospital Erlangen. No additional relevant comorbidities or pharmaceutical treatments were reported in patients or controls. Patients were defined as CIS or RRMS based on Polman et al.[3]. CIS patients did not fulfill diagnostic criteria required for the diagnosis of MS at the time of the study due to the lack of dissemination in time[3]. Controls presented in the clinic with primary headache and no inflammatory condition. All samples were collected and stored at −80 °C using a standardized protocol. The brain tissue used for the analysis of *HBEGF* promoter methylation was obtained from the Multiple Sclerosis and Parkinson's Tissue Bank (Imperial College London), approved by local ethical guidelines. The glial dataset consists of NeuN-negative samples sorted from 38 snap-frozen brain tissue blocks collected within 33 h post-mortem from NAWM tissue of patients with progressive MS ($n = 8$) and white matter tissue of controls ($n = 14$)[37]. The bulk NAWM datasets comprise samples characterized by a lack of inflammatory infiltrates and absence of demyelination of 28 patients with MS, and from the brains of 19 patients without neurological disease and therefore classified as controls[38]. Tissue for immunohistochemical analyses was obtained from the Institute of Neuropathology at the University Freiburg Medical Center. For patient characteristics see the respective supplementary tables.

### EAE

EAE was induced as previously described[52]. In brief, 8–12-week-old female C57Bl/6J mice were immunized using 150 µg of MOG$_{35–55}$ (Genemed Synthesis, 110582) mixed with freshly prepared Complete Freund's Adjuvant (using 20 ml of Incomplete Freund's Adjuvant (BD Biosciences, no. BD263910) mixed with 100 mg of *Myobacterium tuberculosis* H37Ra (BD Biosciences, no. 231141)) at a ratio of 1:1 (v/v at a concentration of 5 mg ml$^{-1}$). All mice received two subcutaneous injections of 100 µl each of the MOG$_{35-55}$/CFA mix. All mice then received a single intraperitoneal injection of 200 ng of pertussis toxin (List Biological Laboratories, no. 180) at a concentration in 200 µl of PBS. Mice received a second pertussis toxin injection at the same concentration 2 d after EAE induction. Mice were monitored and scored daily thereafter. EAE clinical scores were defined as follows: 0, no signs; 1, fully limp tail; 2, hindlimb weakness; 3, hindlimb paralysis; 4, forelimb paralysis; 5, moribund.

### Intranasal delivery of agents

Intranasal delivery of 5-Aza (Sigma Aldrich, no. A2385) or vehicle (PBS) was started once mice developed first symptoms (clinical score ≥ 0.5; approximately 10 d after EAE induction) or in age- and sex-matched naive mice. Either 20 µl of vehicle or of 0.488 mg kg$^{-1}$ (2 mM) 5-Aza (solved in PBS) was applied drop by drop on nostrils. Intranasal delivery of rmHB-EGF (Novus Biologicals, no. 35069) or vehicle (PBS) was started once mice developed first symptoms (clinical score ≥ 0.5; approximately 10 d after EAE induction), or at peak of disease (day 16 post immunization). Either 20 µl of vehicle or of 0.490 mg kg$^{-1}$ (50 µM) rmHB-EGF (solved in PBS) was applied drop by drop on nostrils.

### Lentivirus production

Lentiviral vectors were produced as previously described[18,20,52–54]. Lentiviral vectors were obtained from lentiCRISPRv2 (Addgene. no. 5296155) and lentiCas9-EGFP (Addgene, no. 6359256). CRISPR–Cas9 lentiviral constructs were generated by modifying the pLenti-U6-*sgScramble-Gfap-Cas9-2A-EGFP-WPRE* lentiviral backbone. The *Gfap* promoter is the ABC$_1$D GFAP promoter. sgRNAs were substituted through a three-way cloning strategy using the following primers: U6-PCR-F 5′-AAAGGCGCGCCGAGGGCCTATTT-3′, U6-PCR-R 5′-TTTTTTGGTCTCCCGGTGTTTCGTCCTTTCCAC-3′, cr-RNA-F 5′-AAAAAAGGTCTCTACCG(sgRNA)GTTTTAGAGCT-AGAAATAGCAAGTT-3′, cr-RNA-R 5′-GTTCCCTGCAGGAAAAAAG-CACCGA-3′. Products were amplified using Phusion Master Mix (Thermo Fisher Scientific, no. F548S) and purified using the QIAquick PCR Purification Kit (Qiagen, 28104), followed by digestions using DpnI (NEB, no. R0176S), BsaI-HF (NEB, no. R3535/R3733), AscI (NEB, no. R0558) or SbfI-HF (NEB, no. R3642). Ligations were performed overnight at 16 °C using T4 DNA Ligase Kit (NEB, no. M0202L). Ligations were transformed into NEB Stable Cells (NEB, no. C3040) at 37 °C and single colonies were picked the following day. Plasmid DNA (pDNA) was isolated using QIAprep Spin Miniprep Kit (Qiagen, no. 27104) and the lentiviral plasmids were transfected into HEK293FT cells according to the ViraPower Lentiviral Packaging Mix protocol (Thermo Fisher Scientific, no. K497500) with pLP1, pLP2 and pseudotyped with pLP/VSVG. Medium was changed the next day, and lentivirus was collected 48 h later and concentrated using Lenti-X Concentrator (Clontech, no. 631231) according to the manufacturer's protocol. Concentrates were resuspended in 1/100 of the original volume in PBS. Lentiviral titers were determined using the Lentivirus titration kit (ABM, no. LV900-ABM) according to the manufacturer's instruction.

### I.c.v. delivery of lentivirus

Delivery of lentiviruses via i.c.v. injection was performed as described previously[18,20,52,53]. In brief, mice were anesthetized using 1% isoflurane mixed with oxygen. Heads were shaved and cleaned using 70% ethanol and lidocain gel followed by a medial incision of the skin to expose the skull. The ventricles were targeted bilaterally using the coordinates: ±1.0 (lateral), −0.44 (posterior), −2.2 (ventral) relative to Bregma. Mice were injected with approximately 10$^7$ total IU of lentivirus delivered by two 10-µl injections using a 10-µl Hamilton syringe (Sigma Aldrich, no. 20787) on a Stereotaxic Alignment System (Kopf, no. 1900), sutured and permitted to recover in a separate clean cage. Mice received a subcutaneous injection of 1 mg kg$^{-1}$ meloxicam post i.c.v. injection and 48 h later. Mice were permitted to recover for between 4 and 7 d before induction of EAE. CRISPR–Cas9 sgRNA sequences were designed using a combination of the Broad Institute's sgRNA GPP Web Portal (https://portals.broadinstitute.org/gpp/public/analysis-tools/sgrna-design) and Synthego (https://design.synthego.com/#/validate). sgRNAs used in this study were: *Ahr* 5′-TTGACTTAATTCCTTCAGCG-3′; *Hif1a* 5′- GCTAACAGATGACGGCGACA-3′; *Hbegf* 5′-GGTTTGTGG ATCCAGTGGGA-3′; *Scrmbl* 5′-GCACTACCAGAGCTAACTCA-3′.

### Lentiviral transduction of primary mouse astrocytes

Lentiviral transduction of primary mouse astrocytes was performed according to the lentiviral spinfection protocol by the Broad Institute (https://portals.broadinstitute.org/gpp/public/dir/download?dirpath=protocols/production&filename=Optimization_of_Lentiviral_Spinfection_Oct2018.pdf). In brief, primary mouse astrocytes were cultured and purified as previously described and plated in 12-well culture plates. Astrocytes were transduced with approximately 10$^7$ IU of lentivirus using 4 µg ml$^{-1}$ polybrene (Sigma Aldrich, no. TR-1003-G). Medium was changed the following day and HB-EGF expression was assessed 48 h after transduction.

## Primary mouse astrocyte and microglia cultures

Primary glial cultures were prepared as previously described[52]. In brief, brains of mice aged P0–P3 were dissected into PBS on ice. Brains of 6–8 mice were pooled, centrifuged at 500$g$ for 10 min at 4 °C and resuspended in 0.25% Trypsin-EDTA (Thermo Fisher Scientific, no. 25200-072) at 37 °C for 10 min. DNase I (Thermo Fisher Scientific, no. 90083) was added at 1 mg ml$^{-1}$ to the solution, and the brains were digested for 10 min or more at 37 °C. Trypsin was neutralized by adding DMEM + GlutaMAX (Thermo Fisher Scientific, no. 61965026) supplemented with 10% FBS (Thermo Fisher Scientific, no. 10438026) and 1% penicillin/streptomycin (Thermo Fisher Scientific, no. 10500064), and cells were passed through a 70-µm cell strainer. Cells were centrifuged at 500$g$ for 10 min at 4 °C, resuspended in DMEM + GlutaMAX with 10% FBS and 1% penicillin/streptomycin and cultured in T75 flasks (Sarstedt, no. 83.3911.002), pre-coated with 2 µg ml$^{-1}$ poly-L-lysine (PLL, Provitro, no. 0413), at 37 °C in a humidified incubator with 5% CO$_2$ for 5–7 d until confluency was reached. Mixed glial cells were shaken for 30 min at 180 r.p.m., the supernatant was collected and the medium was changed, and then cells were shaken for at least 2 h at 220 r.p.m. and the supernatant was collected and the medium was changed again. CD11b$^+$ microglia were isolated from the collected supernatant using the CD11b Microbead Isolation kit (Miltenyi, no. 130-049-601) according to the manufacturer's instruction. For stimulation experiments, astrocytes and microglia were detached using TrypLE (Thermo Fisher Scientific, no. 12604013) and seeded in PLL-coated 48-well plates (Sarstedt, no. NC1787625) at a density of 150,000 cells per well.

## Primary human astrocyte cultures

Primary human astrocytes were obtained from ScienCell (no. 1800) and cultured according to the manufacturer's instructions. In brief, cells were passaged in astrocyte medium (ScienCell, no. 1801) until confluency and subsequently plated onto plates pre-coated with 2 µg ml$^{-1}$ PLL (Provitro, no. 0413). For stimulation experiments, astrocytes were detached using TrypLE (Thermo Fisher Scientific, no. 12604013) and seeded in PLL-coated 48-well plates (Sarstedt, no. NC1787625) at a density of 150,000 cells per well.

## Production of astrocyte- and microglia-conditioned medium

Primary mouse astrocytes or microglia were activated with the respective stimuli described in the experiments. After 24 h, the stimulation medium was aspirated, the cells were washed extensively with 1 × PBS and fresh medium was added for another 24 h. After an additional 24 h, ACM or microglia-conditioned medium was collected, cleared from debris by centrifugation and used for stimulation. In experiments where an anti-HB-EGF blocking antibody (R&D, no. AF8239) was used, the ACM was supplemented with 1 µg ml$^{-1}$ α-HB-EGF.

## Primary mouse BMVEC culture

For the isolation and culture of primary mouse BMVECs, brains of mice aged 8–12 weeks were dissected. Meninges, brain stem and cerebellum were removed and brains of ten mice were collected in a 50-ml Falcon tube filled with 13.5 ml of DMEM (Sigma Aldrich, no. D5030). Following trituration with a serological pipette, brains were digested in a mixture of 450 µg ml$^{-1}$ Collagenase D (Sigma Aldrich, no. 11088858001) and 15 µg ml$^{-1}$ DNAse I (Roche, no. 11284932001) for 1 h at 37 °C on an orbital shaker at 180 r.p.m. Following digestion, 10 ml of DMEM was added and the tissue suspension was centrifuged at 1,000$g$ for 10 min at 4 °C. The pellet was resuspended in 25 ml of 20% (w/v) BSA-DMEM and centrifuged at 1,000$g$ for 20 min at 4 °C. The myelin and BSA layer were removed and the pellet was resuspended in 9 ml of DMEM containing 1 mg ml$^{-1}$ Collagenase D and 100 µg ml$^{-1}$ DNAse 1, followed by digestion for 1 h at 37 °C on an orbital shaker at 180 r.p.m. During the digestion, a Percoll gradient (19 ml of 1 × PBS, 1 ml of 10 × PBS, 1 ml of FBS, 10 ml of Percoll (GE Healthcare Biosciences, no. 17-5445-01)) was set up by centrifugation at 3,000$g$ for 1 h at 4 °C with no breaks.

After digestion, 10 ml of DMEM was added onto the digestion mixture and cells were centrifuged at 1,000$g$ for 10 min at 4 °C. The pellet was resuspended in 2 ml of DMEM and added onto the Percoll gradient, followed by centrifugation at 700$g$ for 10 min without break. The interphase was collected and transferred into a separate tube, followed by centrifugation at 1,000$g$ for 10 min at 4 °C. BMVECs were resuspended in DMEM + GlutaMAX (Thermo Fisher Scientific, no. 61965026), including 20% FBS (Thermo Fisher Scientific, no. 10438026), 1% penicillin/streptomycin (Thermo Fisher Scientific, no. 10500064), 0.1% Heparin (Sigma Aldrich, no. H3393) and 0.05% basic fibroblast growth factor (bFGF; Peprotech, no. 10018B), and seeded onto Collagen IV (0.4 mg ml$^{-1}$; Sigma Aldrich, no. C0543)- and fibronectin (0.1 mg ml$^{-1}$; Sigma Aldrich, no. F1141)-pre-coated tissue culture flasks. Puromycin (Sigma Aldrich, no. P8833) was added during the first 2 d of culture. Medium was changed every 2–3 d.

## Primary mouse oligodendrocyte cell culture

For primary mouse oligodendrocyte cultures, primary mixed glial cultures were prepared from B6 P0–P2 mice according to a modified version from ref. [55]. In brief, after removing the meninges, forebrains were collected in a rotating C tube with 1 ml of DMEM (PAN Biotech, P04-05410), placed into a gentleMACS Octo Dissociator with heaters and mechanically dissociated. After dissociation, mixed glial cells were seeded onto PDL (Poly-D-lysine hydrobromide; Sigma Aldrich, no. P6407)-coated (5 µg ml$^{-1}$) T75 flasks. Mixed glial cells were cultured for 7 d at 37 °C and 5% CO$_2$ with medium changed every other day, using DMEM supplemented with 20% FBS and 1% penicillin/streptomycin (Thermo Fisher Scientific, no. 15140122). To purify OPCs, flasks containing mixed glial cells were placed on an orbital shaker at 200 r.p.m. and 37 °C for 18 h. The supernatant containing loosely microglia and OPCs was collected in a petri dish and incubated for 20 min at 37 °C, whereby microglia were attaching. Subsequently, remaining OPCs were seeded into PDL- and laminin-coated (1 µg ml$^{-1}$, Roche, no. 11243217001) cell culture plates and treated with medium containing DMEM/F12 + GlutaMAX (Thermo Fisher Scientific, no. 31331-093), 1% penicillin/streptomycin, B27 Supplement (Thermo Fisher Scientific, no. 12587-010) and N2 Supplement (Thermo Fisher Scientific, no. 17504-044). During the first 24 h, 10 ng ml$^{-1}$ recombinant rat platelet-derived growth factor-AA (PDGF-AA; R&D, no. 1055-AA-050) and 10 ng ml$^{-1}$ recombinant human bFGF (R&D, no. 233-FB) were additionally supplemented. The following day, proliferation medium was removed and fresh medium containing 50 ng ml$^{-1}$ rmHB-EGF (Novus Biologicals, no. 35069), 10 ng ml$^{-1}$ PDGF-AA/bFGF as OPC positive control or 40 ng ml$^{-1}$ T3 (Thermo Fisher Scientific, no. 31331-093) as potent driver of oligodendroglial maturation was added. The differentiation of oligodendrocytes was assessed after 5 d of culture by flow cytometry.

## Neurotoxicity assay

N2A neuronal cells (CCL-131, American Type Culture Collection) were stimulated with 20 ng ml$^{-1}$ TNF-α, 50 ng ml$^{-1}$ HB-EGF (Novus Biologicals, no. 35069) or control medium for 24 h. N2A neuronal cells were detached and washed once with cold 1 × PBS. Live/dead staining was performed with LIVE/DEAD Fixable Aqua Dead Cell Stain Kit (Thermo Fisher Scientific, no. L34957) according to the manufacturer's instructions. In addition, Annexin V Propidium Iodide staining was performed using the APC Annexin V Apoptosis Detection Kit with PI (Biolegend, no. 640932) according to the manufacturer's instruction. Cells were washed once and resuspended in Annexin V Binding buffer before acquisition on a 3L Cytek Northern Lights.

## T cell differentiation

T cells were differentiated as described before[56,57]. In brief, T cells were isolated from spleen and lymph nodes of WT mice. CD4$^+$ T cells were purified using the Naive CD4$^+$ T Cell Isolation Kit (Miltenyi, no. 130104453). Naive cells were cultured at a concentration of

1.5–2.0 × 10[6] per ml. Cells were seeded in the culture plates pre-coated with anti-CD3 antibody (2 µg ml$^{-1}$ for effector T cells, 0.5 µg ml$^{-1}$ for regulatory T cells) and anti-CD28 antibody (2 µg ml$^{-1}$) (clone PV-1; all from Bio X Cell). For the differentiation of $T_H$17 cells, naive T cells were cultured with IL-6 (30 ng ml$^{-1}$, R&D, no. R&D406-ML), TGF-β (3 ng ml$^{-1}$, R&D, no. R&D 240-B) and anti-IFN-γ (10 µg ml$^{-1}$, clone XMG1.2, BioXCell, no. BE0055), for $T_H$1 with IL-12 (10 ng ml$^{-1}$), and anti-IL-4 (10 µg ml$^{-1}$, clone 11B11, BioXCell, no. BE0045), for $T_{Reg}$ TGF-β (3 ng ml$^{-1}$), anti-IL-4 (10 µg ml$^{-1}$) and anti-IFN-γ (10 µg ml$^{-1}$). For $T_H$17 differentiation, cells were supplemented with 10 ng ml$^{-1}$ IL-23 (R&D, no. 1887-ML) after 48 h. HB-EGF (50 ng ml$^{-1}$, Novus Biologicals, no. 35069) was added after 72 h for the rest of the differentiation. After 4 d, production of cytokines was measured by intracellular cytokine staining and subsequent flow cytometry as described above.

### Cell culture experiments and stimulants
The following concentrations were used for stimulation experiments: recombinant mouse TNF-α (Peprotech, no. AF-315-01A) 50 ng ml$^{-1}$, mouse IL-1β (Peprotech, no. 211-11B) 100 ng ml$^{-1}$, mouse HB-EGF (Novus Biologicals, no. 35069) 50 ng ml$^{-1}$, IFN-γ (R&D, no. 485-MI-100/CF) 20 ng ml$^{-1}$, I3S (Sigma, no. I3875-250MG) 50 µg ml$^{-1}$, CH-223191 (Sigma Aldrich, no. C8124) 50 µM, Cobalt(II)-chloride (CoCl$_2$; Sigma Aldrich, no. 232696) 500 µM, 5-Aza (Sigma Aldrich, no. A2385) 10 µM, LPS-EB (InvivoGen, no. tlrl-3pelps) 20 ng ml$^{-1}$, human TNF-α (Peprotech, no. 300-01A) 50 ng ml$^{-1}$, human IL-1β (Peprotech, no. 200-01B) 100 ng ml$^{-1}$, human HB-EGF (Peprotech, no. 100-47) 50 ng ml$^{-1}$.

### I.c.v. injection of cytokines
Delivery of TNF-α, IL-1β ± HB-EGF by i.c.v. injection was performed as described previously[18,20,53]. In brief, mice were anesthetized using 1% isoflurane mixed with oxygen. Heads were shaved and cleaned using 70% ethanol and lidocain gel followed by a medial incision of the skin to expose the skull. The ventricles were targeted bilaterally using the coordinates: ±1.0 (lateral), −0.44 (posterior), −2.2 (ventral) relative to Bregma. Mice were injected with 10 µl of vehicle or cytokine solution containing 100 ng of TNF-α (Peprotech, no. AF-315-01A), IL-1β (Peprotech, no. 211-11B) ± HB-EGF (Novus Biologicals, no. 35069) using a 10-µl Hamilton syringe (Sigma Aldrich, no. 20787) on a Stereotaxic Alignment System (Kopf, no. 1900), sutured and permitted to recover in a separate clean cage. Mice received a subcutaneous injection of 1 mg kg$^{-1}$ meloxicam post i.c.v. injection and were analyzed after 24 h.

### LPC-induced demyelination
LPC-induced demyelination in the corpus callosum was performed as previously described[58]. In brief, a solution of 1% LPC was prepared by mixing LPC (Sigma Aldrich, no. L4129) with sterile 1 × PBS. For injection of LPC + HB-EGF, a solution of 1% LPC containing 100 ng of rmHB-EGF (Novus Biologicals, no. 35069) was prepared in 1 × sterile PBS. For the injection of LPC into the corpus callosum, mice were anesthetized using 1% isoflurane mixed with oxygen. Heads were shaved and cleaned using 70% ethanol and lidocain gel followed by a medial incision of the skin to expose the skull. The corpus callosum was targeted bilaterally using the coordinates: ±1.0 (lateral), +1.3 (anterior), −2.5 (ventral) relative to Bregma. The contralateral side served as internal control. Mice were injected with 2 µl of vehicle (PBS) or LPC ± HB-EGF using a 10-µl Hamilton syringe (Sigma Aldrich, no. 20787) on a Stereotaxic Alignment System (Kopf, no. 1900), sutured and permitted to recover in a separate clean cage. Mice received a subcutaneous injection of 1 mg kg$^{-1}$ meloxicam post i.c.v. injection and 48 h later. Mice were analyzed 7 d post injection.

### Isolation and culture of optic nerves and retinae
Mice were perfused with cold 1 × PBS and optic nerves were removed with the eyeball attached and placed immediately in ice-cold PBS.

Residual tissue was removed, and the optic nerve–retina unit was maintained in 50% Opti-MEM (Thermo Fisher Scientific, no. 31985070), 25% FCS (Thermo Fisher Scientific, no. 10438026) and 25% Hank's Balanced Salt Solution (Thermo Fischer Scientific, no. 14025050), supplemented with 25 mM D-glucose (Sigma Aldrich, no. G8769) and 1% penicillin/streptomycin (Thermo Fisher Scientific, no. 10500064). The tissue was directly transferred into wells of the experimental setup. At the end of the culture period, optic nerves were dissected free from the retina and stimulated with 20 ng ml$^{-1}$ IFN-γ (R&D, no. 485-MI-100/CF).

### Isolation of cells from adult mouse CNS
Cells from the CNS of mice were isolated as previously described[52]. Mice were perfused with cold 1 × PBS and the CNS was isolated and mechanically diced using sterile razors. Brains and spinal cords were processed separately or pooled (if not indicated otherwise) and transferred into 5 ml of enzyme digestion solution consisting of 35.5 µl of papain suspension (Worthington, no. LS003126) diluted in enzyme stock solution (ESS) and equilibrated to 37 °C. ESS consisted of 10 ml of 10 × EBSS (Sigma Aldrich, no. E7510), 2.4 ml of 30% D(+)-glucose (Sigma Aldrich, no. G8769), 5.2 ml of 1 M NaHCO$_3$ (VWR, no. AAJ62495-AP), 200 µl of 500 mM EDTA (Thermo Fisher Scientific, no. 15575020) and 168.2 ml of ddH$_2$O, filter-sterilized through a 0.22-µm filter. Samples were shaken at 80 r.p.m. for 30–40 min at 37 °C. Enzymatic digestion was stopped with 1 ml of 10 × Hi ovomucoid inhibitor solution and 20 µl of 0.4% DNase (Worthington, no. LS002007) diluted in 10 ml of inhibitor stock solution (ISS). 10 × Hi ovomucoid ISS contained 300 mg of BSA (Sigma Aldrich, no. A8806) and 300 mg of ovomucoid trypsin inhibitor (Worthington, no. LS003086) diluted in 10 ml of 1 × PBS and filter-sterilized using a 0.22-µm filter. ISS contained 50 ml of 10 × EBSS (Sigma Aldrich, no. E7510), 6 ml of 30% D(+)-glucose (Sigma Aldrich, no. G8769) and 13 ml of 1 M NaHCO$_3$ (VWR, no. AAJ62495-AP) diluted in 170.4 ml of ddH$_2$O and filter-sterilized through a 0.22-µm filter. Tissue was mechanically dissociated using a 5-ml serological pipette and filtered through a 70-µm cell strainer (Fisher Scientific, no. 22363548) into a fresh 50-ml conical tube. Tissue was centrifuged at 600$g$ for 5 min and resuspended in 10 ml of 30% Percoll solution (9 ml of Percoll (GE Healthcare Biosciences, no. 17-5445-01), 3 ml of 10 × PBS, 18 ml of ddH$_2$O). Percoll suspension was centrifuged at 600$g$ for 25 min with no breaks. Supernatant was discarded and the cell pellet was washed once with 1 × PBS, centrifuged at 500$g$ for 5 min and prepared for downstream applications.

### Isolation of cells from spleens and axillary lymph nodes
Splenic cells were isolated as previously described[52]. Spleens were mechanically dissected and dissociated by passing through a 100-µm cell strainer (Fisher Scientific, no. 10282631). Red blood cells were lysed with ACK lysing buffer (Life Technology, no. A10492) for 5 min and washed with 1 × PBS and prepared for downstream applications. Axillary lymph nodes were mechanically dissected and dissociated by passing through a 100-µm cell strainer (Fisher Scientific, no. 10282631), washed with 1 × PBS and prepared for downstream applications.

### Flow cytometry
Flow cytometry was performed as previously described[52]. Live/dead staining was performed with LIVE/DEAD Fixable Aqua Dead Cell Stain Kit (Thermo Fisher Scientific, no. L34957) according to the manufacturer's instructions. Cells were subsequently stained at 4 °C in the dark for 20 min with flow cytometry antibodies, diluted in FACS buffer (1 × PBS, 2% FBS, 2 mM EDTA). Cells were then washed twice with FACS buffer and resuspended in 1 × PBS for acquisition. Antibodies used in this study were: BV421-CD11b (Biolegend, no. 101235; 1:200), BV480-CD11c (BD, no. 565627, 1:100), BV510-F4/80 (Biolegend, no. 123135, 1:100), BV570-Ly6C (Biolegend, no. 128029, 1:200), BV605-CD80 (BD, no. 563052, 1:100), BV650-CD56 (BD, no. 748098, 1:100), BV650-CD8 (BD, no. 100741, 1:100), PE-eFlour610-CD140a (Thermo Fisher Scientific,

no. 61140180, 1:100), SuperBright780-MHCII (Thermo Fisher Scientific, no. 78532080, 1:200), BV711-CD74 (BD, no. 740748, 1:200), PE-CD45R/B220 (BD, no. 561878, 1:100), PE-CD105 (Thermo Fisher Scientific, no. 12-1051-82, 1:100), PE-Ly6G (BioLegend, no. 127607, 1:200), PE-CD140a (BioLegend, no. 135905, 1:100), PE-O4 (Miltenyi, no. 130117507, 1:100), PE-Ter119 (Biolegend, no. 116207), PE-Ly6C (Biolegend, no. 128007, 1:100), AF488-A2B5 (Novus Biologicals, no. FAB1416G, 1:100), PE-Cy5-CD24 (Biolegend, no. 101811, 1:200), PE-Cy7-CD31 (Thermo Fisher Scientific, no. 25031182, 1:200), PerCP-eFlour710-CD86 (Thermo Fisher Scientific, no. 46086280, 1:100), AF532-CD44 (Thermo Fisher Scientific, no. 58044182, 1:100), PE-Cy5.5-CD45 (Thermo Fisher Scientific, no. 35045180, 1:300), JF646-MBP (Novus Biologicals, no. NBP2-22121JF646, 1:100), APC-Cy7-HB-EGF (Bioss, no. BS-3576R-APC-CY7, 1:100), APC-Cy7-Ly6G (Biolegend, no. 127623, 1:200), AF700-O4 (R&D, no. FAB1326N, 1:200, 1:100), BUV737-CD154 (BD, no. 741735, 1:100), AF660-CD19 (Thermo Fisher Scientific, no. 606019380, 1:100), APC/Fire810-CD4 (Biolegend, no. 100479, 1:100).

For intracellular flow cytometry staining, cells were fixed overnight after surface staining using the eBioscience Foxp3/Transcription Factor Staining Buffer Set (eBioscience, no. 00552300) according to the manufacturer's instructions. For staining of intracellular cytokines, the following antibodies were used: PE-eFlour610-iNOS (eBioscience, no. 61592080, 1:100), BV711-IL17a (Biolegend, no. 506941, 1:100), AF488-HB-EGF (Santa Cruz, no. sc365182 AF488, 1:100), FITC-CXCL12 (Thermo Fisher Scientific, no. MA523547, 1:100), PE-Cy5-FoxP3 (Thermo Fisher Scientific, no. 15-5773-82, 1:200), PE-Cy7-IFNγ (Biolegend, no. 505826, 1:100), PE PerCP-eFlour710-TNF (eBioscience, no. 46732180, 1:200), APC-GM-CSF (eBioscience, no. 17733182), APC-eF780-Ki67 (Thermo Fisher Scientific, no. 506941, 1:100).

For FACS, CNS single-cell suspensions were prepared from brain and spinal cords of EAE mice as previously described[18,20,59]. In brief, all cells were gated on the following parameters: CD105$^{neg}$CD140a$^{neg}$O4$^{neg}$Ter119$^{neg}$Ly-6G$^{neg}$CD45R$^{neg}$. Astrocytes were subsequently gated on: CD11b$^{neg}$CD45$^{neg}$Ly-6C$^{neg}$CD11c$^{neg}$. Compensation was performed on single-stained samples of cells and an unstained control. Cells were sorted on a FACS Aria IIu (BD Biosciences).

### Analysis of multiparameter flow cytometry data
Data were analyzed using the OMIQ platform as previously described[52]. In brief, cells were gated as described previously[54,60]; see also Extended Data Figs. 3a and 8a. For dimensionality reduction, cells were downsampled to an appropriate number per group. Opt-SNE (max. 1,000 iterations, perplexity 30, theta 0.5, verbosity 25) or Uniform Manifold Approximation and Projection (UMAP) (15 neighbors, minimum distance 0.4, 200 Epochs) was performed, followed by PhenoGraph clustering (based on Euclidian distance). SAM[21] was performed on groups using a two-class unpaired approach when two groups were compared (with max. 100 permutations and a false discovery rate cutoff of 0.1).

### RNA isolation and RT−qPCR
RNA isolation and RT−qPCR were performed as previously described[52]. In brief, cells were lysed in 350 µl of RLT buffer (Qiagen) and RNA was isolated using the RNeasy Mini Kit (Qiagen, no. 74004) according to the manufacturer's instructions. Then, 500 ng of RNA of each sample was transcribed into complementary DNA using the High-Capacity cDNA Reverse Transcription Kit (Life Technologies, no. 4368813). Gene expression was assessed by qPCR using the TaqMan Fast Advanced Master Mix (Life Technologies, no. 4444556). The following TaqMan probes were used: *Actb* (Mm02619580_g1), *Ahr* (Mm00478932_m1), *Bdnf* (Mm04230607_s1), *Ccl2* (Mm00441242_m1), *Csf2* (Mm0129-0062_m1), *Cd68* (Mm03047343_m1), *Dnmt1* (Mm01151063_m1), *Dnmt3a* (Mm00432881_m1), *Dnmt3b* (Mm01240113_m1), *Gapdh* (Mm99999915_g1), *Il1b* (Mm00434228_m1), *Icam1* (Mm0051-6023_m1), *Ldha* (Mm01612132_g1), *Lif* (Mm00434762_g1), *Hbegf*

(Mm00439306_m1), *HBEGF* (Hs00181813_m1), *Hif1a* (Mm00468-869_m1), *Mbp* (Mm01266402_m1), *Nos2* (Mm00440502_m1), *Ptprz1* (Mm00478484_m1), *Pdgfra* (Mm00440701_m1), *Plp* (Mm01297-209_m1), *Tnf* (Mm00443258_m1), *Trdmt1* (Mm00438511_m1). qPCR data were analyzed by the delta-delta CT (ΔΔCt) method.

### Isolation of gDNA from mouse and human samples
gDNA from cultured cells, whole blood from EAE mice or whole blood derived from patients with MS and controls was isolated using the DNeasy Blood & Tissue Kit (Qiagen, no. 69506) or the EpiTect LyseAll Lysis Kit (Qiagen, no. 59866) according to the manufacturer's instructions.

### Immunohistochemistry
Immunohistochemistry was performed as previously described[52]. For immunohistochemical analyses, mice were transcardially perfused with cold PBS. For LPC experiments, the mice were additionally perfused with 4% PFA/1 × PBS. The CNS parts (brain, optic system, spinal cord) were dissected and processed for immunofluorescence labeling. The tissue was post-fixed in 4% PFA/1 × PBS at 4 °C for 24 h. After post-fixation, the tissue was dehydrated at 4 °C in 30% sucrose in PBS overnight. By means of liquid nitrogen-cooled 2-methylbutane, the tissue was frozen in tissue-Tek embedding medium and kept at −80 °C for storage. Then, 10-µm (spinal cord, optic nerve)- and 12-µM (brain)-thick cross-cryostat sections (Leica) were obtained on glass slides and stored at −20 °C. For immunohistochemical analyses of HB-EGF and GFAP, spinal cord cross-sections were incubated in acetone for 10 min at −20 °C for post-fixation. After washing the slides in 1 × PBS for 5 min, they were permeabilizated in 0.3% Triton-X for 5 min. After one washing step, the tissue was incubated in blocking buffer (5% BSA/10% donkey serum/0.3% Triton-X/1 × PBS) for 1 h. Slides were incubated overnight at 4 °C with mouse anti-HB-EGF (1:200; Santa Cruz, no. sc365182) and rat anti-GFAP (1:800; Thermo Fischer Scientific, no. 2.2B10) diluted in 1% BSA/1% donkey serum/0.3% Triton-X/1 × PBS. On the following day, three washing steps of 5 min each preceded the incubation with the secondary antibodies for 1 h: donkey anti-rat IgG AF488 (1:500; Thermo Fisher Scientific, no. A21208), donkey anti-mouse IgG AF647 (1:500; Dianova, no. 715-605-151). During the further procedure, sections were washed three times for 5 min and then incubated in Vector Quenching solutions (Biozol, no. VEC-SP-8400-15) for 5 min. After three washing steps, the cross-sections were incubated with DAPI (Sigma, no. D8417) diluted 1:100,000 in antibody solution for 10 min at room temperature and washed five times in 1 × PBS for 3 min. Finally, the slides were cover-slipped with Prolong Gold anti-fade (Thermo Fisher Scientific, no. P36930) and stored at 4 °C for further analysis.

For analysis of SMI32 and Olig2, spinal cord or optic nerve sections were incubated in acetone for 10 min at −20 °C for post-fixation. After washing the slides in 1 × PBS for 5 min, they were incubated in blocking buffer (5% BSA/10% donkey serum/0.3% Triton-X/1 × PBS) for 30 min. Slides were incubated overnight at 4 °C with mouse anti-SMI32 (1:1,000; BioLegend, no. 801701) and rabbit anti-Olig2 (1:200; Abcam, no. ab109186) diluted in 1% BSA/1% donkey serum/0.3% Triton-X/1 × PBS. On the following day, three washing steps of 5 min each preceded the incubation with the secondary antibodies for 1 h: donkey anti-mouse IgG AF488 (1:500; Thermo Fisher Scientific, no. A21202), donkey anti-rabbit IgG AF647 (1:500; Dianova, no. 711605152). During the further procedure, sections were washed three times for 5 min. After this process, the cross-sections were incubated with DAPI diluted 1:100,000 in antibody solution for 10 min at room temperature and washed five times in 1 × PBS for 3 min. Finally, the slides were cover-slipped with Prolong Gold anti-fade and stored at 4 °C for further analysis. For fluoromyelin staining, the Invitrogen FluoroMyelin Green Fluorescent Myelin Staining Kit (Thermo Fisher Scientific, no. F34651) was used according to the manufacturers' instruction.

The retinae were dissected for immunofluorescence retinal flat-mount analysis to quantify ganglion cells. For this, the optic nerve was removed from the eye, which was further opened along the ciliary body. Cornea, vitreous body and lens were removed, and retinal pigment epithelium separated from retina. Retinal tissue was incised four times and placed in a 24-well plate filled with 1 × PBS. After one washing step with 2% Triton-X-100 in 1 × PBS at room temperature, the retinae were frozen in fresh 2% Triton-X-100 in 1 × PBS at −80 °C for storage. For the visualization of retinal ganglion cells, the retinae were used for immunofluorescence free floating labeling against RBPMS. The tissue was thawed and washed two times in 1 × PBS/0.5% Triton-X at room temperature with gentle agitation. The tissue was incubated in 500 µl of blocking solution (1 × PBS/5% BSA/10% donkey serum/2% Triton-X) for 1 h at room temperature. After incubation, the primary antibody was applied to the retinal tissue and incubated overnight at 4 °C with gentle agitation: rabbit anti-RBPMS (1:300; Merck, no. ABN1362) diluted in blocking buffer. After overnight incubation, the retinae were washed with 1 × PBS/2% Triton-X for 5 min and then three times for 10 min with 1 × PBS/0.5% Triton-X with gentle agitation. Secondary antibody was added to the tissue at a concentration of 1:500 and then incubated overnight at 4 °C on a slow rocker: goat anti-rabbit IgG Cy3 (Thermo Fischer, no. A10520). On the third day of staining, the retinae were washed three times for 30 min in 1 × PBS at room temperature and subsequently transferred to microscope slides, ensuring that the ganglion cell layer was facing up. Finally, the slides were cover-slipped with Prolong Gold anti-fade (Thermo Fisher Scientific, no. P36930) and stored at 4 °C for further analysis.

Images of immunofluorescence-labeled sections were acquired using the software Zen 3.0 (blue edition). Stainings were examined using a fluorescence microscope (Axio Observer Z1, Zeiss) at ×5, ×10 or ×20 magnification. Cells were quantified manually in a blinded, unbiased manner by the same investigator. Image processing was performed using Photoshop CS6 (Adobe).

## Immunohistochemistry of human tissue

Formalin-fixed, paraffin-embedded tissue sections of 3-µm thickness were completely deparaffinized in xylene and hydrated before antigen retrieval was performed at 98 °C at pH 9. Next, sections were cooled to room temperature, washed and blocked in PBS containing 5% BSA and 1% Triton-X for 2 h at room temperature. Primary antibodies anti-CD3 (rat, CD3 Monoclonal Antibody (no. 17A2), eBioscience (no. 14003282), 1:100), anti-HBEGF (rabbit, no. LSB1261750, 1:50) and anti-GFAP (chicken, no. ab4674, 1:1,000) were added overnight at 4 °C. After washing in PBS, secondary antibodies were added in PBS containing 5% BSA for 2 h at room temperature. After washes, DAPI was added to stain nuclei for 10 min at room temperature. Sections were mounted using Mowiol (Carl Roth). Images were taken with a ×10 objective (Keyence, BioRevo).

## MRI of mouse spinal cords

All measurements were performed on a preclinical 7T MRI scanner (ClinScan 70/30, Bruker) using a dedicated mouse brain coil. For in vivo imaging, animals were anesthetized using 3% isoflurane and maintained at 1.5% isoflurane. Respiration was monitored by breath sensors and kept constant throughout the entire imaging procedure. Body temperature was stabilized using a heating circulator bath (Thermo Fisher Scientific). An axial T1-weighted spin echo sequence was acquired two times before the administration of contrast agent Gadolinium. The sequence parameters were: field-of-view = 35 × 26.2 × 14 mm$^3$, voxel size = 0.078 × 0.078 × 0.7 mm$^3$, matrix size = 448 × 336 × 20, echo time (TE) = 9 ms, repetition time (TR) = 460 ms, band width (BW) = 205 Hz px$^{-1}$, acquisition time (TA) = 2:39 min s$^{-1}$. After the administration of the contrast agent, an axial T2-weighted turbo spin echo sequence was acquired with a field-of-view of 30 × 30 × 14 mm$^3$, voxel size = 0.094 × 0.094 × 0.7 mm$^3$, matrix size of 320 × 320 × 20,

TE = 53 ms, TR = 3,800 ms, BW = 130 Hz px$^{-1}$, TA = 4:37 min s$^{-1}$. Subsequently, the T1-weighted sequence was repeated four times to measure the contrast agent washout. Difference maps of the T1-weighted images before and after the application of the contrast agent were calculated on the scanner and used for the subsequent manual segmentation to calculate the lesion volume using the MITK workbench[61].

## Multiplex analysis of CSF

The multiplex analysis of selected analytes in the CSF of patients with MS and controls was performed in collaboration with Thermo Fisher Scientific. In brief, the following analytes were measured in CSF without further dilution on the ProcartaPlex Luminex Platform: FGF-2, FGF-21, GAS6, GDNF, HB-EGF, IL-10, IL-33, LIF, CCL-2 (MCP-1), MIF, NGF-β, PDGF-BB, TGF-α, VEGF-A, NSE, S100B, GFAP, LAP, YKL-40 (CHI3L1), CNTF, SCF, Aβ$_{1-42}$, CD44, BACE1, BDNF, TRAIL, CD40L, NRGN. Absolute concentrations were determined using a standard curve. PCA of absolute concentrations and addition clinical parameters was performed using scikit-learn (v.1.1.3) and Python (v.3.5.1). The following parameters were used for PCA in Extended Data Fig. 1: sex, age, Expanded Disability Status Scale (EDSS), therapy, number of cerebral lesions, number of spinal cord lesions, optic neuritis (NNO), number of relapses, disease duration, presence of oligoclonal bands, glucose (mg l$^{-1}$), lactate (mmol l$^{-1}$), EW (mg dl$^{-1}$), albumine (mg l$^{-1}$), IgG (g l$^{-1}$), IgA (g l$^{-1}$), IgM (g l$^{-1}$), Qalb, QIgG, QIgA, QiGM, IgG index, IgA index, IgM index, overall cell count in CSF, CD45 cell count, CD3 cell count, CD4 cell count, CD8 cell count, CD19 cell count, CD19CD138 cell count, CD56 cell count. PCA of absolute concentrations and addition clinical parameters was performed using scikit-learn (v.1.1.3) and Python (v.3.5.1). Clustering by Euclidean distance was performed using the clustermap function from the Python package seaborn (v.0.12) using StandardScaler.

## ELISA

For the analysis of sHB-EGF in the serum of patients with MS and controls, a commercial HB-EGF ELISA kit was used (Thermo Fisher Scientific, no. EHHBEGFX5) according to the manufacturer's instructions. Serum was obtained from the Joint Biobank Munich in the framework of the German Biobank Node. Serum was diluted 1:1. For patient characteristics see the supplementary tables. For the analysis of sHB-EGF in the serum of EAE mice, a commercial HB-EGF ELISA kit was used (Novus Biotech, no. NBP2-62780) according to the manufacturer's instructions.

## Luciferase promoter activity assay

The *HBEGF* promoter activity was assessed using a commercially available promoter–reporter construct (Genecopoeia, no. HPRM20671) according to the manufacturer's instructions. In brief, 30,000 HEK293T cells per well were seeded in a 96-well flat-bottom culture plate and transfected with the Gaussia luciferase reporter construct and a Secreted Alkaline Phosphatase (SEAP) internal control (Genecopoeia, no. SEAP-PA01) using Fugene-HD Transfection Reagent (Promega, no. E2311) as previously described[62]. In brief, a transfection mix consisting of 5 µg of SEAP internal control pDNA, 15 µg of the promoter–reporter construct and 60 µl of the Fugene-HD Transfection Reagent was prepared in 1,000 µl of Opti-MEM and incubated for 15 min at room temperature. In experiments, where a stable HIF1α was transfected into the cells, 5 µg of the pcDNA3 mHIF-1α MYC pDNA (Addgene, no. 44028) was included in the transfection mix. Nontransfected controls were used as reference. After 24 h, the transfection medium was aspirated and cells were stimulated as described for the respective experiments. Following stimulation, the supernatant was collected and cleared from cellular debris by centrifugation. Luciferase activity was assessed using the Secrete-Pair Dual Luminescence Assay Kit (Genecopoeia, no. LF033). Gaussia luciferase activity was normalized to SEAP activity to determine the *HBEGF* promoter activity. Results are represented relative to control samples.

## 5-mC ChIP

Astrocytes were collected according to the ChIP protocol provided by Abcam and as previously described[52]. In brief, glycine was added to a confluent 150-mm$^2$ tissue culture dish at a final concentration of 125 mM for 5 min. Subsequently, cells were washed two times with cold 1 × PBS, and then scraped in 1 × PBS. Cells were pelleted and resuspended in ChIP lysis buffer for 10 min on ice. Samples were sonicated, cell debris pelleted and supernatant used for further immunoprecipitation. Each sample was diluted 1:10 with RIPA buffer (Thermo Fisher Scientific, no. 89900). Primary antibodies were added to the sample for 1 h at 4 °C. Sheared chromatin was immunoprecipitated with prepared A/G beads according to the Abcam protocol overnight at 4 °C with rotation. The next day, the protein-bound magnetic beads were washed once with low-salt buffer, once with high-salt buffer and once with LiCl buffer. Then, 120 µl of elution buffer was added to the samples for 15 min while vortexing gently. Next, 5 M NaCl and RNase A were added to the sample according to the protocol and incubated while shaking overnight at 65 °C. Proteinase K was added for 1 h at 60 °C. DNA was purified using QIAquick PCR Purification Kit (Qiagen, no. 28104). qPCR was performed using Fast SYBR Green Master Mix (Thermo Fisher Scientific, no. 4385612). Anti-IgG immunoprecipitation and input were used as controls. We used mouse anti-5-mC antibody (Abcam, ab10805) and mouse IgG polyclonal isotype control (Abcam, no. ab37355). PCR primers were designed with Primer3 (ref. [63]) to generate 50–150-base pair (bp) amplicons. Primer sequences used were: HBEGF1-F: 5′-CTGAATGCCAACCCAGCC-3′, HBEGF1-R: 5′-GGGCTGATGTGTTTCTTTTCC-3′; HBEGF2-F: 5′-GATACCTAGTGTGGAGCGGG-3′, HBEGF2-R: 5′-GAGTGTGGGTGTAGGGTGAA-3′; HBEGF3-F: 5′-TACACCCACACTCCAGTCAC-3′, HBEGF3-R: 5′-GAATAAGGCTCCGGGGAAGG-3′; HBEGF4-F: 5′-TCCTTCCCCGGAGCCTTATT-3′, HBEGF4-R: 5′-CGCGGTTCGTCTGTCTGTTC-3′; HBEGF5-F: 5′-GGGACACGTGGGAAGGTC-3′, HBEGF5-R: 5′-CGGACAACACTGGACAAGAG-3′.

## Bisulfite conversion and HRM

gDNA was bisulfite converted using the EpiTect Fast DNA Bisulfite Kit (Qiagen, no. 59826) according to the manufacturer's instructions. For the analysis of human samples, an EpiTect PCR Control DNA Set (Qiagen, no. 59695) was used. Following bisulfite conversion, methylation was assessed by methylation-sensitive HRM and DNA sequencing using the MeltDoctor HRM Master Mix (Thermo Fisher Scientific, no. 4415440). For the design of HRM-sensitive bisulfite specific primers (BSP), the Methyl Primer Express Software (Thermo Fisher Scientific, no. 4376041) was used. Following HRM, results were analyzed using the High Resolution Melt Software (Thermo Fisher Scientific, no. A29881, v.3.2). Difference plots normalized the 0% methylated control sample were generated and the area under the curve was calculated using a custom Python script. In addition, amplified samples were analyzed by DNA sequencing by Eurofins genomics.

## WGBS

For WGBS, gDNA of FACS-sorted astrocytes was isolated using the QIAmp DNA Micro Kit (Qiagen, no. 56304) according to the manufacturer's instructions. Bisulfite conversion of 100 ng of isolated gDNA was performed as previously described. Following bisulfite conversion, library preparation and amplification were performed using the QIAseq Methyl Library Kit (Qiagen, no. 180502) according to the manufacturer's instructions. Libraries were analyzed on an Illumina Novaseq (PE 150 bp) in collaboration with BGI Tech Solutions. We first applied the Nf-core/methylseq bioinformatics pipeline[64] to raw fastq files of all samples with parameters --clip_r1 10 --clip_r2 10 --three_prime_clip_r1 10 --three_prime_clip_r2 10 --non_directional --relax_mismatches with the genome version mm10. Then we merged reads that mapped to both strands to specific CpG groups. The numbers of methylated reads dividing the sum of methylated and unmethylated reads were used as

methylation levels. Not assigned (NA) was assigned to methylation levels of CpGs whose coverage was less than 10. CpGs with 10× coverage in at least three samples in each group (Naive, Peak) were used as inputs of Limma[65] to identify the differential methylation positions. The *cis* regulation annotation was downloaded from ENCODE with version mm10 (ref. [66]) and gene annotation was downloaded from UCSC with version mm10 (ref. [67]). Visualizations of differential methylation positions, transcription factor binding sites (HIF1α and AhR), NCBI RefSeq transcripts and CpG islands (CGIs) from UCSC (http://genome.ucsc.edu) were conducted using the Gviz Bioconductor package v.1.42.0 (ref. [68]).

## Bulk RNA-seq

For bulk RNA-seq of astrocytes during the course of EAE, single-cell suspensions were prepared from cortices as previously[52] described. Astrocytes were isolated using the Anti-ACSA-2 MicroBead Kit (Miltenyi, no. 130-097-678) and the pellets flash frozen in liquid nitrogen. RNA isolation and library preparation using Single Primer Isothermal Amplification (SPIA) were performed by BGI Tech Solutions. Libraries were analyzed on a DNBseq (PE 100 bp) by BGI Tech Solutions. Data filtering, removal of adapter contamination and low-quality reads, read filtering and genome mapping were performed by BGI Tech Solutions. Differential expression analysis was performed on normalized counts using R and DESeq2 (1.38.0). A $P < 0.05$ was used to determine differentially expressed genes. Visualization was performed with ggplot2.

## Methylation analyses of human CNS specimens

Methylation analyses of human CNS specimens were performed based on the data published by Kular et al.[37] and Hyunh et al.[38]. DNA methylation of the glial nuclei and the bulk NAWM was assessed using Illumina Infinium Human Methylation EPIC/850K and 450K BeadChips, respectively. Of note, five CpGs at the *HBEGF* locus were specific to EPIC/850K and therefore could not be assessed in the bulk NAWM samples. Quality assessment of Illumina 450K and EPIC data was performed using QC report from the minfi package. All samples passed quality control and were subsequently processed using the ChAMP v.2.9.10 (ref. [69]) and minfi v.1.24.0 (ref. [70]) R packages. Upon loading raw IDAT files into ChAMP, probes were filtered by detection $P > 0.01$, bead count < 3 in at least 5% of the samples, single nucleotide polymorphisms (minor allele frequency > 1% in European population)[71,72] and cross-reactivity as identified by Nordlund et al.[73] and Chen et al.[74]. After filtering for probes located on X and Y chromosomes, 700,482 probes remained. Probes were subjected to within-sample normalization (ssNoob), which corrects for two different probe designs (type I and type II probes). Slide effects, as identified using PCA, were corrected using empirical Bayes methods[75] implemented in the ComBat function of the SVA Bioconductor package v.3.26.0. Reference-free cell-type deconvolution was performed using RefFreeEWAS[76] v.2.2, as no validated cell-based reference models exist for deconvolution of DNA methylation in the cellular fraction. The optimal number of fractions for deconvolution was determined by calculating the deviance-boots (epsilon value) over 100 iterations for the range of 1 to 6 fractions, with minimum deviance with 3 fractions. The fractions were obtained by solving the model $Y = M × Ω − T$ (where $Y$ is the original beta methylation matrix, $M$ is the cell-type-specific beta methylation matrix, $Ω$ is the cell proportion matrix and $T$ is the number of cell types to deconvolute) using the non-negative matrix factorization method where the fractions represent the estimated proportion of cells in the mixture belonging to this cell type, which was used as a covariate in the linear model. The limma Bioconductor package v.3.34.9 (ref. [65]) was used for detection of differentially methylated CpGs with $M$ values ($Mi = \log_2(βi1 − βi)$) as input. Of note, in the analysis of the glial methylome, since several glial samples with different brain locations were used from the same donor, limma was conducted using

within-individual comparisons in a linear mixed model by estimating the average correlation within the individual before using the function duplicateCorrelation, which was then used in the lmFit step. This step was not necessary in the analysis of bulk NAWM due to the absence of repeated individuals. The disease was tested as an exposure, empirical Bayes was used to moderate the standard errors towards a common value and *P* values were corrected using a Benjamini–Hochberg *P* value correction. For both glial and bulk NAWM analyses, the following covariates were included in the model: sex, age and deconvoluted cell type proportions. Putative transcription factor binding at the CpG-related sequences in the *HBEGF* locus was performed by mapping the genomic location of the recognition sequence to the CpGs targeted by the EPIC probes with JASPAR CORE database 2022 (http://expdata.cmmt.ubc.ca/JASPAR/downloads/UCSC_tracks/2022/). Of note, a transcription factor was assigned to a CpG only if the CpG base falls directly in the binding site.

### Pathway and statistical analysis of bulk RNA-seq data

GSEAPreranked analyses were used to generate enrichment plots for bulk RNA-seq using MSigDB molecular signatures for canonical pathways. For pre-ranked gene set enrichment analysis (GSEA), GSEA software by the Broad Institute[77,78] was used and genes were ranked based on the logarithmic fold change (logFC) and adjusted *P* value. Protein class analysis was performed using PANTHER[79], using genes enriched in astrocytes at the peak of EAE (logFC > 1, adjusted *P* < 0.05).

### Analysis of scRNA-seq data

For the analysis of *Hbegf* gene regulatory networks, publicly available single-cell data from EAE mice, published by Wheeler et al.[20], were used. In brief, Seurat[80] was used to perform log-normalization and filtering of scRNA-seq profiles. Following dimensionality reduction, the Louvain algorithm was used to perform unsupervised clustering of cell types. Visualization of UMAP plots was performed using ggplot2. To identify regulatory networks and *Hbegf* target genes, the clustered results were subjected to further analysis using NicheNet[28]. In brief, astrocytes were defined as sender cells, monocytes, microglia/macrophages, oligodendrocytes and neurons, and T cells were defined as receiver cells. The following preconstructed databases were downloaded: ligand–target matrix (https://zenodo.org/record/3260758/files/ligand_target_matrix.rds), ligand–receptor database (https://zenodo.org/record/3260758/files/lr_network.rds), weighted networks (https://zenodo.org/record/3260758/files/weighted_networks.rds). Following ranking of the ligands based on predicted ligand activity, the 20 top-ranked ligands were visualized using a dotplot. Active ligand–target genes of the top-ranked ligands were visualized using a heatmap. Next, cell-type-specific ligands were defined by an expression higher than the average + s.d. To define ligand–target links of interest, interactions belonging to the 40% lowest scores were removed. For the visualization of ligand–target interactions, a Circos plot was generated based on the cell-type-specific ligands and their respective target genes. Target genes of astrocyte-expressed *Hbegf* were extracted from the ligand–target matrix and subjected to further analysis using Enrichr[81]. To identify biological processes for which *Hbegf* target genes were enriched, gene ontology enrichment GO Biological Process 2021 (GO:0008150) was used. Potential target cell types were identified by gene set enrichment of *Hbegf* target genes using Descartes Cell Types and Tissue 2021 (ref. 82).

### Prediction of transcription factor binding sites

Putative transcription factor binding sites were identified using JASPAR[51] (https://jaspar.genereg.net/) on the 1,000-bp upstream region of the mouse (c36649866-36648847 *Mus musculus* strain C57BL/6J chromosome 18, GRCm39) and human (c140347589-140346602 *Homo sapiens* chromosome 5, GRCh38.p13 Primary Assembly) *Hbegf* locus with an 85% relative profile score threshold.

Conserved transcription binding sites were identified using MULAN (mulan.dcode.org). The putative transcription binding sites of the HB-EGF promoter–reporter construct (GeneCopoeia, no. HPRM20671) were identified using JASPAR based on the promoter sequence provided by the manufacturer. The respective binding sites can be found in Supplementary Table 3.

### Standard protocol approvals, registrations and patient consent

Experiments on human tissue were performed in accordance with the Declaration of Helsinki. An overview of the sample characteristics is provided in Supplementary Tables 1, 2, 4 and 6. The analyses of CSF, whole blood and serum samples for proteomic profiling and epigenetic analysis were approved by the standing ethical committee (14/18S) at Technical University Munich and the ethical committee at the University Hospital Erlangen. Immunohistochemical analyses of human tissues were conducted under the oversight of the local Research Ethics Committee of the University Freiburg Medical Center under the protocol number 10008/09. Written, informed consent was obtained from every patient. The animal study was reviewed and approved by Bavarian State Authorities (Regierung von Unterfranken, AZ 55.2.2-2532-2-1306, AZ 55.2.2-2532-2-1722).

### Statistical analysis

For the statistical analysis of two groups, a two-sample unpaired *t*-test was performed. For analysis of multiple groups, one-way analysis of variance (ANOVA) with Tukey's or Dunnett's multiple comparisons test was applied. For multiple testing, a two-way ANOVA with Dunnett's or Sidak's multiple comparisons test was applied. For regression analysis of EAE experiments, *P* values were calculated by testing whether the slopes were significantly different (two-tailed *t*-test). Family-wise significance and confidence level was set at *P* < 0.05. If not otherwise described, statistical examinations were carried out using GraphPad Prism 9 (v.9.5.1). Linear regression analysis of clinical scores for EAE experiments was performed using GraphPad Prism 9 (v.9.5.1) with the treatment start as starting point. Additional information on the study design, the number of replicates and the statistical tests used is provided in the figure legends. Graphs and illustrations were created using Adobe Illustrator (v.1.0.) or Inkscape (v.1.1).

### Reporting summary

Further information on research design is available in the Nature Portfolio Reporting Summary linked to this article.

## Data availability

All supplementary tables with processed data have been submitted with this manuscript. Bulk RNA-seq of ACSA2+ astrocytes with EAE and WGBS data have been deposited in the Gene Expression Omnibus (GEO) under the SuperSeries accession number GSE245383. DNA methylation of the glial nuclei and the bulk NAWM was assessed using Illumina Infinium Human Methylation EPIC/850K and 450K BeadChips, respectively, available under the GEO accession numbers GSE166207 and GSE40360, respectively. For the analysis of putative transcription factor binding sites, the JASPAR CORE database was used. Source data are provided with this paper.

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

## Acknowledgements

M.L. and V.R. were funded by an ERC Starting Grant by the European Research Council (ERC) (grant no. HICI 851693). V.R. was supported by a Heisenberg fellowship and Sachmittel support provided by the German Research Foundation (Deutsche Forschungsgemeinschaft, DFG, Project ID 401772351, grant nos. RO4866-1/1, 2/-1, 3/1, 4/1, 5/1, 6/1) as well as in transregional and collaborative research centers provided by the German Research Foundation (DFG, Project ID 408885537-TRR274, Project ID 261193037-CRC1181, Project ID 270949263-GRK2162, Project ID 405969122-FOR2886, Project ID 505539112-GB.com). T.T. was funded by the Kommission für Klinische Forschung (KKF), Klinikum rechts der Isar. L.L. and T. Beyer were funded by transregional research centers provided by the German Research Foundation (DFG, Project ID 408885537-TRR274). A.P. received funding by the German Research Foundation (DFG, Project ID 270949263-GRK2162, Project ID 405969122-FOR2886). O.V. received funding by the German Research Foundation (DFG, Project ID 270949263-GRK2162). M.A.W. received funding from the National Institutes of Health (grant nos. R01MH130458, R00NS114111). C.G. was funded by the German Research Foundation (Project ID 505539112-GB.com). F.J.Q. received funding from the National Institutes of Health, the NMSS and the Progressive MS Alliance (grant nos. NS087867, ES025530, ES032323, AI126880 and AI149699). M.J., M.N. and Y.H. received funding from the ERC under the European Union's Horizon 2020 Research and Innovation Program (grant agreement no. 818170), the Swedish Research Council, the Swedish Brain Foundation and the Knut and Alice Wallenberg Foundation. L.K. received financial support from the Swedish Research Council (grant no. 2021-02977) and the Margaretha af Ugglas Foundation. B.H. and M.J. received funding for the study by the European Union's Horizon 2020 Research and Innovation Program (grant no. MultipleMS, EU RIA 733161). B.H. also received funding from the DFG (German Research Foundation) under Germany's Excellence Strategy within the framework of the Munich Cluster for Systems Neurology (grant no. EXC 2145 SyNergy—ID 390857198) and the Clinspect-M consortium funded by the BMBF. B.H. is associated with DIFUTURE (Data Integration for Future Medicine, BMBF 01ZZ1804[A-I]). The Biobank of the Department of Neurology as part of the Joint Biobank Munich in the framework of the German Biobank Node supported the study. Computational analyses were enabled by resources provided by the Swedish National Infrastructure for Computing (SNIC) at Uppsala Multidisciplinary Center for Advanced Computational Science (UPPMAX) partially funded by the Swedish Research Council through grant agreement no. 2018-05973. We furthermore thank E. Ewing for his assistance in the computational analyses of glial and bulk epigenome datasets, and E. Barleon for excellent technical assistance in the immunohistochemical analyses of human CNS specimens.

## Author contributions

M.L. performed most in vitro and in vivo experiments. L.L., O.V., A.P., T.T., J.Z. and U.J.N. assisted with in vitro and in vivo experiments. L.L. and F.Z. performed immunohistochemical analyses of mouse tissue. Y.H., M.N. and L.K. performed bioinformatical analysis of epigenetic sequencing and array data. M.A. and L.M. performed primary mouse

oligodendrocyte cultures. E.W. and M.D. performed immunostaining on human CNS specimens. L. Nagel, F.B.L. and J.H. performed MRI imaging of mouse spinal cords. T. Beyer and L. Nirschl performed an EAE experiment and sorted the samples by FACS. M.L. and V.R. designed the study and edited the manuscript. A.C., M.A.W., M.J., J.W., T. Bäuerle, C.G., B.H., M.P. and F.J.Q. provided unique reagents and/or discussed and interpreted findings. M.L. and V.R. wrote the manuscript with input from all authors.

## Competing interests

The authors declare no competing interests.

## Additional information

**Extended data** is available for this paper at https://doi.org/10.1038/s41590-024-01756-6.

**Correspondence and requests for materials** should be addressed to Veit Rothhammer.

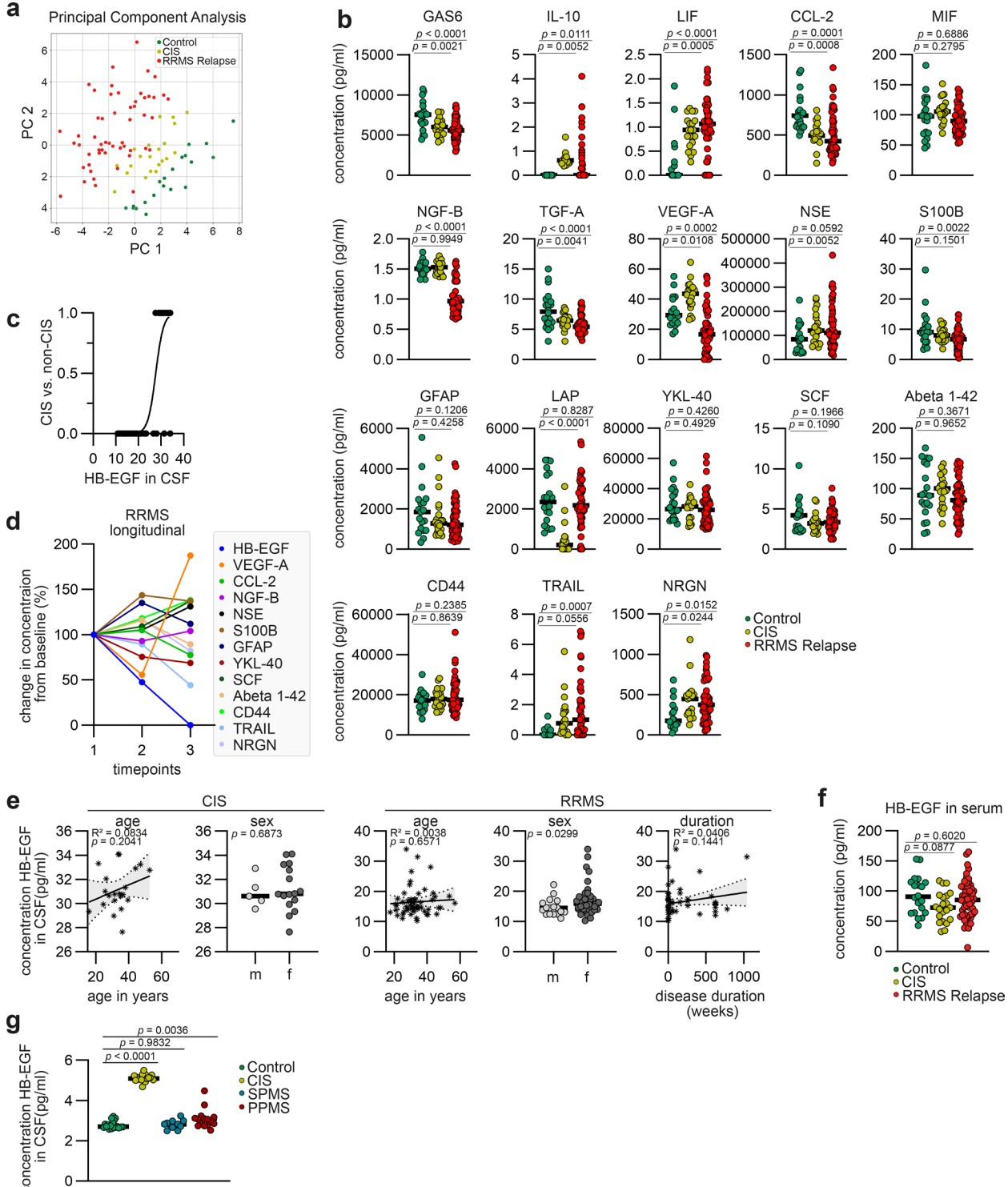

**Extended Data Fig. 1 | Proteomic profiling of CSF from MS and control patients. a**, Principal Component Analysis (PCA) of the CSF abundance of the measured analytes together with clinical parameters (Supplementary Table 1; Age, EDSS, Therapy, number of cerebral lesions, number of spinal lesions, optic neuritis, relapse frequency, number of relapses, disease duration (in weeks), cell count CSF) in controls (n = 20), CIS (n = 21), and RRMS (n = 54) patients. **b**, CSF levels of GAS6, IL-10, LIF, CCL-2, MIF, NGF-β, TGF-α, VEGF-A, NSE, S100B, GFAP, LAP, YKL-40, SCF, Aβ$_{1-42}$, CD44, TRAIL, and NRGN in (n = 20), CIS (n = 21), and RRMS (n = 54) patients. **c**, logistic regression of CSF HB-EGF levels in CIS (n = 21) vs. non-CIS (n = 74).**d**, change in concentration of HB-EGF, VEGF-A,

NGF-β, CCL-2, NSE, S100B, GFAP, YKL-40, SCF, Aβ$_{1-42}$, CD44, TRAIL, NRGN from baseline (first timepoint) in a RRMS patient (mean time between timepoints is 85 days). **e**, correlation between HB-EGF levels in the CSF with age, sex, and disease duration in CIS (left) and RRMS (right) patients. CIS n = 21, RRMS n = 54. **f**, serum concentration of HB-EGF in controls (n = 43), CIS (n = 21), and RRMS (n = 54) patients. **g**, CSF concentration of HB-EGF in control (n = 20), CIS (n = 21), SPMS (n = 12), and PPMS patients (n = 15) measured by single-plex ELISA. Patient characteristics are provided in Supplementary Table 2. Data shown as mean ± SD. Unpaired t-test in (**e**), One-way ANOVA with Dunnett's multiple comparisons test (tested against controls) in (**b**, **f**, **g**).

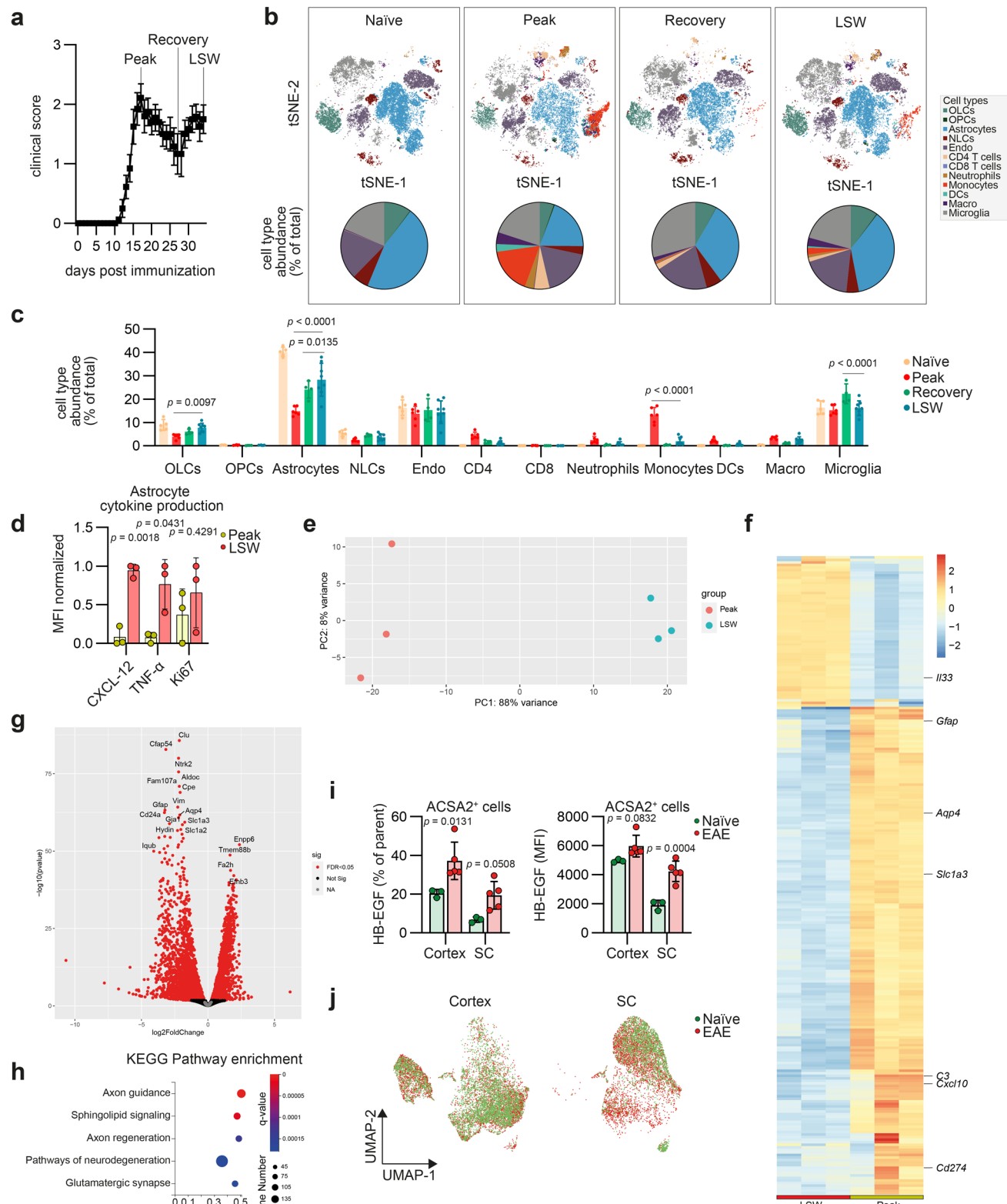

**Extended Data Fig. 2 | Regulation of HB-EGF by astrocytes during autoimmune neuroinflammation. a**, EAE development in wild-type mice with the timepoints analyzed (naïve, peak, recovery, LSW) indicated. n = 20. **b**, tSNE plots of CNS cells (upper, downsampled to 30000 cells) and cell type abundance (% of singlets; **c**) analyzed by high-dimensional flow cytometry. OLCs, oligodendrocyte lineage cells; OPCs, oligodendrocyte precursor cells; NLCs, neuronal lineage cells; Endo, endothelial cells; DCs, dendritic cells; Macro, macrophages. Naïve n = 5, Peak n = 6, Recovery n = 4, LSW n = 8. **d**, median fluorescence intensity (MFI) of CXCL-12, TNF-α, and Ki67 in astrocytes during peak and late-stage worsening (LSW) quantified by intracellular flow cytometry.

n = 3 per group. **e**, principal component analysis (PCA), heatmap (**f**), and volcano plot (**g**) of differential gene expression in ACSA2⁺ astrocytes during peak and LSW analyzed by RNA-Seq. n = 3 per group. **h**, KEGG Pathway enrichment of LSW astrocytes. **i**, relative expression (% of parent, left) and MFI (right) of HB-EGF production by cortical and spinal cord astrocytes in naïve (n = 3) and EAE mice (n = 5). **j**, UMAP plot of ACSA2⁺ cells (downsampled to 5000 cells) in naïve and EAE mice analyzed by high dimensional flow cytometry. Data shown as mean ± SD if not indicated otherwise. Data shown as mean ± SEM in (**a**). Two-way ANOVA with Tukey's multiple comparisons test in (**c**), unpaired t-test with Holm-Sidak correction in (**d**), Two-way ANOVA with Sidak's multiple comparisons test in (**h**).

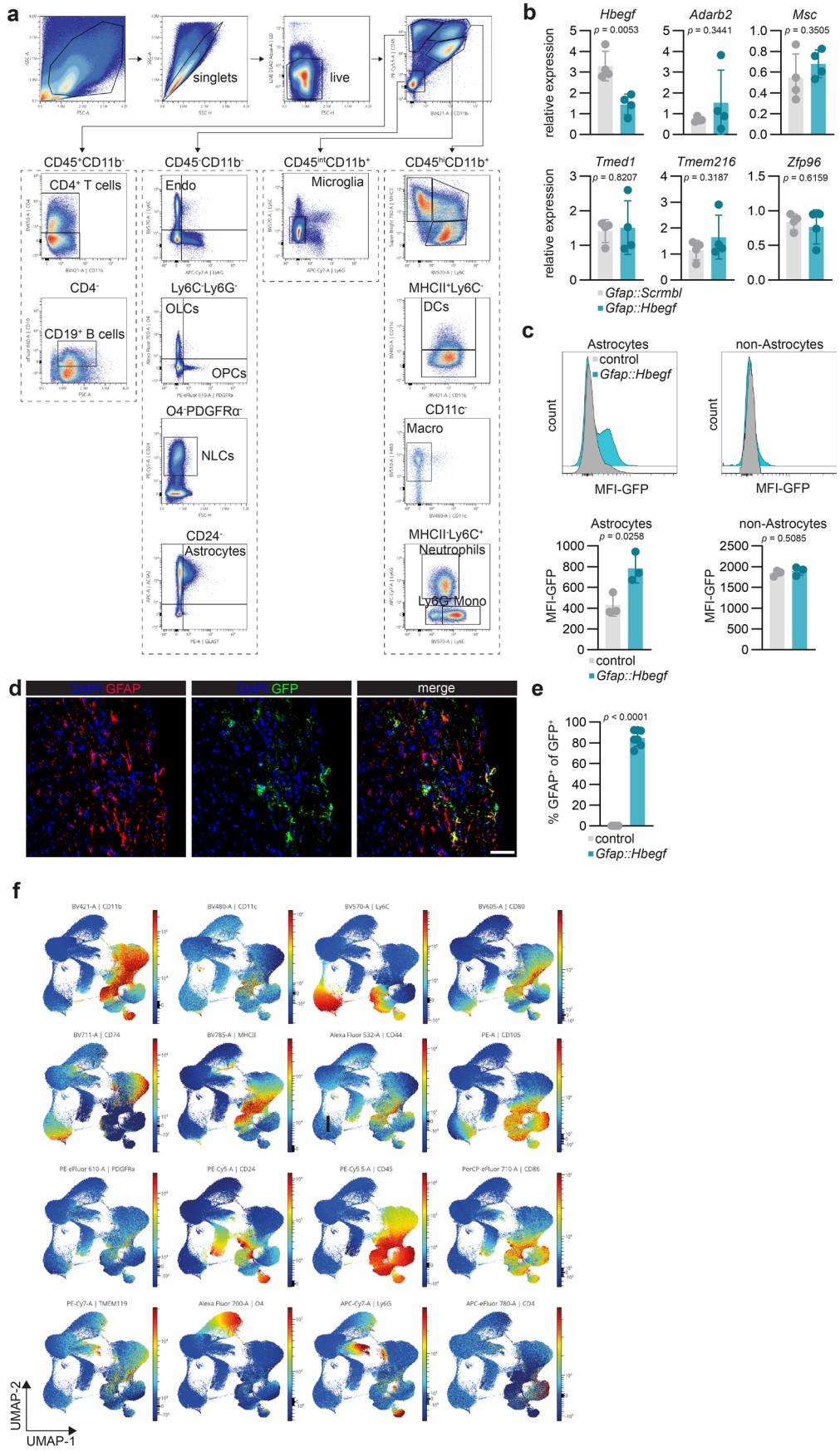

**Extended Data Fig. 3 | See next page for caption.**

**Extended Data Fig. 3 | Loss of astrocyte-derived HB-EGF worsens neuroinflammation. a**, gating strategy used for the identification of CNS-resident and -infiltrating cell populations by high dimensional flow cytometry. **b**, RT-qPCR analysis of potential off target genes in ACSA2+ astrocytes isolated from mice injected with a *Hbegf*-targeting (*Gfap::Hbegf*) or control (*Gfap::Scrmbl*) CRISPR/Cas9 construct. n = 4 per group. **c**, representative histograms and flow cytometric quantification of GFP-reporter signal in astrocytes vs. non-astrocytes obtained from mice injected with *Hbegf*-targeting (*Gfap::Hbegf*) CRISPR/Cas9 lentiviral particles or PBS (control). n = 3 per group. **d**, representative immunostaining and quantification (**e**) of GFP+ GFAP+ cells in spinal cords from injected with *Hbegf*-targeting (*Gfap::Hbegf*, n = 7) CRISPR/Cas9 lentiviral particles or PBS (control, n = 6). **f**, UMAP plots of CNS cells in *Gfap::Scrmbl* and *Gfap::Hbegf* mice analyzed by high-dimensional flow cytometry; colors indicate median fluorescence intensity (MFI). Data shown as mean ± SD. Unpaired t-test in (**b**, **c**, **e**).

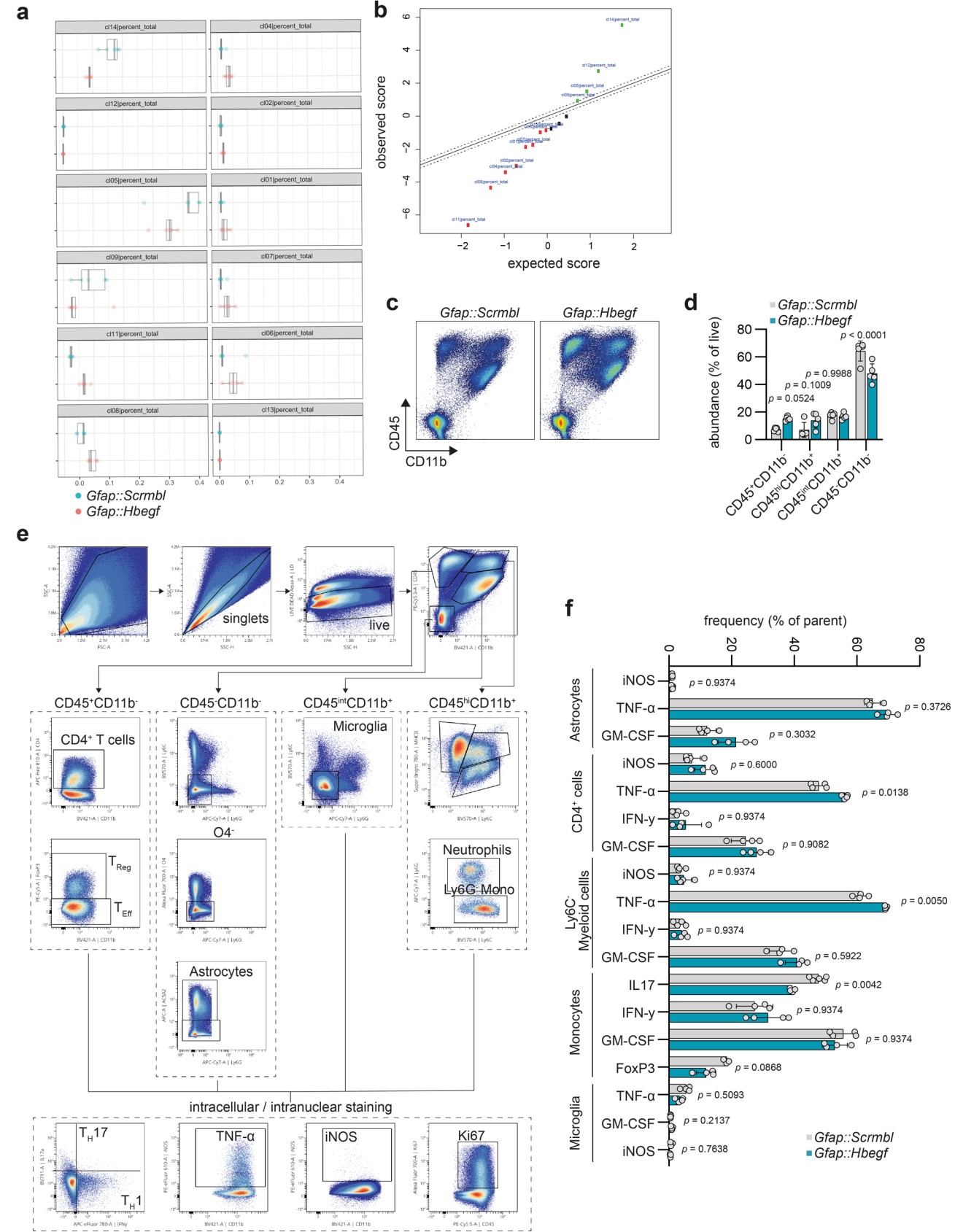

**Extended Data Fig. 4 | See next page for caption.**

**Extended Data Fig. 4 | The effect of astrocyte-specific HB-EGF abrogation during EAE. a**, Significance Analysis of Microarrays (SAM) of FlowSOM clusters from CNS cells obtained from *Gfap::Scrmbl* and *Gfap::Hbegf* mice analyzed by high-dimensional flow cytometry. n = 5 per group. **b**, SAM plot depicting significant alterations in cluster abundance in *Gfap::Scrmbl* and *Gfap::Hbegf* mice analyzed by high-dimensional flow cytometry. n = 5 per group. **c**, representative scatter plots and quantification (**d**) of CD45⁺CD11b⁻ infiltrating lymphocytes,

CD45$^{hi}$CD11b$^+$ myeloid cells, CD45$^{int}$CD11b$^+$ microglia, and CD45$^-$CD11b$^-$ cells in the CNS of *Gfap::Scrmbl* and *Gfap::Hbegf* mice. n = 5 per group. **e**, gating strategy used for the quantification of cytokines and intranuclear factors by intracellular flow cytometry. **f**, cytokine expression (% of parent) by major cell populations in the CNS of *Gfap::Scrmbl* and *Gfap::Hbegf* mice. n = 4 per group. Data shown as mean ± SD. Two-way ANOVA with Sidak's multiple comparisons test in (**d**, **f**).

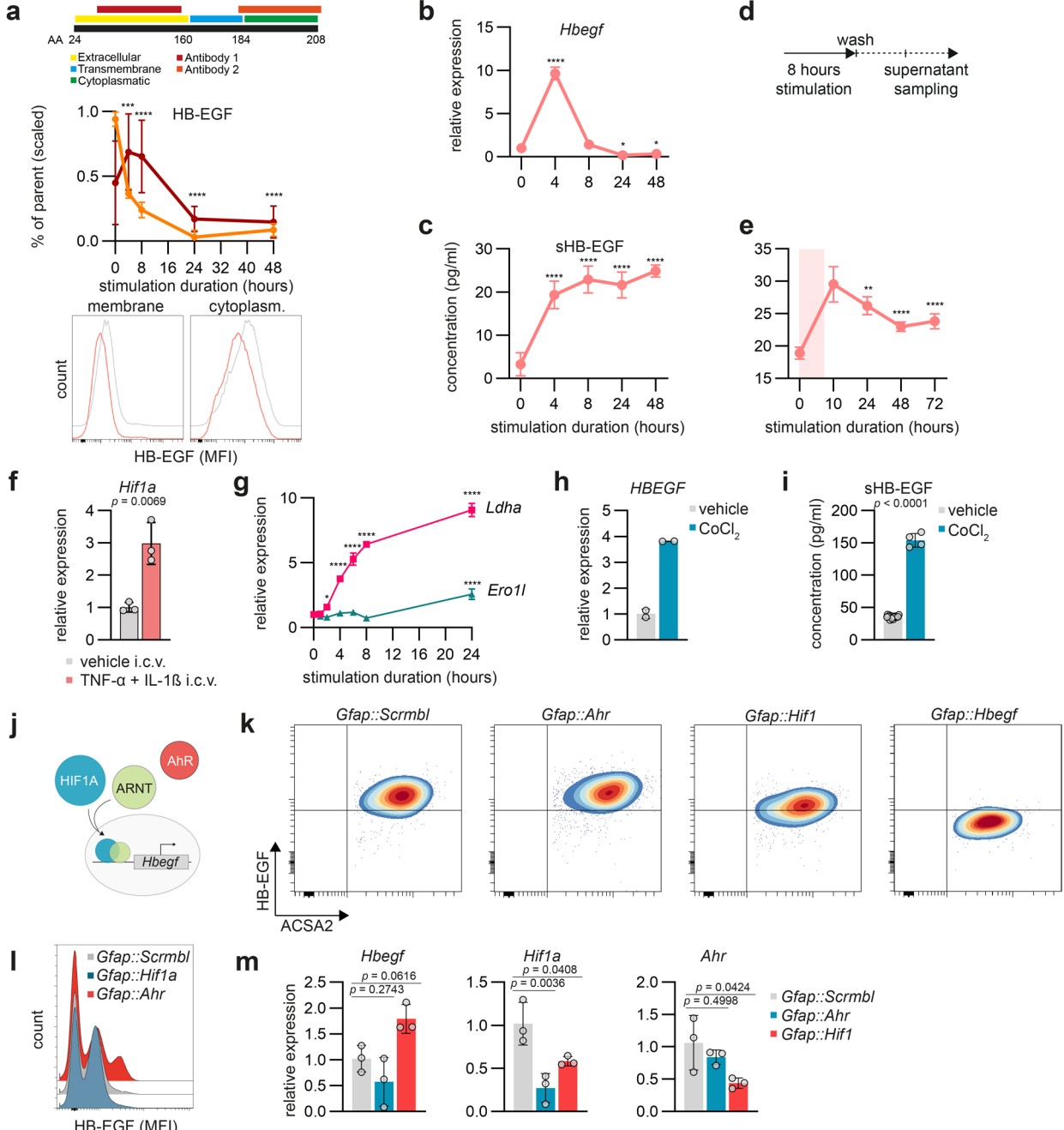

**Extended Data Fig. 5 | Inflammatory and hypoxic regulators of HB-EGF in astrocytes. a**, schematic of binding domains, flow cytometric quantification, and representative histograms of membranous HB-EGF (antibody 1) and cytoplasmatic HB-EGF (antibody 2) in astrocytes in response to stimulation with TNF-α and IL-1β over 48 hours. n = 4 per timepoint. **b**, RT-qPCR analysis of *Hbegf* expression and quantification of soluble HB-EGF (sHB-EGF) in the supernatant of primary mouse astrocyte in response to stimulation with TNF-α and IL-1β over 48 hours. n = 4 per timepoint. **d**, schematic of stimulation and supernatant sampling, as well as quantification of sHB-EGF (**e**) in primary mouse astrocytes stimulated with TNF-α and IL-1β for 8 hours, followed by extensive washing before supernatant sampling. n = 4/6 per timepoint. **f**, RT-qPCR analysis of *Hif1a* expression by ACSA2⁺ astrocytes following i.c.v. injection of TNF-α and IL-1β or vehicle. n = 3 per group. **g**, RT-qPCR analysis of *Ldha* and *Ero1l* expression as positive controls for HIF1α related signaling in primary mouse astrocytes stimulated with CoCl₂ over 24 hours. n = 4 per timepoint. **h**, RT-qPCR analysis

of HBEGF expression by human astrocytes under pseudohypoxic conditions (CoCl2). n = 2 per group. **i**, and Enzyme-linked Immunosorbent Assay (ELISA) of soluble HB-EGF (sHB-EGF) in the supernatant of primary mouse astrocytes under pseudohypoxic conditions (CoCl2). n = 4/16 per group. **j**, schematic of transcriptional competition between HIF1α and AhR. **k**, representative scatterplots of HB-EGF expression in primary mouse astrocytes (ACSA2 + GFP+) transduced with a control (Gfap::Scrmbl), AhR (Gfap::Ahr), HIF1α (Gfap::Hif1), or HB-EGF (Gfap::Hbegf) targeting CRISPR/Cas9 vector, quantified by intracellular flow cytometry. n = 3 per group. **l**, representative histograms depicting HB-EGF staining in astrocytes obtained from *Gfap::Scrmbl*, *Gfap::Ahr* and *Gfap::Hif1a* mice during late-stage EAE. **m**, RT-qPCR analysis of Hbegf, Ahr, Hif1a, and Ldha in ACSA2+ astrocytes in *Gfap::Scrmbl*, *Gfap::Hif1a*, and *Gfap::Ahr* mice. n = 3 per group. Data shown as mean ± SD. One-way ANOVA with Dunett's multiple comparisons test (tested against control) in (**a**, **b**, **c**, **e**, **m**), unpaired *t*-test in (**f**, **i**), Two-way ANOVA with Sidak's multiple comparisons test in (**g**).

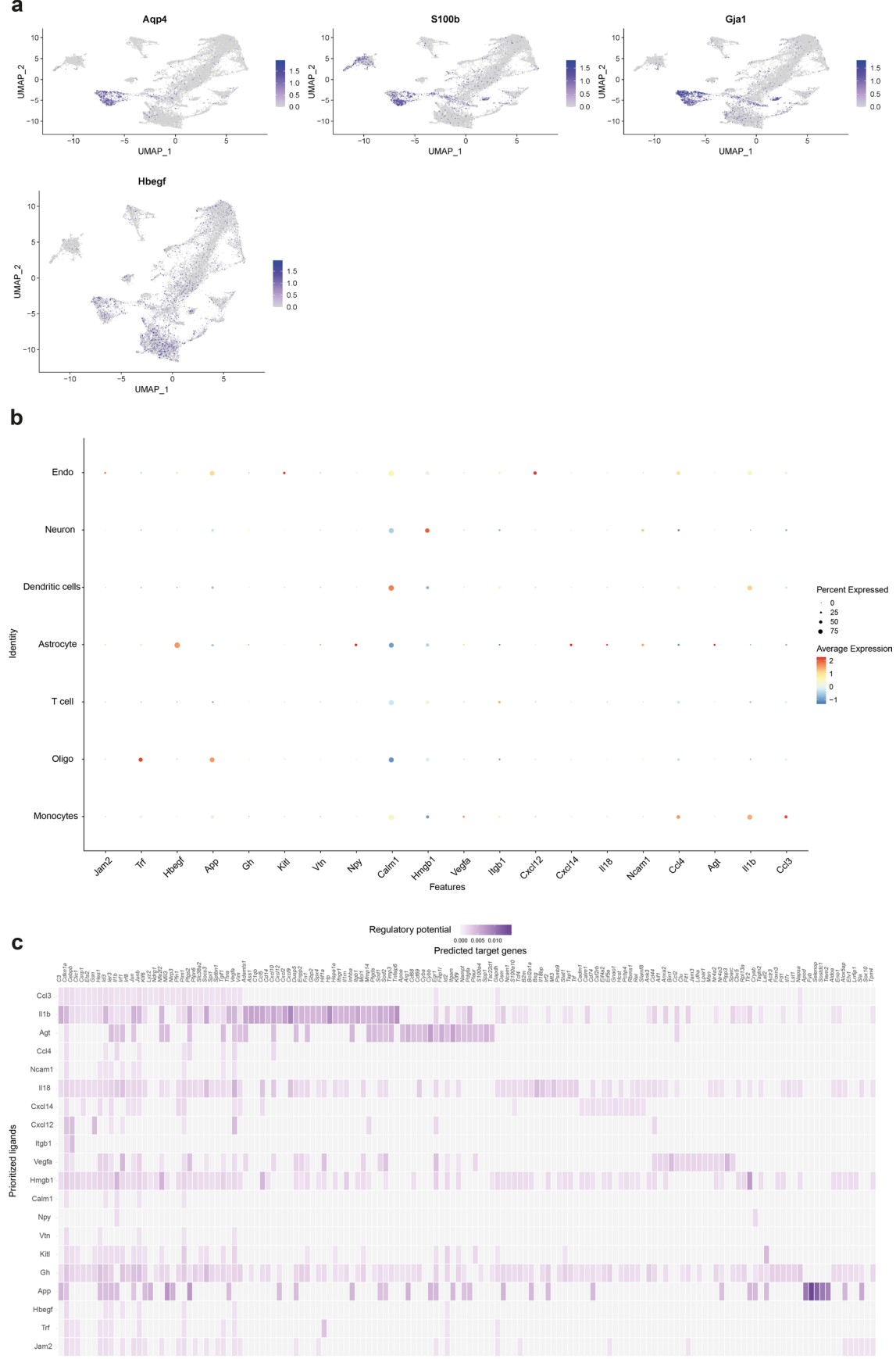

**Extended Data Fig. 6 | NicheNet analysis of Hbegf target genes. a**, UMAP plots of CNS cells analyzed by single-cell RNA Seq by Wheeler et al.[20]; color coded expression of the astrocyte markers *Apq4*, *S100b*, *Gja1* and *Hbegf*. **b**, NicheNet[28] 20 top-ranked ligands and their average expression by cell types in single-cell RNA Seq data of CNS cells by Wheeler et al.[20]. **c**, active ligand-target genes identified by NicheNet[28] in single-cell RNA Seq data of CNS cells by Wheeler et al.[20].

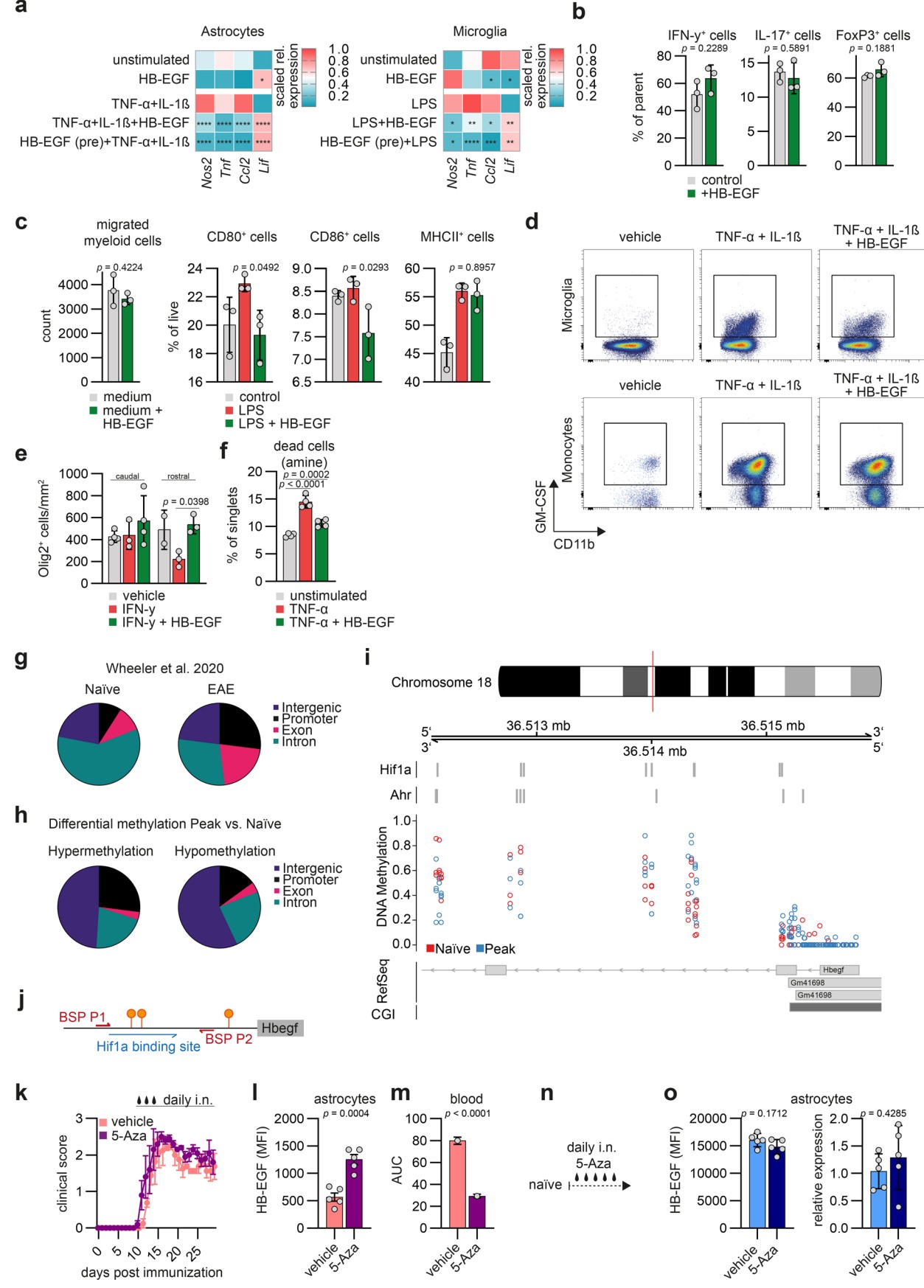

**Extended Data Fig. 7 | See next page for caption.**

**Extended Data Fig. 7 | Anti-inflammatory and tissue-protective effects of HB-EGF on CNS-resident and non-resident cell types. a**, Min. max. normalized expression of inflammatory genes in astrocytes (left) and microglia (right) ± stimulation with TNF-α, IL-1β or LPS ± HB-EGF, or after pre-stimulation with HB-EGF (pre). n = 2/4 per group. **b**, percent of parent of IFN-γ, IL-17, and FoxP3 by CD4⁺ T cells differentiated ± HB-EGF quantified by flow cytometry. n = 3 per group. **c**, counts of migrated CD11b⁺ myeloid cells (left), and expression of co-regulatory molecules CD80, CD86 and MHCII by monocytes (right). n = 3 per group. **d**, representative scatter plots of GM-CSF production by CD45$^{int}$CD11b⁺ microglia and CD45$^{hi}$CD11b⁺Ly6C⁺ monocytes in mice i.c.v. injected with vehicle or TNF-α, IL-1β ± HB-EGF. n = 3 per group. **e**, Olig2⁺ immunostaining in ON stimulated with vehicle or IFN-γ ± HB-EGF. n = 2-4 per group. **f**, live/dead staining of neuronal cells following TNF-α ± HB-EGF stimulation. n = 4 per group. **g**, DNA methylation in EAE astrocytes. Data obtained from Wheeler et al.[20]. **h**, DNA hyper- (left) and hypomethylation (right) in EAE astrocytes (naïve and peak). **i**, quantification of DNA methylation at HIF1α and AhR-binding sites in the *Hbegf* locus in astrocytes during peak EAE (n = 4) and naïve (n = 3) stages. **j**, schematic of HRM-PCR amplification. **k**, EAE in mice intranasally treated with vehicle (n = 4) / 5-Aza (n = 5) starting at symptom onset. **l**, MFI of HB-EGF in EAE astrocytes following vehicle or 5-Aza treatment. n = 5 per group. **m**, AUC of methylation at HIF1α binding sites in the *Hbegf* promoter after vehicle / 5-Aza treatment. n = 4 per group. **n**, schematic of treatment with 5-Aza in naïve mice. **o**, MFI of HB-EGF expression (left) and relative *Hbegf* expression (right) in astrocytes from naïve mice intranasally treated with 5-Aza / vehicle for 14 days. n = 5 per group. Data shown as mean ± SD / as mean ± SEM in (**k**). Two-way ANOVA with Dunnett's multiple comparisons test in (**a**) / with Sidak's multiple comparisons test in (**e**), unpaired t-test in (**b**, **c**, **l**, **m**, **o**), One-way ANOVA with Dunnett's multiple comparisons test in (**f**).

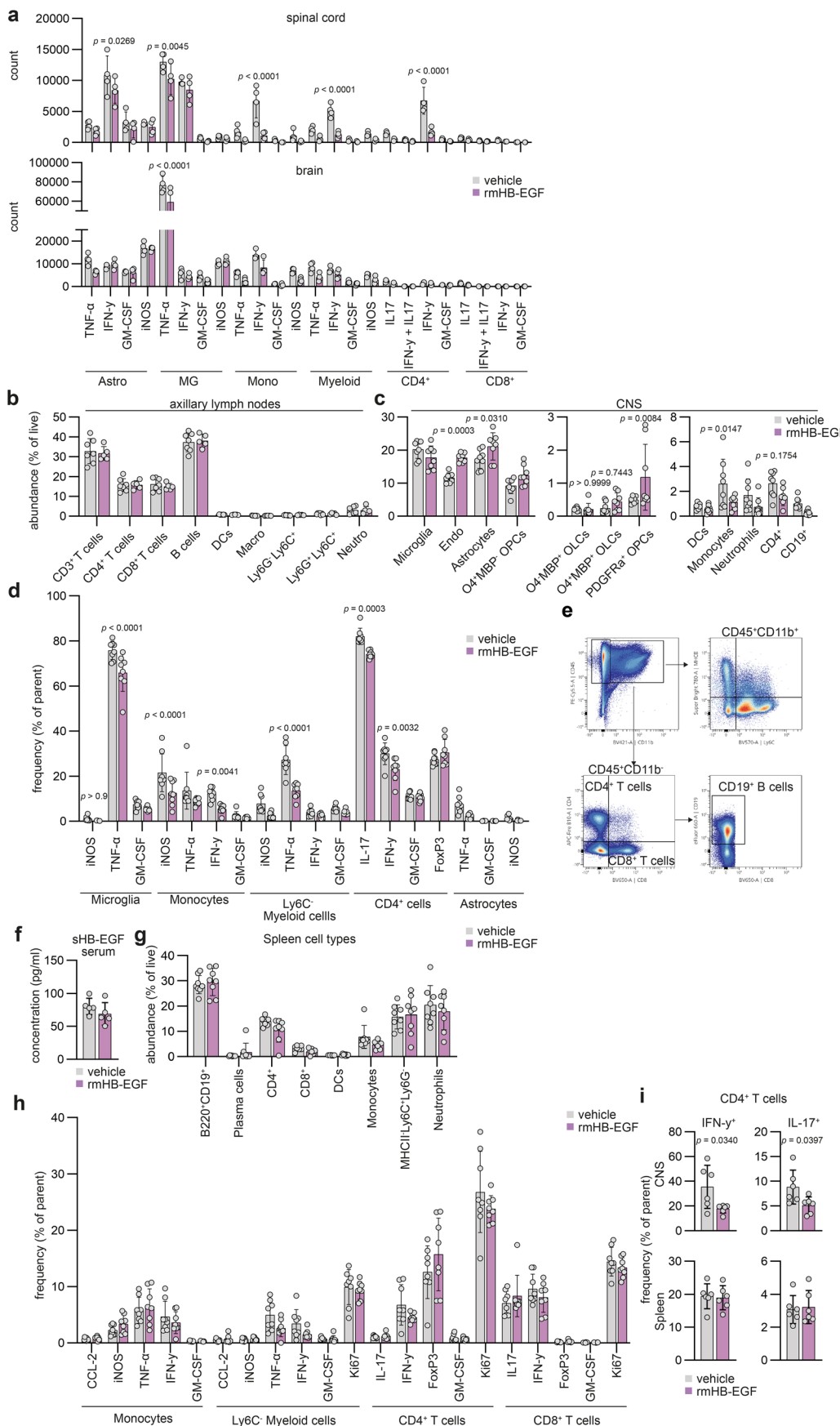

**Extended Data Fig. 8 | See next page for caption.**

**Extended Data Fig. 8 | Intranasal treatment with recombinant HB-EGF ameliorates CNS inflammation. a**, absolute counts of cytokines produced by microglia, monocytes, Ly6C⁻ myeloid cells, CD4⁺ T cells, and astrocytes in the spinal cord or brain of vehicle or rmHB-EGF treated mice (starting symptom onset). n = 4 per group. **b**, abundance (% of live) of major immune cell populations in draining axillary lymph nodes of mice intranasally treated with vehicle (n = 7) or recombinant HB-EGF (rmHB-EGF, n = 5) starting at symptom onset. **c**, abundance (% of live) of CNS-resident and CNS-infiltrating cell types in EAE mice intranasally treated with vehicle or recombinant mouse HB-EGF (rmHB-EGF) starting at symptom onset. DC, dendritic cells, Endo, endothelial cells; OPCs, oligodendrocyte progenitor cells; OLCs, oligodendrocyte lineage cells. n = 8 per group. **d**, relative expression (% of parent) of cytokines produced by microglia, monocytes, Ly6C⁻ myeloid cells, CD4⁺ T cells, and astrocytes in the CNS of vehicle or rmHB-EGF treated mice. n = 8 per group. **e**, gating strategy used for the analysis of splenic cell types by high dimensional flow cytometry. **f**, concentration of soluble HB-EGF (sHB-EGF) in sera of vehicle- or rmHB-EGF treated mice. n = 5 per group. **g**, abundance of cell populations (% of live) in spleens of mice treated with vehicle or rmHB-EGF. n = 8 per group. **h**, relative expression (% of parent) of cytokines produced by monocytes, Ly6C- myeloid cells, CD4⁺ and CD8⁺ T cells, in spleens of vehicle or rmHB-EGF treated mice. n = 8 per group. **i**, frequency of $T_H1$ (IFN⁻γ⁺) and $T_H17$ (IL-17⁺) CD4⁺ T cells in the CNS and spleen of mice that were intranasally treated with vehicle or rmHB-EGF starting at peak of disease. n = 6 per group. Data shown as mean ± SD. Two-way ANOVA with Sidak's multiple comparisons test in (**a-d**, **g**, **h**). Unpaired t-test in (**f**, **i**).

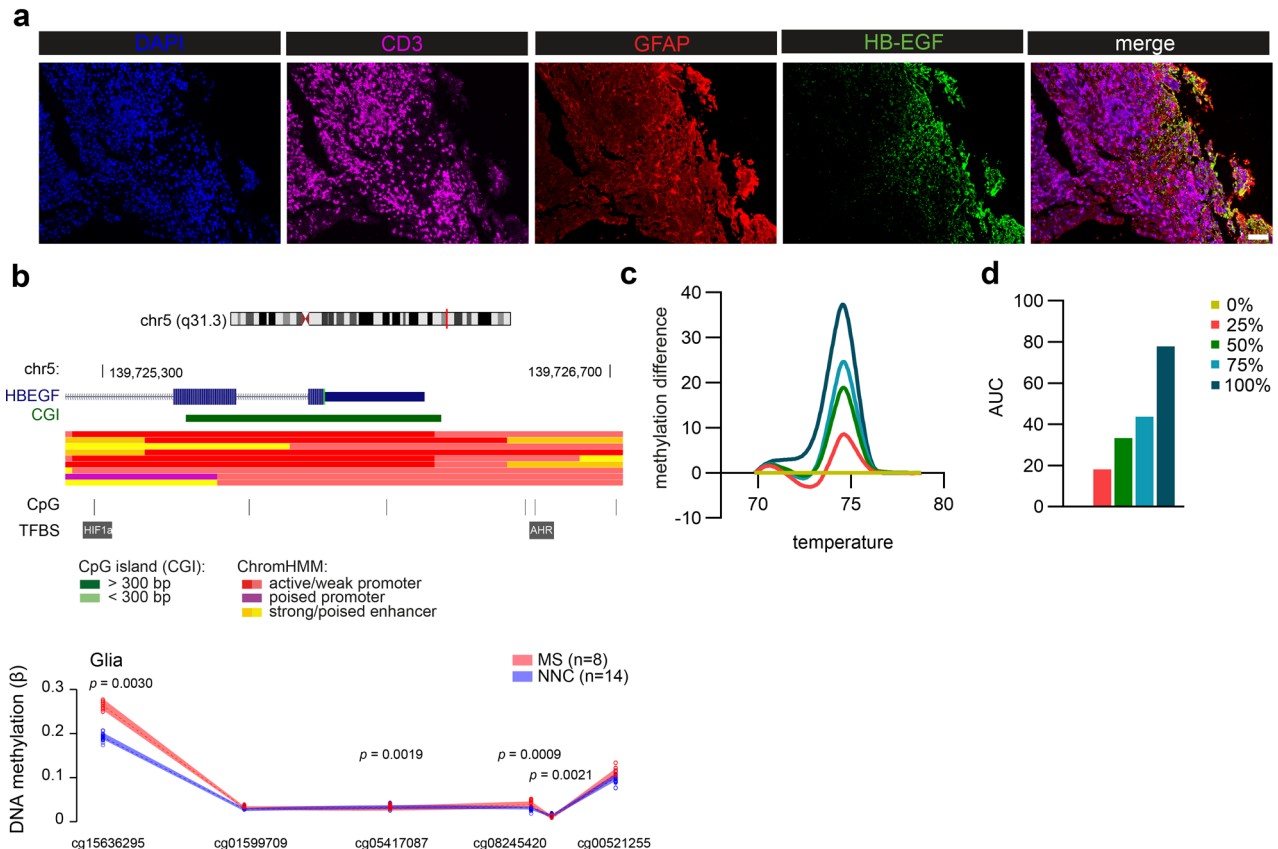

**Extended Data Fig. 9 | Analysis of HB-EGF in human pathology.**
**a**, immunostaining of a stereotactic biopsy from a patient with acute disseminated encephalomyelitis (ADEM; lower). DAPI staining (blue) of nuclei, CD3 staining (purple) of T cells, GFAP staining (red) of astrocytes, HB-EGF staining (green). Scale bar 100 μm. Patient characteristics are provided in Supplementary Table 4. **b**, ChromHMM features and genomic location of differentially methylated CpGs in the *HBEGF* locus in NeuN⁻ glial cells from MS patients or non-neurological controls (NNCs). n = 8 MS, n = 14 NNCs. **c**, difference plot and area under the curve (AUC; **d**) of 0%, 25%, 50%, 75%, and 100% methylated genomic DNA used as a standard control.

# Reporting Summary

## Statistics

For all statistical analyses, confirm that the following items are present in the figure legend, table legend, main text, or Methods section.

| n/a | Confirmed | |
|---|---|---|
| ☐ | ☒ | The exact sample size (*n*) for each experimental group/condition, given as a discrete number and unit of measurement |
| ☐ | ☒ | A statement on whether measurements were taken from distinct samples or whether the same sample was measured repeatedly |
| ☐ | ☒ | The statistical test(s) used AND whether they are one- or two-sided<br>*Only common tests should be described solely by name; describe more complex techniques in the Methods section.* |
| ☐ | ☒ | A description of all covariates tested |
| ☐ | ☒ | A description of any assumptions or corrections, such as tests of normality and adjustment for multiple comparisons |
| ☐ | ☒ | A full description of the statistical parameters including central tendency (e.g. means) or other basic estimates (e.g. regression coefficient) AND variation (e.g. standard deviation) or associated estimates of uncertainty (e.g. confidence intervals) |
| ☐ | ☒ | For null hypothesis testing, the test statistic (e.g. *F*, *t*, *r*) with confidence intervals, effect sizes, degrees of freedom and *P* value noted<br>*Give P values as exact values whenever suitable.* |
| ☒ | ☐ | For Bayesian analysis, information on the choice of priors and Markov chain Monte Carlo settings |
| ☒ | ☐ | For hierarchical and complex designs, identification of the appropriate level for tests and full reporting of outcomes |
| ☐ | ☒ | Estimates of effect sizes (e.g. Cohen's *d*, Pearson's *r*), indicating how they were calculated |

*Our web collection on statistics for biologists contains articles on many of the points above.*

## Software and code

Policy information about availability of computer code

| Data collection | Illumina NovaSeq, DNBSeq, ProcartaPlex Luminex Platform, Cytek Northern Lights; Cytek SpectroFlo (v3.0), Zeiss Zen Black (v.2011), ClinScan 70/30 (Bruker) |
|---|---|
| Data analysis | Visualization and statistics were performed using GraphPad Prism (v.9.5.1); Differential expression analysis of bulk RNA-Seq data was performed using R (v.4.3.2) and DESeq2 (1.38.0); Analysis and visualization of WGBS data was performed using awk and Bismark (v0.24.0), IGV (v.2.5.1.), DSS (v.2.30.1), annotatr (v.1.8.0), and genomation (v.1.4.2), ggplot2 (v3.1.0); GSEA (v 6.3); MSigDB (v 6.2); High Resolution Melt Software (Thermo Fisher Scientific, #A29881, v.3.2), FIJI (v1.53); JASPAR; OMIQ, Adobe Illustrator (v.1.0); Adobe Photoshop (v.24.0.1), Inkscape (v.1.1) |

For manuscripts utilizing custom algorithms or software that are central to the research but not yet described in published literature, software must be made available to editors and reviewers. We strongly encourage code deposition in a community repository (e.g. GitHub). See the Nature Portfolio guidelines for submitting code & software for further information.

# Data

Policy information about <u>availability of data</u>

All manuscripts must include a <u>data availability statement</u>. This statement should provide the following information, where applicable:

- Accession codes, unique identifiers, or web links for publicly available datasets
- A description of any restrictions on data availability
- For clinical datasets or third party data, please ensure that the statement adheres to our <u>policy</u>

Bulk RNA Sequencing of ACSA2+ astrocytes during EAE and WGBS data have been deposited into the Gene Expression Omnibus (GEO) under the SuperSeries accession number GSE225606. DNA methylation of the glial nuclei and the bulk NAWM was assessed using Illumina Infinium Human Methylation EPIC/850K and 450K BeadChips, respectively, available under the GEO accession numbers GSE166207 and GSE40360, respectively.

# Human research participants

Policy information about <u>studies involving human research participants and Sex and Gender in Research.</u>

| | |
|---|---|
| Reporting on sex and gender | Information on sex for the cohorts in Table 1. The sex-distrubution in these cohorts reflects the overall prevalence of Multiple Sclerosis. |
| Population characteristics | Information on sex, age, disease duration, disability (EDSS), treatment, and additional clinical parameters can be found in the respective Supplementary Tables |
| Recruitment | Patients were prespectively recruited in our Neuroimmunology Outpatient Departments for diagnostic procedures including CSF sampling. Patients were only included in this study if the diagnosis of Multiple Sclerosis was confirmed. CSF, whole blood and serum samples for multiplex and singleplex analyses was obtained from the Joint Biobank Munich in the framework of the German Biobank node and the Biobank at the University Hospital Erlangen. The brain tissue used for the analysis of HBEGF promoter methylation was obtained from the Multiple Sclerosis and Parkinson's Tissue Bank (Imperial College London). Tissue for immunohistochemical analyses was obtained from the Institute of Neuropathology at the University Freiburg Medical Center. |
| Ethics oversight | Experiments on human tissue were performed in accordance with the Declaration of Helsinki. The analyses of CSF, whole blood, and serum samples for proteomic profiling and epigenetic analysis was approved by the standing ethical committee (14/18S) at Technical University Munich and the ethical committee at the University Hospital Erlangen. Immunohistochemical analyses of human tissues were conducted under the oversight of local Research Ethics Committee of the University Freiburg Medical Center under the protocol number 10008/09. |

Note that full information on the approval of the study protocol must also be provided in the manuscript.

# Field-specific reporting

Please select the one below that is the best fit for your research. If you are not sure, read the appropriate sections before making your selection.

☒ Life sciences  ☐ Behavioural & social sciences  ☐ Ecological, evolutionary & environmental sciences

For a reference copy of the document with all sections, see nature.com/documents/nr-reporting-summary-flat.pdf

# Life sciences study design

All studies must disclose on these points even when the disclosure is negative.

| | |
|---|---|
| Sample size | N numbers range from n=3 to n=54, with n= individual mouse/patient, based on previously published work with the same stimulation paradigms and readout (PMID: 36266581; PMID: 33888612). For all cell based experiments, no prior calculations of sample sizes were performed. |
| Data exclusions | No data was excluded from the analysis |
| Replication | To ensure replication, all deep sequencing data were repeated in 3-4 mice per group per timepoint. For in vitro and in vivo experiments, experiments were repeated at least 3 times. All attempts at replication were successful. |
| Randomization | Samples and mice were randomly allocated into biological groups. |
| Blinding | Experimenters were blinded to biological group during EAE scoring. Immunohistochemical analyses were performed blinded. For all other experiments, no blinding was required as it would not affect of the quantitive results. |

# Reporting for specific materials, systems and methods

We require information from authors about some types of materials, experimental systems and methods used in many studies. Here, indicate whether each material, system or method listed is relevant to your study. If you are not sure if a list item applies to your research, read the appropriate section before selecting a response.

## Materials & experimental systems

| n/a | Involved in the study |
|-----|----------------------|
| ☐ | ☒ Antibodies |
| ☐ | ☒ Eukaryotic cell lines |
| ☒ | ☐ Palaeontology and archaeology |
| ☐ | ☒ Animals and other organisms |
| ☒ | ☐ Clinical data |
| ☒ | ☐ Dual use research of concern |

## Methods

| n/a | Involved in the study |
|-----|----------------------|
| ☒ | ☐ ChIP-seq |
| ☐ | ☒ Flow cytometry |
| ☒ | ☐ MRI-based neuroimaging |

## Antibodies

| | |
|---|---|
| Antibodies used | Flow Cytometry Antibodies:<br>BV421-CD11b (Biolegend, #101235; 1:200), BV480-CD11c (BD, #565627, 1:100), BV510-F4/80 (Biolegend,# 123135, 1:100), BV570-Ly6C (Biolegend,# 128029, 1:200), BV605-CD80 (BD, #563052, 1:100), BV650-CD56 (BD, #748098, 1:100), BV650-CD8 (BD, #100741, 1:100), PE-eFlour610-CD140a (Thermo Fisher Scientific, #61140180, 1:100), SuperBright780-MHCII (Thermo Fisher Scientific, #78532080, 1:200), BV711-CD74 (BD, #740748, 1:200), PE-CD45R/B220 (BD, #561878, 1:100), PE-CD105 (Thermo Fisher Scientific, # 12-1051-82, 1:100), PE-Ly6G (BioLegend, #127607, 1:200), PE-CD140a (BioLegend, #135905, 1:100), PE-O4 (Miltenyi, # 130117507, 1:100), PE-Ter119 (Biolegend, #116207), PE-Ly6C (Biolegend, #128007, 1:100), AF488-A2B5 (Novus Biologicals, #FAB1416G, 1:100), PE-Cy5-CD24 (Biolegend, #101811, 1:200), PE-Cy7-CD31 (Thermo Fisher Scientific, #25031182, 1:200), PerCP-eFlour710-CD86 (Thermo Fisher Scientific, #46086280, 1:100), AF532-CD44 (Thermo Fisher Scientific, #58044182, 1:100), PE-Cy5.5-CD45 (Thermo Fisher Scientific, #35045180, 1:300), JF646-MBP (Novus Biologicals, #NBP2-22121JF646, 1:100), APC-Cy7-HB-EGF (Bioss, #BS-3576R-APC-CY7, 1:100), APC-Cy7-Ly6G (Biolegend, #127623, 1:200), AF700-O4 (R&D, #FAB1326N, 1:200, 1:100), BUV737-CD154 (BD, #741735, 1:100), AF660-CD19 (Thermo Fisher Scientific, #606019380, 1:100), APC/Fire810-CD4 (Biolegend, #100479, 1:100), PE-eFlour610-iNOS (eBioscience, #61592080, 1:100), BV711-IL17a (Biolegend, #506941, 1:100), AF488-HB-EGF (SantaCruz, #sc-365182 AF488, 1:100), FITC-CXCL12 (Thermo Fisher Scientific, #MA523547, 1:100), PE-Cy5-FoxP3 (Thermo Fisher Scientific, #15-5773-82, 1:200), PE-Cy7-IFNγ (Biolegend, #505826, 1:100), PE PerCP-eFlour710-TNF (eBioscience, #46732180, 1:200), APC-GM-CSF (eBioscience, #17733182), APC-eF780-Ki67 (Thermo Fisher Scientific, #506941, 1:100).<br><br>Immunohistochemistry Antibodies:<br>mouse anti-HB-EGF (1:200; Santa Cruz; #sc365182), rat anti-GFAP (1:800; Thermo Fischer Scientific, #2.2B10), donkey anti-rat IgG AF488 (1:500; Thermo Fisher Scientific, #A21208), donkey anti-mouse IgG AF647 (1:500; Dianova, #715-605-151), mouse anti-SMI32 (1:1000; BioLegend, #801701), rabbit anti-Olig2 (1:200; Abcam, # ab109186), anti-mouse IgG AF488 (1:500; Thermo Fisher Scientific, #A21202), donkey anti-rabbit IgG AF647 (1:500; Dianova, #711605152), rabbit anti-RBPMS (1:300; Merck, ABN1362), goat anti-rabbit IgG Cy3 (Thermo Fischer, A10520), rat anti-human CD3 (1:100, eBioscience 14-0032-82), rabbit anti-human HB-EGF (1:50, LS-B12617-50), chicken anti-human GFAP (1:1000, ab4674). |
| Validation | All commercial antibodies in this study were validated, based on the manufacturers' websites. Antibodies were used for the appropriate animal host and application(s), as per the information provided on those websites:<br>1. BV421-CD11b (1:200; Biolegend, #101235):<br>https://www.biolegend.com/en-us/products/brilliant-violet-421-anti-mouse-human-cd11b-antibody-7163?GroupID=BLG10427<br>• Doni A, et al. 2015. J Exp Med. 212:905.<br>• Däbritz J, et al. 2016. Sci Rep. 6:20584.<br>• Chai Y, et al. 2016. PLoS One. 11: 0162853.<br>• Moderzynski K, et al. 2016. PLoS Negl Trop Dis. .<br>• Su Y, et al. 2022. J Hematol Oncol. 15:99.<br>• Hou X, et al. 2020. Cell Reports. 28(1):172-189.e7..<br>• Liu J, et al. 2019. Immunity. 50:600.<br>• Ilinykh PA, et al. 2020. Cell Host & Microbe. 27(6):976-991.<br>• Miller CM, et al. 2020. J Virol. 94:00:00.<br>• Li Q, et al. 2019. Neuron. 101:207.<br>• Klemm F, et al. 2020. Cell. 181(7):1643-1660.e17.<br>• Yan L, et al. 2021. Front Cell Neurosci. 15:750373.<br><br>2. BV480-CD11c (1:100; BD, #565627):<br>https://www.bdbiosciences.com/en-au/products/reagents/flow-cytometry-reagents/research-reagents/single-color-antibodies-ruo/bv480-mouse-anti-human-cd11c.566184<br>• Knapp W. W. Knapp .. et al., ed. Leucocyte typing IV : white cell differentiation antigens. Oxford New York: Oxford University Press; 1989:1-1182.<br>• Stacker SA, Springer TA. Leukocyte integrin P150,95 (CD11c/CD18) functions as an adhesion molecule binding to a counter-receptor on stimulated endothelium. J Immunol. 1991; 146(2):648-655. (Clone-specific: ELISA).<br>• Visser L, Shaw A, Slupsky J, Vos H, Poppema S. Monoclonal antibodies reactive with hairy cell leukemia. Blood. 1989; 74(1):320-325. (Immunogen: Immunocytochemistry (cytospins), Immunohistochemistry, Immunoprecipitation). |

3. BV510-F4/80 (1:100; Biolegend, #123135):
https://www.biolegend.com/en-us/products/brilliant-violet-510-anti-mouse-f4-80-antibody-8934
- Schaller E, et al. 2002. Mol. Cell. Biol. 22:8035. (IHC)
- Stevceva L, et al. 2001. BMC Clin Pathol. 1:3. (IHC)
- Kobayashi M, et al.2008. J. Leukoc. Biol. 83:1354.
- Poeckel D, et al. 2009. J Biol Chem. 284:21077.
- Glass AM, et al. 2013. J. Immunol. 190:4830.
- Koehm S, et al. 2007. J. Allergy Clin. Immunol. 120:570. (IHC)
- Rankin AL, et al. 2010. J. Immunol. 184:1526. (IHC)
- Sasi SP, et al. 2014. J Biol Chem. 289:14178.
- Thakus VS, et al. 2014. Toxicol Lett. 230:322.
- Watson NB, et al. 2015. J Immunol. 194:2796.
- Hirakawa H, et al. 2015. PLoS One. 10:119360.
- Radtke AJ, et al. 2020. Proc Natl Acad Sci U S A. 117:33455-65. (SB)

4. BV570-Ly6C (1:200; Biolegend, #128029):
https://www.biolegend.com/en-us/products/brilliant-violet-570-anti-mouse-ly-6c-antibody-7392
- Harsha Krovi S, et al. 2020. Nat Commun. 4.790277778.
- Sepe JJ, et al. 2022. JACC Basic Transl Sci. 7:915.
- Linnerbauer M, et al. 2022. Front Immunol. 12:800128.
- Li J, et al. 2020. Cancer Discov. .
- Wu X, et al. 2021. Elife. 10:.
- Li J, et al. 2020. Cancer Immunol Res. 0.529166667.
- Haase C, et al. 2022. Nat Methods. 19:1622.
- Stump CT, et al. 2021. Open Biol. 11:210245.
- Ajina R, et al. 2021. Cancer Immunol Res. 9:386.
- Hulsmans M et al. 2017. Cell. 169(3):510-522 .
- Li J, et al. 2018. Immunity. 49:178.
- , et al. 2021. Eur J Immunol. 51:2708.

5. BV605-CD80 (1:100; BD, #563052):
https://www.bdbiosciences.com/en-de/products/reagents/flow-cytometry-reagents/research-reagents/single-color-antibodies-ruo/bv605-hamster-anti-mouse-cd80.563052
- Bluestone JA. New perspectives of CD28-B7-mediated T cell costimulation. Immunity. 1995; 2(6):555-559. (Biology).
- Boussiotis VA, Gribben JG, Freeman GJ, Nadler LM. Blockade of the CD28 co-stimulatory pathway: a means to induce tolerance. Curr Opin Immunol. 1994; 6(5):797-807. (Biology).
- Hathcock KS, Laszlo G, Pucillo C, Linsley P, Hodes RJ. Comparative analysis of B7-1 and B7-2 costimulatory ligands: expression and function. J Exp Med. 1994; 180(2):631-640. (Biology).

6. BV650-CD56 (1:100; BD, #748098):
https://www.bdbiosciences.com/en-de/products/reagents/flow-cytometry-reagents/research-reagents/single-color-antibodies-ruo/bv650-rat-anti-mouse-cd56-ncam-1.748098
- Fujita T, Chen MJ, Li B, et al. Neuronal transgene expression in dominant-negative SNARE mice.. J Neurosci. 2014; 34(50):16594-604. (Clone-specific: Fluorescence activated cell sorting).
- Li S, Nie EH, Yin Y, et al. GDF10 is a signal for axonal sprouting and functional recovery after stroke.. Nat Neurosci. 2015; 18(12):1737-45. (Clone-specific: Fluorescence activated cell sorting).
- Rougon G, Deagostini-Bazin H, Hirn M, Goridis C. Tissue- and developmental stage-specific forms of a neural cell surface antigen linked to differences in glycosylation of a common polypeptide.. EMBO J. 1982; 1(10):1239-44. (Biology).

7. BV650-CD8 (1:100; Biolegend, #100741):
https://www.biolegend.com/en-us/products/brilliant-violet-650-anti-mouse-cd8a-antibody-7635
- Schädlich IS, et al. 2022. iScience. 25:104470.
- Flamar AL, et al. 2020. Immunity. 52(4):606-619.e6..
- Wiesner DL, et al. 2020. Cell Host Microbe. 614:27.
- Boyd DF, et al. 2020. Nature. 587:466.
- Kloepper J, et al. 2016. Proc Natl Acad Sci U S A. 113: 4476-4481.
- Schönberger K, et al. 2022. Cell Stem Cell. 29:131.
- Suresh R, et al. 2020. J Immunother Cancer. 8:.
- Arce Vargas F et al. 2018. Cancer cell. 33(4):649-663 .
- Piepke M, et al. 2021. J Neuroinflammation. 18:265.
- Sauter M, et al. 2022. iScience. 25:103677.
- Coleby R, et al. 2021. Clin Exp Rheumatol. :39.
- Abou-Hamad J, et al. 2022. iScience. 25:105524.

8. PE-eFlour610-CD140a (1:100; Thermo Fisher Scientific, #61140180):
https://www.thermofisher.com/antibody/product/CD140a-PDGFRA-Antibody-clone-APA5-Monoclonal/61-1401-80
9. SuperBright780-MHCII (1:200; Thermo Fisher Scientific, #78532080):
https://www.thermofisher.com/antibody/product/MHC-Class-II-I-Ab-Antibody-clone-AF6-120-1-Monoclonal/78-5320-80
10. BV711-CD74 (1:200; BD, #740748):
https://www.bdbiosciences.com/en-de/products/reagents/flow-cytometry-reagents/research-reagents/single-color-antibodies-ruo/bv711-rat-anti-mouse-cd74.740748
- Bertolino P, Rabourdin-Combe C. The MHC class II-associated invariant chain: a molecule with multiple roles in MHC class II biosynthesis and antigen presentation to CD4+ T cells. Crit Rev Immunol. 1996; 16(4):359-379. (Biology).
- Bikoff EK, Huang LY, Episkopou V, van Meerwijk J, Germain RN, Robertson EJ. Defective major histocompatibility complex class II assembly, transport, peptide acquisition, and CD4+ T cell selection in mice lacking invariant chain expression. J Exp Med. 1993;

177(6):1699-1712. (Biology).
• Bodmer H, Viville S, Benoist C, Mathis D. Diversity of endogenous epitopes bound to MHC class II molecules limited by invariant chain. Science. 1994; 263(5151):1284-1286. (Biology).

11. PE-CD45R/B220 (1:100; BD, #561878):
https://www.bdbiosciences.com/en-de/products/reagents/flow-cytometry-reagents/research-reagents/single-color-antibodies-ruo/pe-rat-anti-mouse-cd45r-b220.561878
• Allman DM, Ferguson SE, Cancro MP. Peripheral B cell maturation. I. Immature peripheral B cells in adults are heat-stable antigenhi and exhibit unique signaling characteristics. J Immunol. 1992; 149(8):2533-2540. (Biology).
• Asensi V, Kimeno K, Kawamura I, Sakumoto M, Nomoto K. Treatment of autoimmune MRL/lpr mice with anti-B220 monoclonal antibody reduces the level of anti-DNA antibodies and lymphadenopathies. Immunology. 1989; 68(2):204-208. (Clone-specific).
• Ballas ZK, Rasmussen W. Lymphokine-activated killer cells. VII. IL-4 induces an NK1.1+CD8 alpha+beta- TCR-alpha beta B220+ lymphokine-activated killer subset. J Immunol. 1993; 150(1):17-30. (Biology).

12. PE-CD105 (1:100; Thermo Fisher Scientific, #12105182):
https://www.thermofisher.com/antibody/product/CD105-Endoglin-Antibody-clone-MJ7-18-Monoclonal/12-1051-82
13. PE-Ly6G (1:200; BioLegend, #127607):
https://www.biolegend.com/en-us/products/pe-anti-mouse-ly-6g-antibody-4777
• Lee T, et al. 2014. Mol Biol Cell. 25:583.
• Hernández-Santana YE, et al. 2020. Life Sci Alliance. 3:00.
• D'Alessandro G, et al. 2020. Eur J Immunol. 50:705.
• DeSouza-Vieira T, et al. 2020. Cell Rep. 33:108317.
• Bowling S, et al. 2020. Cell. 181(6):1410-1422.e27.
• Guo H, et al. 2020. Curr Protoc Immunol. 131:e107.
• Liang J, et al. 2021. Cancer Manag Res. 13:6977.
• Zhao D, et al. 2021. Innate Immun. 27:533.
• Zhou R, et al. 2022. EBioMedicine. 75:103762.
• Volmari A, et al. 2021. Hepatol Commun. 5:2104.
• Da Mesquita S, et al. 2021. Nature. 593:255.
• Combes F, et al. 2018. Neoplasia. 20:848.

14. PE-CD140a (1:100; BioLegend, #135905):
https://www.biolegend.com/en-us/products/pe-anti-mouse-cd140a-antibody-6253
• Gabitova-Cornell L, et al. 2020. Cancer Cell. 38(4):567-583.e11.
• Vercauteren Drubbel A, et al. 2021. Cell Stem Cell. .
• Zelic M, et al. 2021. Cell Reports. 35(6):109112.
• Buechler MB, et al. 2021. Nature. 593:575.
• Stoupa A, et al. 2018. EMBO Mol Med. 10:.
• Wagner G, et al. 2017. Sci Rep. 7:40881.
• Huang Z et al. 2017. Cell metabolism. 26(3):493-508 .
• Salzer MC et al. 2018. Cell. 175(6):1575-1590 .
• Chen M, et al. 2020. Sci Adv. 6:eaax9605.
• Wu R, et al. 2019. J Cell Mol Med. 24:1684.
• Biffi G, et al. 2018. Cancer Discov. 2:282.
• Cardot-Ruffino V, et al. 2020. Genesis. 58:e23359.

15. PE-O4 (1:100; Miltenyi, #130117507):
https://www.miltenyibiotec.com/DE-en/products/o4-antibody-anti-human-mouse-rat-o4.html
• G. Kantzer, C. et al. (2017) Anti-ACSA-2 defines a novel monoclonal antibody for prospective isolation of living neonatal and adult astrocytes. Glia (6) 65: 990 - 1004
• Bansal, R. et al. (1989) Multiple and novel specificities of monoclonal antibodies O1, O4, and R-mAb used in the analysis of oligodendrocyte development. J. Neurosci. Res. (4) 24: 548 - 557
• Zhang, S. C. (2001) Defining glial cells during CNS development. Nat. Rev. Neurosci. 2: 840 - 843
• Sommer, I. and Schachner, M. (1981) Monoclonal antibodies (O1 to O4) to oligodendrocyte cell surfaces: an immunocytological study in the central nervous system. Dev. Biol. 83: 311 - 327
• Jungblut, M. et al. (2012) Isolation and characterization of living primary astroglial cells using the new GLAST-specific monoclonal antibody ACSA-1. Glia (6) 60: 894 - 907
16. PE-Ter119 (1:100; Biolegend, #116207):
https://www.biolegend.com/en-us/products/pe-anti-mouse-ter-119-erythroid-cells-antibody-1867
• Guo H, Cooper S, Friedman A, et al. 2017. PLoS One. 10.1371/journal.pone.0150809.
• Grigsby SM, et al. 2021. Cancers (Basel). 13:.
• Furrer R, et al. 2021. Sci Adv. 7:eabi4852.
• Sun D, et al. 2021. Cell Stem Cell. .
• Schloss MJ, et al. 2022. Nat Immunol. 23:605.
• Xhima K, et al. 2020. Sci Adv. 6:eaax6646.
• Wong J, et al. 2015. Elife. 3: 07839.
• Hodzic D, et al. 2022. PLoS Biol. 20:e3001811.
• Hiraishi Y, et al. 2018. Sci Rep. 8:18052.
• Silva C, et al. 2019. Cell Physiol Biochem. 52:503.
• Papafragkos I, et al. 2022. Front Immunol. 13:889075.
• Endo Y, et al. 2020. FASEB J. 34:16086.

17. PE-Ly6C (1:100; Biolegend, #128007):
https://www.biolegend.com/en-us/products/pe-anti-mouse-ly-6c-antibody-4904
• Petersen B, et al. 2014. J Leukoc Biol. 95:809.
• Zuchtriegel G, et al. 2016. PLoS Biol. 14: 1002459.

- Jiang W, et al. 2017. Sci Rep. 7:6501.
- Gerwing M, et al. 2020. Mol Imaging Biol. 1.959027778.
- Tan L, et al. 2022. Biochem Biophys Rep. 32:101351.
- Zhou W, et al. 2019. Cell Syst. 0.597916667.
- Fu R, et al. 2020. Sci Rep. 10:1455.
- Radovanovic I, et al. 2014. J Immunol. 193:1290.
- Jiang W, et al. 2021. Oncol Lett. 22:625.
- Zhang YS, et al. 2018. Cancer Biol Ther. 19:735.
- Farsakoglu Y et al. 2019. Cell reports. 26(9):2307-2315 .
- Park JG, et al. 2021. iScience. 24(9):102941.

18. AF488-A2B5 (1:100; Novus Biologicals, #FAB1416G):
https://www.novusbio.com/products/a2b5-antibody-105_fab1416g

- Á Moreno-Gar, A Bernal-Chi, T Colomer, A Rodríguez-, C Matute, S Mato: Gene Expression Analysis of Astrocyte and Microglia Endocannabinoid Signaling during Autoimmune Demyelination Biomolecules, 2020;10(9):. 2020-01-01 [PMID: 32846891]

19. PE-Cy5-CD24 (1:200; Biolegend, #101811):
https://www.biolegend.com/en-ie/products/pe-anti-mouse-cd24-antibody-343

- Springer T, et al. 1978. Eur. J. Immunol. 8:539. (WB)
- Crowley M, et al. 1989. Cell. Immunol. 118:108. (FA)
- Veillette A, et al. 1989. J. Exp. Med. 170:1671. (FA)
- Pandelakis A Flavell RA 1999 JEM 189:855 (FC, IHC)
- Liu JQ, et al. 2007 J. Immunol. 178:6227. (FC, IF)
- Chappaz S, et al. 2007. Blood doi:10.1182/blood-2007-02-074245. (FC)
- Rucci F, et al. 2010. Proc Natl Acad Sci USA. 107:3024. (FC)
- Teague TK, et al. 2010. Int Immunol. 22:387. (FC)
- Gracz AD, et al. 2010. Am J. Physiol Gastrointest Liver Physiol. 298:590. (FC)
- Chen CY, et al. 2008. Endocrinology. 10:1210. (FC, IHC)
- Qui Q, et al. 2010. J. Immunol. 184:1681. (FC)

20. APC-Cy7-HB-EGF (Bioss, #BS-3576R-APC-CY7)
https://www.biossusa.com/products/bs-3576r-apc
21. AF488-HB-EGF (SantaCruz, #sc-365182 AF488)
https://www.scbt.com/p/hb-egf-antibody-h-1
22. PE-Cy7-CD31 (1:200; Thermo Fisher Scientific, #25031182):
https://www.thermofisher.com/antibody/product/CD31-PECAM-1-Antibody-clone-390-Monoclonal/25-0311-82
23. PerCP-eFlour710-CD86 (1:100; Thermo Fisher Scientific, #46086280):
https://www.thermofisher.com/antibody/product/CD86-B7-2-Antibody-clone-GL1-Monoclonal/46-0862-80
24. AF532-CD44 (1:100; Thermo Fisher Scientific, #58044182):
https://www.thermofisher.com/antibody/product/CD44-Antibody-clone-IM7-Monoclonal/58-0441-82
25. PE-Cy5.5-CD45 (1:200; Thermo Fisher Scientific, #35045180):
https://www.thermofisher.com/antibody/product/CD45-Antibody-clone-30-F11-Monoclonal/35-0451-80
26. JF646-MBP (1:100; Novus Biologicals, #NBP2-22121JF646):
https://www.novusbio.com/products/mbp-antibody-2h9_nbp2-22121jf646
27. APC-Cy7-Ly6G (1:200; Biolegend, #127623):
https://www.biolegend.com/en-us/products/apc-cyanine7-anti-mouse-ly-6g-antibody-6755

- Fleming TJ, et al. 1993. J. Immunol. 151:2399. (FC)
- Daley JM, et al. 2008. J. Leukocyte Biol. 83:1. (FC)
- Dietlin TA, et al. 2007. J. Leukocyte Biol. 81:1205. (FC)
- Daley J, et al. 2007. J. Leukocyte Biol. doi:10.1189. (Deplete)
- Tadagavadi RK, et al. 2010. J. Immunol. 185:4904.
- Sumagin R, et al. 2010. J. Immunol. 185:7057.
- Guiducci C, et al. 2010. J. Exp Med. 207:2931.
- Fujita M, et al. 2011. Cancer Res. 71:2664.
- Van Leeuwen, et al. 2008. Arterioscler. Thromb. Vasc. Biol. 28:84. (IHC)
- Kowanetz M, et al. 2010. P. Natl. Acad. Sci. USA 107:21248. [supplementary data] (IHC)
- Esbona K, et al. 2016. Breast Cancer Res. 18:35. (IHC)
- Wojtasiak M, et al. 2010. J. Gen. Virol. 91:2158. (FC, Deplete)

28. AF700-O4 (1:200; R&D, #FAB1326N):
https://www.rndsystems.com/products/oligodendrocyte-marker-o4-alexa-fluor-700-conjugated-antibody-o4_fab1326n

- Schachner, M. et al. (1981) Dev. Biol. 83:328.
- Bansal, R. et al. (1989) J. Neurosci. Res. 24:548.
- Bansal, R. and Pfeiffer, S.E. (1989) Proc. Natl. Acad. Sci. USA 86:6181.
- Gard, A. et al. (1995) Dev. Biol. 167:596.
- Reynolds, R. and Hardy, R. (1997) J. Neurosci. Res. 47:455.
- Ono, K. et al. (1997) J. Neurosci. Res. 48:212.
- Pang, Y. et al. (2000) J. Neurosci. Res. 62:510.
- Cai, Z. et al. (2001) Brain Res. 898:126.

29. AF660-CD19 (1:100; Thermo Fisher Scientific, #606019380):
https://www.thermofisher.com/antibody/product/CD19-Antibody-clone-eBio1D3-1D3-Monoclonal/606-0193-80
30. APC/Fire810-CD4 (1:100; Biolegend, #100479):
https://www.biolegend.com/en-us/products/apc-fire-810-anti-mouse-cd4-antibody-19552

- Dialynas DP, et al. 1983. J. Immunol. 131:2445. (Block, IP)
- Dialynas DP, et al. 1983. Immunol. Rev. 74:29. (IP, Deplete)
- Wu L, et al. 1991. J. Exp. Med. 174:1617. (Costim)
- Godfrey DI, et al. 1994. J. Immunol. 152:4783. (Block)

- Gavett SH, et al. 1994. Am. J. Respir. Cell. Mol. Biol. 10:587. (Deplete)
- Schuyler M, et al. 1994. Am. J. Respir. Crit. Care Med. 149:1286. (Deplete)
- Ghobrial RR, et al. 1989. Clin. Immunol. Immunopathol. 52:486. (Deplete)
- Israelski DM, et al. 1989. J. Immunol. 142:954. (Deplete)
- Zheng B, et al. 1996. J. Exp. Med. 184:1083. (IHC)
- Frei K, et al. 1997. J. Exp. Med. 185:2177. (IHC)
- Felix NJ, et al. 2007. Nat. Immunol. 8:388. (Block)
- Radtke AJ, et al. 2020. Proc Natl Acad Sci U S A. 117:33455-65. (SB)

31. PE-eFlour610-iNOS (1:100; Thermo Fisher Scientific, #61592080):
https://www.thermofisher.com/antibody/product/iNOS-Antibody-clone-CXNFT-Monoclonal/61-5920-80
32. BV711-IL17a (1:100; Biolegend, #506941):
https://www.biolegend.com/en-us/products/brilliant-violet-711-anti-mouse-il-17a-antibody-12030
- Kennedy J, et al. 1996. J. Interferon Cytokine Res. 16:611.
- Schubert D, et al. 2004. J. Immunol. 172:4503. (ICFC)
- Infante-Duarte C, et al. 2000. J. Immunol. 165:6107. (ICFC, ELISA Capture)
- Harrington LE, et al. 2005. Nature Immunol. doi:10.1038/ni1254. (ICFC, ELISA Capture)
- Nekrasova T, et al. 2005. J. Immunol. 175:2734. (ELISPOT Capture)
- Yen D, et al. 2006. J. Clin. Invest. 116:1310. (Neut)
- Ehirchiou D, et al. 2007. J. Exp. Med. 204:1519. (ICFC)
- Kang SG, et al. 2007. J. Immunol. 179:3724. (ICFC)
- Smith E, et al. 2008. J. Immunol. 181:1357. (Neut)
- Neufert C, et al. 2007. Eur. J. Immunol. 37:1809.
- Wang C, et al. 2009. Mucosal Immunol 2:173. (ICFC)
- Cui Y, et al. 2009. Invest. Ophth. Vis. Sci. 50:5811. (ICFC)

33. FITC-CXCL12 (1:100; Thermo Fisher Scientific, # MA523547):
https://www.thermofisher.com/antibody/product/CXCL12-Antibody-clone-79018-Monoclonal/MA5-23547
34. PE-Cy5-FoxP3 (1:100; Thermo Fisher Scientific, #15577382):
https://www.thermofisher.com/antibody/product/FOXP3-Antibody-clone-FJK-16s-Monoclonal/15-5773-82
35. PE-Cy7-IFNγ (1:100; Biolegend, #505826):
https://www.biolegend.com/en-us/products/pe-cyanine7-anti-mouse-ifn-gamma-antibody-5865
- Abrams J, et al. 1992. Immunol. Rev. 127:5. (ELISA, Neut)
- Sander B, et al. 1993. J. Immunol. Meth. 166:201. (ELISA, Neut)
- Abrams J, et al. 1995. Curr. Prot. Immunol. John Wiley and Sons, New York. Unit 6.20. (ELISA, Neut)
- Yang X, et al. 1993. J. Immunoassay 14:129. (ELISA)
- Klinman D, et al. 1994. Curr. Prot. Immunol. John Wiley and Sons, New York. Unit 6.19. (ELISPOT)
- Sander B, et al. 1991. Immunol. Rev. 119:65. (IHC)
- Ferrick D, et al. 1995. Nature 373:255. (FC)
- Ko SY, et al. 2005. J. Immunol. 175:3309. (FC)
- Peterson KE, et al. 2000. J. Virol. 74:5363. (Neut)
- DeKrey GK, et al. 1998. Infect. Immun. 66:827. (Neut)
- Dzhagalov I, et al. 2007. J. Immunol. 178:2113. (ELISA)
- Lawson BR, et al. 2007. J. Immunol. 178:5366. (FC)

36. PE PerCP-eFlour710-TNF (1:100; Thermo Fisher Scientific, #46732180):
https://www.thermofisher.com/antibody/product/TNF-alpha-Antibody-clone-MP6-XT22-Monoclonal/46-7321-80
37. APC-GM-CSF (1:100; Thermo Fisher Scientific, #17733182):
https://www.thermofisher.com/antibody/product/GM-CSF-Antibody-clone-MP1-22E9-Monoclonal/17-7331-82
- Medina-Reyes EI, et al. 2015. Environ Res. 136:424.
- Guillaumond F, et al. 2015. PNAS. 112:2473.
- Sharma SK, et al. 2015. J Immunol. 194:5529.
- Rodero MP, et al. 2014. J. Invest. Dermatol. 7:1991-7.

38. AF700-Ki67 (1:100; BioLegend, #652419):
https://www.biolegend.com/en-us/products/alexa-fluor-700-anti-mouse-ki-67-antibody-10366
- Medina-Reyes EI, et al. 2015. Environ Res. 136:424.
- Guillaumond F, et al. 2015. PNAS. 112:2473.
- Sharma SK, et al. 2015. J Immunol. 194:5529.
- Rodero MP, et al. 2014. J. Invest. Dermatol. 7:1991-7.
39. APC-eF780-Ki67 (1:100; Thermo Fisher Scientific, # 47569882):
https://www.thermofisher.com/antibody/product/Ki-67-Antibody-clone-SolA15-Monoclonal/47-5698-82

# Eukaryotic cell lines

Policy information about cell lines and Sex and Gender in Research

| | |
|---|---|
| Cell line source(s) | HEK293T (Invitrogen, #K1711)<br>HEK293FT (ThermoFisher, #R70007)<br>Human Astrocytes (ScienCell,#1800)<br>N2a cells (CAmerican Type Culture Collection, #CL-131) |
| Authentication | Cell lines were authenticated prior to receipt by the commercial vendor using the STR-based method |

| Mycoplasma contamination | Cells tested negative for mycoplasma contamination by the commcercial vendor and upon receipt. |
|---|---|
| Commonly misidentified lines<br>(See ICLAC register) | No commonly misidentified cell lines were used. |

# Animals and other research organisms

Policy information about studies involving animals; ARRIVE guidelines recommended for reporting animal research, and Sex and Gender in Research

| Laboratory animals | C57BL/6J (The Jackson Laboratory, #000664). Experiments were initiated in 8-12 week old mice. |
|---|---|
| Wild animals | The study did not involve wild animals. |
| Reporting on sex | EAE was induced in female mice only due to differences in susceptability and disease severity (PMID: 7517126; PMID: 15081249; PMID: 33190849). No sex-based analysis have been performed. For in vitro experiments, both sex were used. |
| Field-collected samples | Study did not involve field-collected samples |
| Ethics oversight | Bavarian State Authoritites (Regierung von Oberbayern, AZ 55.2-2532.Vet_02-19-49; Regierung von Unterfranken, AZ 55.2.2-2532-2-1306, 55.2.2-2532-2-1722). |

Note that full information on the approval of the study protocol must also be provided in the manuscript.

# Flow Cytometry

## Plots

Confirm that:

☒ The axis labels state the marker and fluorochrome used (e.g. CD4-FITC).

☒ The axis scales are clearly visible. Include numbers along axes only for bottom left plot of group (a 'group' is an analysis of identical markers).

☒ All plots are contour plots with outliers or pseudocolor plots.

☒ A numerical value for number of cells or percentage (with statistics) is provided.

## Methodology

| Sample preparation | Isolation of cells from adult mouse CNS<br>Mice were perfused with cold 1× PBS and the CNS was isolated and mechanically diced using sterile razors. Brains and spinal cords were processed separately or pooled (if not indicated otherwise) and transferred into 5 ml of enzyme digestion solution consisting of 35.5 µl papain suspension (Worthington, #LS003126) diluted in enzyme stock solution (ESS) and equilibrated to 37°C. ESS consisted of 10 ml 10× EBSS (Sigma-Aldrich, #E7510), 2.4 ml 30% D(+)-glucose (Sigma-Aldrich, #G8769), 5.2 ml 1 M NaHCO3 (VWR, #AAJ62495-AP), 200 µl 500 mM EDTA (Thermo Fisher Scientific, #15575020), and 168.2 ml ddH2O, filter-sterilized through a 0.22-µm filter. Samples were shaken at 80 rpm for 30–40 min at 37°C. Enzymatic digestion was stopped with 1 ml of 10× hi ovomucoid inhibitor solution and 20 µl 0.4% DNase (Worthington, #LS002007) diluted in 10 ml inhibitor stock solution (ISS). 10× hi ovomucoid inhibitor stock solution contained 300 mg BSA (Sigma-Aldrich, #A8806) and 300 mg ovomucoid trypsin inhibitor (Worthington, #LS003086) diluted in 10 ml 1× PBS and filter sterilized using a 0.22-µm filter. ISS contained 50 ml 10× EBSS (Sigma-Aldrich, #E7510), 6 ml 30% D(+)-glucose (Sigma-Aldrich, #G8769), and 13 ml 1 M NaHCO3 (VWR, #AAJ62495-AP) diluted in 170.4 ml ddH2O and filter-sterilized through a 0.22-µm filter. Tissue was mechanically dissociated using a 5-ml serological pipette and filtered through a 70-µm cell strainer (Fisher Scientific, #22363548) into a fresh 50-ml conical tube. Tissue was centrifuged at 600g for 5 min and resuspended in 10 ml of 30% Percoll solution (9 ml Percoll (GE Healthcare Biosciences, #17-5445-01), 3 ml 10× PBS, 18 ml ddH2O). Percoll suspension was centrifuged at 600g for 25 min with no breaks. Supernatant was discarded and the cell pellet was washed once with 1× PBS, centrifuged at 500g for 5 min and prepared for downstream applications.<br><br>Isolation of splenic cells<br>Spleens were mechanically dissected and dissociated by passing through a 100-µM cell strainer (Fisher Scientific, 10282631). Red blood cells were lysed with ACK lysing buffer (Life Technology, A10492-01) for 5 minutes and washed with 0.5% BSA and 2 mM EDTA at pH 8.0 in 1× PBS and prepared for downstream applications. |
|---|---|
| Instrument | Cytek Northern Lights (Cytek) |
| Software | SpectroFlo (v3.0) |
| Cell population abundance | For cell population abundances see the respective figures. Population abundances are depicted as percent of live cells. |

Gating strategy | Following FSC/SSC discrimination and doublet exclusion (SSC-H/SSC-A), hematopoetic/myeloid cells were differentiated from non hematopoetic/myeloid cells by CD45/CD11b. Subsets were then further gated based on the expression of specific surface markers (see Extended Data Figures 3b, 4e, 8e).

☒ Tick this box to confirm that a figure exemplifying the gating strategy is provided in the Supplementary Information.

