## [Peer Review File · Nature Immunology]

Peer Review Information

Journal: Nature Immunology

Manuscript Title: Astrocyte-produced HB-EGF limits autoimmune CNS pathology

Corresponding author name(s): Professor Veit Rothhammer

Reviewer Comments & Decisions:

Decision Letter, initial version:
--

24th Apr 2023

Dear Dr Rothhammer,

Thank you for sending a response to reviewers concerns regarding your article, "Astrocyte-produced HB-EGF limits autoimmune CNS pathology". In light of reviewers comments we cannot accept the manuscript for publication, but would be interested in considering a revised version that addresses these serious concerns as outlined in your response.

We hope you will find the referees' comments useful as you decide how to proceed. If you wish to submit a substantially revised manuscript, please bear in mind that we will be reluctant to approach the referees again in the absence of major revisions.

If you choose to revise your manuscript taking into account all reviewer and editor comments, please highlight all changes in the manuscript text file [OPTIONAL: in Microsoft Word format].

* If you have not done so already please begin to revise your manuscript so that it conforms to our Article format instructions at <http://www.nature.com/ni/authors/index.html>. Refer also to any guidelines provided in this letter.

The Reporting Summary can be found here:

[REDACTED]

If you wish to submit a suitably revised manuscript we would hope to receive it within 6 months. If you cannot send it within this time, please let us know. We will be happy to consider your revision so long as nothing similar has been accepted for publication at Nature Immunology or published elsewhere.

Nature Immunology is committed to improving transparency in authorship. As part of our efforts in this direction, we are now requesting that all authors identified as 'corresponding author' on published papers create and link their Open Researcher and Contributor Identifier (ORCID) with their account on the Manuscript Tracking System (MTS), prior to acceptance. ORCID helps the scientific community achieve unambiguous attribution of all scholarly contributions. You can create and link your ORCID from the home page of the MTS by clicking on 'Modify my Springer Nature account'. For more information please visit www.springernature.com/orcid.

Thank you for the opportunity to review your work.

Sincerely,

Stephanie Houston

Editor
Nature Immunology

Reviewers' Comments:

Reviewer #1:
Remarks to the Author:
MAJOR CONCERNS

In this manuscript, authors demonstrate a novel role for HB-EGF in neuroinflammation and show a very compelling ROC analysis using HB-EGF levels to discriminate CIS vs RRMS patients. In mouse systems, CNS HB is associated with disease improvement, and they use targeted inactivation of Hbegf in astrocytes as well as HB-EGF delivery in mice to demonstrate causality.

These findings are interesting and may be therapeutically important, although why the authors test these 14 analytes (leading up to the HB-EGF finding) is not provided - given the importance of this foundational experiment, this selection should be better justified.

According to Table I, RRMS patients tested are in remission (at least that's what I gleaned from the spreadsheet). Would an HB-EGF signal assessed at a peak relapse for RRMS (which don't seem to be assessed here) be potentially like a CIS signal and simply reflects overall burden of inflammation?

The structure of the manuscript - that is the back and forth between human and mouse - was problematic. Specifically, the little "hump" in clinical scores during the EAE chronic phase which they call "LSW" is what the authors use as a proxy for RRMS. While a discordance between what was done in MS versus EAE does not necessarily diminish the importance of the EAE findings, caution needs to be taken in drawing a direct line between "LSW" and the RRMS patients sampled herein.

Because the manuscript covers so much ground (human EAE, mouse MS, characterization of multiple glial cell and immune cell types, epigenetic changes etc etc) one is left with the impression that 2 or 3 papers have been condensed into a single study that is overall superficial in nature. This limited my enthusiasm for (and the readability of) the manuscript. See examples below:

Missing controls:

- In the CRISPR-Cas9 system that was driven by GFAP, no evidence is given that there were no off-target effects.
- Lines 187-190: the data described in Ext Data Fig 5a would be more compelling if one could compare these results against an HB-EGF-only control. Is an inflammatory stimulus required for HB-EGF to have effect on the measured gene expression?
- Lines 190-196 and Fig 4f-g: How do the authors control for the possibility that exogenous TNF/IL1b/HB-EGF added to the donor cells is driving response in recipient cells, with little or no impact from the conditioned medium on its own?
- Lines 2286-287 and Fig 6g: It is difficult to accept the suggestion that 5-Aza reverses late-stage EAE epigenetic Hbegf suppression without the baseline naïve control showing HB-EGF MFI in the absence of 5-Aza. At present, can one not as easily conclude the 5-Aza slightly increases HB-EGF expression

independent of inflammation?

Contradicting data:

- Fig 3m: The authors suggest that the AhR ligand I3S reduces activation of the Hbegf promotor, supported by Fig 3l yet apparently contradicted by Fig 3m, which does not appear to indicate a difference in Hbegf expression in the presence or absence of IS3. This may be resolved by appropriate statistical tests for both Fig 3l and Fig 3m. An alternate approach may be in vitro use of the Gfap::Hif1a or Gfap::Ahr crispr systems, if feasible.
- Line 312-315: The modest reduction in IFNg+ and IL17+ CD4+ T cells shown in Ext Data Fig 8c is substantially less convincing than the profiling done elsewhere in Fig 8, and only weakly supports the claim that rmHB-EGF during peak EAE “attenuated CNS inflammation”
- Fig 3j: this appears to contradict the idea that HB-EGF is part of a “rapid response mechanism” suggested earlier and in Fig 3c. How can these observations be reconciled?
- Fig. 6H (which compares RRMS vs control and leaves out CIS) contradicts the patient data in Fig. 1d in which RRMS vs controls are similar.

Overinterpretation of data:

- Lines 172-173: the authors do not appear to demonstrate that HIF1a controls HB-EGF. There does not appear to be evidence presented that the pseudohypoxic conditions result in HIF1a production, and the differences shown in Fig 3p are quite modest. To make this claim, it would seem more convincing to directly overexpress HIF1a or use a linked reporter system to show that HIF1a levels are somehow linked to HB-EGF expression, among other approaches.
- Lines 199-201 and Fig 4h: the authors fail to provide data to support the claim that HB-EGF blockade “increased pro-inflammatory gene expression in microglia”.
- Figure 4: it is not obvious to this reviewer how the legend claim that “HB-EGF exerts... protective effects on CNS-resident and -infiltrating cell types” is supported by the data presented
- Lines 228-232, Fig 5a-c, Ext Data Fig e-g: Given the large or lacking error bars, low numbers, and lack of statistics, it is not clear whether the claims in this section are indeed supported by the data presented.
- Lines 270-273, Fig 6c, Ext Data Fig 6b-c: Given the large or lacking error bars and lack of statistics, it is not clear whether the claims in this section are supported by the data presented.
- Line 303-304: Neither Fig 7e nor Ext Data Fig 7c does not support the claim that anything other than monocytes were decreased by this treatment.
- Line 306-308: Neither Fig 7j nor Ext Data Fig 7d support the claim that astrocyte “inflammatory potential” was changed by this treatment.

Issues with data quality:

- Line 93, Ext Data Fig 1h: it is difficult for the reader to interpret this experiment given the explanations present in the figure legend and text. How many biological replicates were used? Is there a statistical basis to claim different cellular composition between disease stages and not just animal(s) tested? Please clarify
- Fig 2b: this reviewer is unsure how to interpret this and subsequent regression analyses (Fig 7b, 7m), and the parameters used for the analysis. These analyses do not appear to be described in the methods. This presumably is a linear regression, which seems inappropriate given clear nonlinearity of the data. Please clarify.

- Fig 3d: These data show that fewer astrocytes contain HB-EGF 24h post TNF+IL1b. However, a difference of a couple percent positive astrocytes seems unlikely to be physiologically important. Moreover, will flow staining show both immature intracellular and immature membrane bound HB-EGF? These data – and all that follow – would be more convincing with some measurement of secreted HB-EGF (as in Fig 3h), which would give better insight into later experiments looking at ACM.
- Fig 3f, g, h, i, l, m: please clarify in the legend the length of time these experiments were incubated, as Fig 3c indicates that this is important.
- Fig 4e: “Descartes Cell Types and Tissue 2021” lacks explanation in legend and methods used are not described in the methods (line 915). The database/methodology also do not appear to be cited anywhere. Please fix.
- Fig 4f-i: these data cannot be interpreted as the incubation times are not described. Please fix.

MINOR CONCERNS

Results:

- Line 77: based on Ext Data Fig 1f, it is slightly misleading to claim that serum HB-EGF is reduced given that this is not strictly supported by the stats shown
- Line 97, Ext Data Fig 1i: it is unclear to this reviewer how the enriched pathways associate with “neuroprotective functions”, nor immediately obvious what the comparison being shown is (KEGG analysis of genes upregulated in peak vs LSW?). Better clarity may help here
- Lines 148-149: If HB-EGF is part of a rapid response mechanism, one would expect that protein levels would also be increased soon after expression depicted in Fig 3c. This may be a useful addition to support this speculation.
- Lines 151-152 and Fig 3e: the text and legend state that multiple predicted binding sites exist, as supported by supp table 2. Yet, Fig 3e appears to only show one predicted site. Is there a significance to this, or is this a representative example? Clarity in the legend may be helpful here.
- Lines 154-155: A minor point, but the authors state that CoCl₂, DFO, and DMOG all increase both membrane-bound and sHB-EFG production, yet these data are only shown for CoCl₂. Re-writing this line may improve understanding.
- Lines 179-182 and Fig 4b-d: it may be helpful to clarify in the text or legend how this analysis was conducted. E.g., what was the criteria used to decide “target genes” and “ligands” in 4b? Are the genes submitted to GO analysis only those that are HB-EGF target genes expressed by the dataset of Fig 4a, or of all mouse target genes? Similar question for Fig 4d – this reviewer finds these analyses confusing.
- Fig 4e is missing in the figure legend.

Figures:

- Lines 72-75, ext data fig 1d: this is an interesting case report, but difficult for the reader to interpret without knowing what the timepoints correspond to
- Fig 1j: this panel does not feel informational.
- Fig 2c: it would be helpful to add numbers to the legend. Are these absolute counts? What is the range? These data would likely be better accessible in a bar chart.
- Fig 2d-n: the figure title implies that all of these experiments correspond to cells collected at the end stage of EAE (day 25). Is this correct? Please clarify.

- Fig 2d-n: if as above these data are after EAE, what of steady-state? Are the changes shown dependent on induced neuroinflammation? Does Hbegf inactivation in homeostasis have similar effects? This may be a useful addition if the data exist.
- Fig 2j: it is difficult to discern which rows correspond to which labelled cell type. Adding horizontal lines between cell types may improve this, or greater spacing between labels.
- Fig 3c: the statistical tests used are not described in the legend. Are the comparisons each made relative to time 0? Were corrections for multiplicity made? Please clarify.
- Fig 3j: indicating x-axis units in the figure proper may improve clarity
- Fig 4f and g: the description of these experiments is somewhat difficult to follow. The schematic implies that donor cells are stimulated with TNF+IL1b and/or HB-EGF, but the x-axis of the graph implies that these compounds are added to ACM afterwards. Better clarity here may be helpful.
- Fig 5f is not referred to in the text
- Fig 5e-f: what is the rationale for doing this experiment in ex vivo optic nerve as opposed to quantification using the same systems established in prior experiments?
- Fig 7a: Presumably "MRT" should be "MRI"? The acronym is not referred to elsewhere

Discussion:

- Line 368-369: Given that Ext Data Fig 6e suggests that intranasal 5-Aza had no or worsened disease scores in mice, it seems inappropriate to suggest it as a potential human therapeutic.

Reviewer #2:

Remarks to the Author:

In summary, the authors identify a factor, HB-EGF, that is increased in the CSF of patients with CIS vs RRMS. HB-EGF is also increased in astrocytes in the acute phase of EAE but not in the late, chronic stage. The authors go on to demonstrate that HB-EGF attenuates proinflammatory signaling and exerts trophic effects on OPCs and neurons. The authors also show that HB-EGF expression is driven by HIF1 α and that methylation of HIF1 α binding sites is driven by inflammatory stimulation. Finally, intranasal HB-EGF is administered both at the beginning of symptom onset (acute EAE) and at disease peak (affecting chronic EAE) reduced disease and improved recovery.

The role of astroglial HB-EGF has been previously explored in MS but there are multiple new aspects, including the therapeutic effect of HB-EGF in both acute and chronic EAE. Moreover, the study is rigorously conducted and employs a number of synergistic models and datasets. The data is of high quality and presented in a lucid fashion.

A major critique of this study is that it is unclear whether CSF from RRMS patients was sampled during a relapse or during remission. The description is somewhat ambiguous: the extended data Fig1/b suggest that the RRMS patients were sampled during a relapse while a supplementary table lists RRMS patients as being in remission. This should be clarified.

If the latter is true, the upregulation of HB-EGF in CIS may only indicate the presence of acute lesions. The reduction of HB-EGF over time may then only reflect the decrease of acute demyelination in chronic MS patients rather than an evolving failure of tissue-protective mechanisms that leads to progression.

Other shortcomings are:

1. It is striking that progressive MS patients are not included in the CSF study, given that the authors

refer to MS progression throughout the introduction and discussion.

2. The authors suggest that increased promoter methylation at Hifa/ARNT binding sites might be responsible for reduced HB-EGF production in astrocytes over time. There is only one time point (disease peak) presented vs naïve mice. Promoter methylation e.g. at disease onset and during the late disease stage would make a more convincing argument for increasing promoter methylation during the disease course.

3. The initial observation has been done in CIS/MS patients, and the remainder of the study is conducted in mice and cell cultures. Looping back to human tissue and demonstrating e.g. increased promoter methylation at Hifa/ARNT binding sites in astrocytes within chronic active vs acute lesions, would translate the in vitro and in vivo observations back into MS.

Minor:

1. Blocking of HB-EGF in LPS-activated microglia incubated with ACM derived from hypoxic astrocytes did produce significant differences in Il1b or Csf2 (Fig 4h), therefore, this finding does not support "the idea that astrocyte-derived HB-EGF directly controls the inflammatory potential of microglia".

2. Line 333-334. "The negative correlation between HB-EGF levels in the CSF with the number of CNS lesions in CIS patients, ... ". The legend to Fig. 1f describes a linear regression analysis of HB-EGF concentration in the CSF of RRMS patients and lesion number.

Author Rebuttal to Initial comments

See inserted PDF

Reviewer #1

(Remarks to the Author)

MAJOR CONCERNS

In this manuscript, authors demonstrate a novel role for HB-EGF in neuroinflammation and show a very compelling ROC analysis using HB-EGF levels to discriminate CIS vs RRMS patients. In mouse systems, CNS HB-EGF is associated with disease improvement, and they use targeted inactivation of Hbegf in astrocytes as well as HB-EGF delivery in mice to demonstrate causality

- 1. These findings are interesting and may be therapeutically important, although why the authors test these 14 analytes (leading up to the HB-EGF finding) is not provided - given the importance of this foundational experiment, this selection should be better justified.**

We thank the Reviewer for the opportunity to clarify this point. The aim of this foundational experiment was to assess changes in protective factors in the CSF of MS patients at different disease stages. While in the recent years technological advancements have enabled untargeted CSF proteome profiling by mass spectrometry (e.g. PMID: 33087358, PMID: 35732154), the translatability of the methodology to the clinic is limited. To facilitate the clinical translation of our findings, we performed a multiplex analysis with a large panel of analytes using the ProcartaPlex Luminex platform in collaboration with Thermo Fisher Scientific. The analytes investigated in this experiment were chosen based on their preliminary relevance in the context of neuroinflammation and tissue-repair. In the revised version of the manuscript, we have analyzed an additional set of analytes, amounting to a total of 28 soluble factors with relevance in the context of tissue-repair and regeneration. The complementation of these new data supports our prior findings and reveals HB-EGF as the factor that is most significantly dysregulated between CIS and RRMS patients. Undeniably, this targeted approach cannot lead to the discovery of novel factors, which is an important limitation we now bring forward in the revised version of the discussion. However, we still believe that our initial analysis as well as the additional set of parameters analyzed are of great relevance and might possess translational potential. Finally, identification of HB-EGF as relevant factor in the CSF underlines the validity of both this approach and our findings.

- 2. According to Table I, RRMS patients tested are in remission (at least that's what I gleaned from the spreadsheet). Would an HB-EGF signal assessed at a peak relapse for RRMS (which don't seem to be assessed here) be potentially like a CIS signal and simply reflects overall burden of inflammation?**

We thank the Reviewer for the opportunity to clarify this and apologize for the unclarity in our description. RRMS patients and CIS patients used for analyses in Figure 1 were all sampled during relapse. Hence, both groups were sampled during acute inflammation, highlighting the fundamental difference between

the first (CIS) and subsequent relapses (RRMS) in MS. Therefore, the upregulation of HB-EGF in CIS patients, and its downregulation during peak relapse in RRMS cannot be explained by the overall burden of inflammation, which is present in both groups. To make this clearer, we highlighted this in the main text (lines 50-52) and figure legend in the revised version of the manuscript. Please also note that PBMCs initially used for the assessment of HB-EGF promoter methylation in PBMCs were sampled during remission, as this is the timepoint where the epigenetically-mediated suppression of HB-EGF supposedly affects regeneration and recovery. In the revised version of the manuscript, we have significantly expanded this dataset and now provide data on the extent of HBEGF promoter methylation in additional patient groups, including CIS, RRMS during relapse, and RRMS during remission, as well as SPMS patients.

3. The structure of the manuscript – that is the back and forth between human and mouse - was problematic. Specifically, the little “hump” in clinical scores during the EAE chronic phase which they call “LSW” is what the authors use as a proxy for RRMS. While a discordance between what was done in MS versus EAE does not necessarily diminish the importance of the EAE findings, caution needs to be taken in drawing a direct line between “LSW” and the RRMS patients sampled herein.

We agree with the Reviewer that the mouse model ineffectively recapitulates the complexity of the human disease and that no direct comparisons can be made based on the presented data. In the revised version of the manuscript, we have highlighted this limitation in lines 126-130 and 596-600. Nevertheless, we would like to point out that animals during LSW reach the same extent of clinical disability as during peak stages of EAE (e.g. Fig. 2a, Extended Data Fig. 2a), while the cellular composition in the CNS (Extended Data Fig. 2b) and the inflammatory capacity of both infiltrating and CNS-resident cell types (e.g. astrocytes, see Fig. 2, Extended Data Fig. 2) is distinct and specific, therefore representing two separated events and entities of autoinflammatory activation, which at least to some extent might recapitulate disease pathology in MS.

4. Because the manuscript covers so much ground (human EAE, mouse MS, characterization of multiple glial cell and immune cell types, epigenetic changes etc etc) one is left with the impression that 2 or 3 papers have been condensed into a single study that is overall superficial in nature. This limited my enthusiasm for (and the readability of) the manuscript.

We thank the Reviewer for this assessment. We would like to emphasize that the breadth of our study, including mechanistic insights, interspecies comparisons, and therapeutic aspects, rather than being arbitrary, highlights the biological relevance of astrocyte-derived HB-EGF in the context of MS. Due to the novel nature of our discovery of HB-EGF in the context of autoimmune CNS inflammation, we indeed sought to provide multiple synergetic lines of evidence reinforcing its implication in MS pathogenesis and exploring the potential underlying mechanisms. We believe our multi-layered, cross-species molecular, clinical and functional investigation was necessary as the outcomes converge to revealing HB-EGF as a pivotal player in tissue protection following autoimmune neuroinflammation in both animal models and human disease. We furthermore think that the present study follows a stringent line, describing the identification of HB-EGF in MS (Fig. 1), deciphering its producers (Fig. 2), its relevance

in the context of neuroinflammation (Fig. 3), its inducing signals relevant to MS (Fig. 4), its beneficial functions on the inflammatory capacity of CNS-infiltrating and -resident cells (Fig. 5), as well as its trophic effects (Fig. 6), why these protective effects are lost at later stages in disease (Fig. 7), and how restoring this lack of HB-EGF by exogenous supplementation can improve recovery and attenuate neuroinflammation (Fig. 8). Finally, we conclude the study by looping back to human pathology and provide evidence that similar mechanisms are at play in MS (Fig. 9). However, if the reviewer or the editorial team feel that a different narrative would benefit our findings, we will be happy to re-arrange the narrative and manuscript according to the suggestions.

See examples below:

Missing controls:

5. In the CRISPR-Cas9 system that was driven by GFAP, no evidence is given that there were no off-target effects.

We agree with the Reviewer that it is vital to demonstrate knockout specificity, as previously shown in Figures 2 and Extended Data Figure 3. Please note that the CRISPR/Cas9 system used in this study has extensively been used before (PMID: 36893254, PMID: 36599367, PMID: 36993446, PMID: 32051591, PMID: 33888612) demonstrating its relevance and potential for the investigation of astrocyte-specific functions. Nevertheless, in response to the Reviewers comment, we have screened for potential off-target effects and analyzed their regulation in our lentiviral knockout approach both in vitro and in vivo. Indeed, we observed no alteration in putative off-target genes (Table 1) by RT-qPCR analyses both in vitro and in vivo (Figure 1).

Table 1. putative off-target effects of the CRISPR/Cas9-Hbegf targeting vector.

Number	OFF TARGET SITE help_outline	MISMATCHES help_outline	CHROMOSOME help_outline	CUT SITE help_outline	PAM help_outline	GENE help_outline
1	GGTTTGTGGATATA GTGGGG	3	chr18	83,315,115	AGG	
2	GCTTTATGGATACA GTGGGA	3	chr9	54,805,148	AGG	
3	GGTCTGTGTGTCCA GTGGGA	3	chr10	77,893,170	GGG	
4	GGTTTGTGTATCGA GTTGGA	3	chr19	10,534,245	TGG	Tmem216
5	TGTTTGTGGATCCA TTGGCA	3	chr14	79,764,166	TGG	
6	GGTCTGTGGAACCA GTAGGA	3	chr9	21,509,182	TGG	Tmed1
7	GGTTTTGGGATGCA GTGGGA	3	chr7	42,586,952	AGG	Zfp977
8	GGTTTTGGGATGCA GTGGGA	3	chr7	43,116,040	AGG	Zfp936
9	AGGCTGTGGATCCA GTGGGA	3	chr13	8,533,551	AGG	Adarb2
10	GGTTTCTGGAACCA GTGGTA	3	chr1	14,755,119	AGG	Msc

Figure 1. Off-target effects of lentiviral CRISPR/Cas9 system. Expression of *Hbegf* and putative off target genes in ACSA2⁺ sorted astrocytes (upper) or primary mouse astrocytes transduced with *Gfap::Scrambl* control or *Gfap::Hbegf* targeting lentivirus. Unpaired t-test.

Moreover, we have performed additional analyses by immunohistochemistry and intracellular flow cytometry to demonstrate the cell type specific targeting of astrocytes using the GFP fluorescence reporter expressed under control of the GFAP promoter. Indeed, we observed a high overlap between GFAP⁺ astrocytes and GFP⁺ cells, demonstrating effective targeting of astrocytes, but not other cells in our CRISPR/Cas9 system (see Fig. 2 below).

Altogether, these results demonstrate high target-specificity without relevant off-target effects in the model system used. If the reviewer or the editor feels that this control dataset should be included in the revised version of the manuscript, we are happy to do so.

Figure 2. Cell-type specific targeting of lentiviral GFAP-driven CRISPR/Cas9 system. *a*, Percentage of GFAP⁺ cells of GFP⁺ cells and representative immunostaining in *Gfap::Hbegf* mice. Control mice that were injected with PBS were used as controls. *b*, flow cytometric analysis of GFP reporter staining (driven by GFAP) in astrocytes and non-astrocytes of *Gfap::Hbegf* mice and controls. Unpaired t-test.

6. Lines 187-190: the data described in Ext Data Fig 5a would be more compelling if one could compare these results against an HB-EGF-only control. Is an inflammatory stimulus required for HB-EGF to have effect on the measured gene expression?

We thank the Reviewer for bringing this important point forward. In response to the comment, we have performed additional experiments suggested by the Reviewer and included an unstimulated, as well as an HB-EGF only control in the revised version of the manuscript (Extended Data 7a). Overall, we have observed that the anti-inflammatory effects of HB-EGF are strongest upon inflammatory activation of the target cells, which also underlines the relevance of HB-EGF during inflammatory conditions proposed in this manuscript.

7. Lines 190-196 and Fig 4f-g: How do the authors control for the possibility that exogenous TNF/IL1b/HB-EGF added to the donor cells is driving response in recipient cells, with little or no impact from the conditioned medium on its own?

We thank the Reviewer for the opportunity to clarify this point. Indeed, after 24 hours of activation with TNF- α and IL-1 β , we performed extensive washes of our astrocyte cultures before adding fresh culture medium without stimulatory cytokines. The conditioned medium was collected only after extensive washing and thus devoid of the initial cytokines used during the initial stimulation. This “new” stimulation-

cytokine free medium was then added onto recipient cells. In the revised version of the manuscript, we have included a more detailed description in the main text and methods section, as well as a schematic outlining the experimental design. Please also note that the respective approach has been used in several studies already (e. g. PMID: 29769726, PMID: 33888612, PMID: 36893254).

- 8. Lines 2286-287 and Fig 6g: It is difficult to accept the suggestion that 5-Aza reverses late-stage EAE epigenetic Hbegf suppression without the baseline naïve control showing HB-EGF MFI in the absence of 5-Aza. At present, can one not as easily conclude the 5-Aza slightly increases HB-EGF expression independent of inflammation?**

We thank the Reviewer for this important comment. In the revised version of the manuscript, we have performed additional experiments assessing the extent of Hbegf promoter methylation in naïve mice (Fig. 7d), as well as non-inflammatory control patients (Fig. 9), overall demonstrating only limited promoter hypermethylation in the absence of autoimmune inflammation. Additionally, in response to the Reviewers comment, we have intranasally treated naïve mice with 5-Aza or vehicle for 14 days and analyzed the effect on HB-EGF expression in astrocytes (Extended Data Fig. 8h-i), revealing no effect of the hypomethylating agent 5-Aza in the absence of inflammation. We have therefore concluded that inflammation induced hypermethylation is required for the effect of 5-Aza on HB-EGF expression by astrocytes.

Contradicting data:

- 9. Fig 3m: The authors suggest that the AhR ligand I3S reduces activation of the Hbegf promotor, supported by Fig 3l yet apparently contradicted by Fig 3m, which does not appear to indicate a difference in Hbegf expression in the presence or absence of IS3. This may be resolved by appropriate statistical tests for both Fig3l and Fig 3m. An alternate approach may be in vitro use of the Gfap::Hif1a or Gfap::Ahr crispr systems, if feasible.**

We thank the Reviewer for highlighting this point. Indeed, stimulation with I3S resulted in no statistically significant difference of Hbegf expression by astrocytes when corrected for multiple testing. This lack of nominal significance was most likely driven by the much stronger positive regulation of Hbegf by pseudohypoxic conditions and is well expected, as a relative decrease from low basal expression will result in a smaller difference between group means compared to a strong inducer. Nevertheless, blockage of AhR signaling by CH-223191 synergistically increases Hbegf expression in addition to a pseudohypoxic stimulus when compared to stimulation with a pseudohypoxic stimulus alone, demonstrating the competitive effects of HIF1a and AhR signaling in the regulation of Hbegf. Moreover, individual testing of controls versus I3S treated controls highlights the significant reduction of Hbegf expression upon exposure to the AhR activating ligand I3S (see Fig. 3 below).

Figure 3. Control of astrocytic Hbegf expression by Ahr. RT-qPCR analysis of Hbegf expression in primary mouse astrocytes in response to 24 hours of stimulation with I3S. Unpaired t-test.

Moreover, we have observed both antagonistic and cooperative effects of HIF1 α and AhR signaling (e.g. Fig. 4k, Extended Data Fig. 5j), such multilevel mutual interaction between both transcription factors has recently been highlighted in several studies (PMID: 36725335, PMID: 35988806, PMID: 35473600). Nevertheless, to define net effects of AhR and HIF1 α on Hbegf expression in our system, we have performed the experiments suggested by the Reviewer: *in vitro* knockout experiments of Ahr and Hif1 α (Fig. 4l-m) demonstrated the upregulation of HB-EGF by astrocytes when AhR signaling is abrogated, while defunct HIF1 α signaling reduces astrocytic HB-EGF expression. Thus, this new dataset suggests that (i) AhR activation reduces HB-EGF expression, while (ii) pseudohypoxia activating HIF1 α enhances Hbegf expression, and that (iii) interaction of AhR and Hif1 α exists on multiple levels most likely by their competition for ARNT binding.

10. Line 312-315: The modest reduction in IFN γ + and IL17+ CD4+ T cells shown in Ext Data Fig 8c is substantially less convincing than the profiling done elsewhere in Fig 8, and only weakly supports the claim that rmHB-EGF during peak EAE “attenuated CNS inflammation”

We thank the Reviewer for the opportunity to address this point. In the animal model used in this study, CD4 T cells drive the disease during early stages, but play a less important role during late stage neuroinflammation, where sustained inflammation and degeneration are perpetuated due to prior T cell-mediated activation of glial cells. This interpretation is in line with our data, where the effect of late-stage intranasal administration of HB-EGF only leads to a minor, but nevertheless significant reduction in pro-inflammatory cytokines by T cells in the CNS, while the main alterations are found in glial cells.

To strengthen the point that rmHB-EGF administration attenuates inflammatory processes, we have performed additional treatment experiments and analyzed the extent of inflammatory infiltration and activation at earlier timepoints (day 10, Fig. 8a-e), when T cell responses play a major role. Indeed, this approach revealed significant effects of rmHB-EGF administration on the inflammatory potential of infiltrating immune cells, particularly in the spinal cord, which is the predominant site of infiltration in this model. Together, this new set of experiments further strengthens the characteristics of early versus late stage autoimmune neuroinflammation conceptualized throughout the manuscript and supports the anti-inflammatory potential of exogenous HB-EGF supplementation.

11. Fig 3j: this appears to contradict the idea that HB-EGF is part of a “rapid response mechanism” suggested earlier and in Fig 3c. How can these observations be reconciled?

We thank the Reviewer for bringing this important point forward. Please note that the rapid upregulation of HB-EGF occurs in response to stimulation with TNF α and IL1 β , while the sustained upregulation occurs in response to pseudohypoxic stimuli. In the revised version of the manuscript, we have reconciled these data by analyzing the response kinetics to TNF α /IL1 β by a luciferase reporter system as used for CoCl $_2$ (Fig.3j), as well as by analyzing the temporal expression on mRNA level in response to CoCl $_2$ as performed in Fig. 3c. We found that both stimuli lead to a rapid upregulation of Hbegf after 4 hours of stimulation (Fig. 4b, 4e, 4j). However, while continuous stimulation with TNF α /IL1 β suppressed Hbegf expression (Fig. 4b, 4j), pseudohypoxic conditions result in a sustained upregulation of Hbegf (Fig. 4e, 4j).

12. Fig. 6H (which compares RRMS vs control and leaves out CIS) contradicts the patient data in Fig. 1d in which RRMS vs controls are similar.

We thank the Reviewer for making this important point. Please note that the sample sets analyzed in Figure 1 and Figure 6 in the initial version of the manuscript were biologically and technically different and chosen to answer different specific questions at hand: Figure 1 described the absolute concentration of HB-EGF in the CSF and serum of controls, CIS, and RRMS patients during relapse, showing differential regulation of HB-EGF during first or subsequent relapses. In contrast, Fig. 6h referred to the methylation of the HBEGF promoter in RRMS patients during remission compared to non-inflammatory control patients. This is in line with our hypothesis, as we argue that HB-EGF promoter methylation and its subsequent suppression is the result of inflammatory activation during a relapse. To further strengthen this point, we have analyzed the extent of methylation of the HBEGF promoter in MS at additional disease stages, including CIS, RRMS during relapse, RRMS during remission, and SPMS, validating our hypothesis and strengthening the concept brought forward in this manuscript.

Overinterpretation of data:

13. Lines 172-173: the authors do not appear to demonstrate that HIF1 α controls HB-EGF. There does not appear to be evidence presented that the pseudohypoxic conditions result in HIF1 α production, and the differences shown in Fig 3p are quite modest. To make this claim, it would seem more convincing to directly overexpress HIF1 α or use a linked reporter system to show that HIF1 α levels are somehow linked to HB-EGF expression, among other approaches.

We thank the Reviewer for these great suggestions. Accordingly, we have performed additional experiments where we co-transfected a stable and active HIF1 α and assessed the effects on HBEGF promoter activation in a linked luciferase reporter assay (Fig. 4i), highlighting a positive regulation of HB-EGF by HIF1 α .

14. Lines 199-201 and Fig 4h: the authors fail to provide data to support the claim that HB-EGF blockade “increased pro-inflammatory gene expression in microglia”.

We agree with the Reviewer that further inflammatory genes need to be assessed in order to make that claim. Therefore, we have performed additional HB-EGF blockage experiments and analyzed additional pro-and anti-inflammatory factors (Cd68, Il1b, Tnf, Lif) expressed by microglia (Fig. 5j). Additionally, we have toned down our statement in the main text in order to avoid overinterpretation.

15. Figure 4: it is not obvious to this reviewer how the legend claim that “HB-EGF exerts... protective effects on CNS-resident and -infiltrating cell types” is supported by the data presented

We thank the Reviewer for pointing this out and have toned down the statement in the revised version of the manuscript, solely focusing on the anti-inflammatory effects of HB-EGF in Figure 4, while Figure 5 describes the trophic effects of HB-EGF on oligodendrocytes and neurons.

16. Lines 228-232, Fig 5a-c, Ext Data Fig e-g: Given the large or lacking error bars, low numbers, and lack of statistics, it is not clear whether the claims in this section are indeed supported by the data presented.

We apologize for this oversight and have now repeated these experiments, increased sample size and provided appropriate statistics to support our findings. Furthermore, we have significantly restructured Figures 5 and Extended Data Fig. 7 and rephrased the respective sections in the manuscript to provide a clearer picture of the trophic effects of HB-EGF on oligodendrocytes.

17. Lines 270-273, Fig 6c, Ext Data Fig 6b-c: Given the large or lacking error bars and lack of statistics, it is not clear whether the claims in this section are supported by the data presented.

In response to the Reviewers comment, we have significantly expanded our analyses in order to support the statistical validity of our findings. We have isolated astrocytes from naïve and EAE mice at peak and recovery stages and analyzed the extent of promoter methylation around Hif1a binding sites by targeted high-resolution melt analysis following bisulfite conversion (Fig. 7d). These data demonstrate that hypermethylation of the Hbegf promoter in close proximity to Hif1a binding sites is present during peak of EAE, and persist throughout recovery stages, further strengthening our observations from whole genome bisulfite sequencing (Fig. 7c).

18. Line 303-304: Neither Fig 7e nor Ext Data Fig 7c does not support the claim that anything other than monocytes were decreased by this treatment.

We thank the Reviewer for raising this important point. Indeed, monocytes are among the most important cell types during late stages of autoimmune CNS inflammation. While we observed an overall reduction of infiltrating immune cells (including monocytes, T cells, B cells, etc) in rmHB-EGF treated animals, the strongest effects are highlighted by their reduced production of pro-inflammatory cytokines (Extended Data Fig. 7d). This was also validated in a new set of experiments, where we intranasally administered

HB-EGF starting at symptom onset and analyzed the cellular composition and pro-inflammatory activation of CNS-resident and -infiltrating immune cells at an earlier timepoint. At this timepoint, infiltrating immune cells are the main drivers of disease and strongly drive spinal cord pathology in the mouse model used in this study, while during later stages CNS-resident cells like microglia or astrocyte perpetuate the inflammatory environment and contribute to disease progression. We have observed a significant reduction of CD4 T cells, monocytes, and dendritic cells, as well as their pro-inflammatory activation at these earlier stages (Fig. 9a-e). Moreover, transcriptional analyses of astrocytes at later timepoints following rmHB-EGF treatment revealed a decrease in pro-inflammatory gene expression, and an increase in tissue-protective gene expression (Fig. 9m), overall indicating that rmHB-EGF has the capacity to attenuate inflammatory signaling during early and late stages of EAE.

19. Line 306-308: Neither Fig 7j nor Ext Data Fig 7d support the claim that astrocyte “inflammatory potential” was changed by this treatment.

We agree with the Reviewer that the cytokines analyzed by intracellular flow cytometry do not fully reflect the inflammatory activation of astrocytes. In order to address this, we have performed additional experiments and analyzed the inflammatory/protective potential of ACSA2-sorted astrocytes during EAE mice by RT-qPCR, revealing a decrease in pro-inflammatory gene expression and an increase in tissue-trophic gene expression (Fig. 9m).

Issues with data quality:

20. Line 93, Ext Data Fig 1h: it is difficult for the reader to interpret this experiment given the explanations present in the figure legend and text. How many biological replicates were used? Is there a statistical basis to claim different cellular composition between disease stages and not just animal(s) tested? Please clarify

We apologize for the lack of clarity. Indeed, the data shows the concatenation of biological replicates at each timepoint. In order to make this more clear, we have included additional boxplots and appropriate statistical tests in the revised version of the manuscript (Extended Data Fig. 2b-c).

21. Fig 2b: this reviewer is unsure how to interpret this and subsequent regression analyses (Fig 7b, 7m), and the parameters used for the analysis. These analyses do not appear to be described in the methods. This presumably is a linear regression, which seems inappropriate given clear nonlinearity of the data. Please clarify.

We thank the Reviewer for his comment. While linear regression analyses are commonly used for statistical comparisons of EAE experiments (e.g. PMID: 20574007, PMID: 31813625), we have added additional information in the respective figure legends and the methods section of the revised manuscript in order to improve clarity and enhance data quality as suggested.

22. Fig 3d: These data show that fewer astrocytes contain HB-EGF 24h post TNF+IL1b. However, a difference of a couple percent positive astrocytes seems unlikely to be physiologically important.

*We thank the Reviewer for the opportunity to address this important point. Please note that astrocytes require activation to express HB-EGF to a high extent, while its levels remain low during basal conditions (Fig. 2d-g). In these lines, a decrease of 50% from basal HB-EGF levels following stimulation with TNF α /IL1b represents a significant observation. Moreover, the functional and physiological relevance of HB-EGF produced by astrocytes is supported by the observations of enhanced disease severity and inflammation in *Gfap::Hbegf* mice and in vitro experiments, where blockage of astrocyte-derived HB-EGF using a HB-EGF specific antibody leads to alterations in the tissue-protective polarization of microglia. In order to further strengthen this point, we have performed multiple ELISA measurements of sHB-EGF following stimulation of primary mouse astrocytes with TNF α /IL1b. These measurements demonstrate that astrocytes are able to rapidly increase their secretion of sHB-EGF, from basal levels around 10 pg/ml up to 30 pg/ml in the supernatant within 8 hours of stimulation with TNF α /IL1b. After 8 hours, however, we observed no further increase in sHB-EGF, but its levels remained stable in the supernatant, indicating no relevant further de novo synthesis of HB-EGF after the initial peak. Overall, we argue that these responses represent physiologically relevant mechanisms that control inflammatory and degenerative processes in the CNS (as demonstrated in Fig. 3).*

23. Moreover, will flow staining show both immature intracellular and immature membrane bound HB-EGF? These data – and all that follow – would be more convincing with some measurement of secreted HB-EGF (as in Fig 3h), which would give better insight into later experiments looking at ACM.

This is an important point brought forward by the Reviewer. In the revised version of the manuscript, we have added additional datasets and used antibodies that stain the immature membrane bound form and the cytoplasmic domain of HB-EGF and quantified their regulation in response to stimulation with TNF α /IL1b (Extended Data Fig. 5a). We have furthermore performed multiple ELISA measurements, as described in a previous response, demonstrating a rapid upregulation, followed by a steady decline in HB-EGF production by activated astrocytes. Finally, the physiological relevance of this upregulation was addressed in additional experiments using blocking antibodies as outlined before.

24. Fig 3f, g, h, i, l, m: please clarify in the legend the length of time these experiments were incubated, as Fig 3c indicates that this is important.

We apologize for the lack of this information and have added the respective timepoints in the revised version of the manuscript.

25. Fig 4e: “Descartes Cell Types and Tissue 2021” lacks explanation in legend and methods used are not described in the methods (line 915). The database/methodology also do not appear to be cited anywhere. Please fix.

We apologize for the lack of this information and have provided additional information in the figure legends and methods section of the revised manuscript.

26. Fig 4f-i: these data cannot be interpreted as the incubation times are not described. Please fix.

In response to the Reviewers comments, we have included an improved schematic outlining the experimental design and described the incubation times in detail in the respective figure legend.

MINOR CONCERNS

Results:

27. Line 77: based on Ext Data Fig 1f, it is slightly misleading to claim that serum HB-EGF is reduced given that this is not strictly supported by the stats shown

In response to the Reviewers comment, we have rephrased and toned down this statement in the revised version of the manuscript (lines 145-147).

28. Line 97, Ext Data Fig 1i: it is unclear to this reviewer how the enriched pathways associate with “neuroprotective functions”, nor immediately obvious what the comparison being shown is (KEGG analysis of genes upregulated in peak vs LSW?). Better clarity may help here

We thank the Reviewer for the opportunity to clarify this. Indeed, Figure 1i (now Extended Data Fig. 2g) shows the enrichment of dysregulated genes in peak vs. LSW astrocytes that are associated to roles in axon regeneration, axon guidance, and sphingolipid signaling - altogether pathways that have been shown to play important roles in MS (and in particular during recovery phases) (PMID: 31813625, PMID: 26419927, PMID: 33888612). To improve clarity, we have moved these data into Extended Data Fig. 2 and rephrased the respective section in the main text.

29. Lines 148-149: If HB-EGF is part of a rapid response mechanism, one would expect that protein levels would also be increased soon after expression depicted in Fig 3c. This may be a useful addition to support this speculation.

We thank the Reviewer for this helpful suggestion and have accordingly incorporated additional flow cytometric measurements on the temporal expression of both membrane-bound and cytoplasmic HB-EGF in the revised version of the manuscript (Fig. 4, Extended Data Fig. 5).

30. Lines 151-152 and Fig 3e: the text and legend state that multiple predicted binding sites exist, as supported by supp table 2. Yet, Fig 3e appears to only show one predicted site. Is there a significance to this, or is this a representative example? Clarity in the legend may be helpful here.

We thank the Reviewer for pointing this out. Indeed, this is a representative example. As suggested by the Reviewer, we have clarified this point in Figure Legend 4d of the revised manuscript.

31. Lines 154-155: A minor point, but the authors state that CoCl₂, DFO, and DMOG all increase both membrane-bound and sHB-EFG production, yet these data are only shown for CoCl₂. Re-writing this line may improve understanding.

We apologize for this lack of clarity. Indeed, while we have shown the upregulation of Hbegf in response to stimulation with CoCl₂, DFO, and DMOG, we have only provided ELISA measurement of HB-EGF in response to CoCl₂ in the initial version of the manuscript. To enhance clarity, we now only show stimulations performed with CoCl₂ in the revised version of the manuscript.

32. Lines 179-182 and Fig 4b-d: it may be helpful to clarify in the text or legend how this analysis was conducted. E.g., what was the criteria used to decide “target genes” and “ligands” in 4b? Are the genes submitted to GO analysis only those that are HB-EGF target genes expressed by the dataset of Fig 4a, or of all mouse target genes? Similar question for Fig 4d – this reviewer finds these analyses confusing.

We thank the Reviewer for the opportunity to clarify this. In fact, the ligands and target genes (Fig.4b, now Fig. 5b) are predicted by NicheNet (PMID: 3181926) by combining the expression data obtained from scRNA-Seq with a prior knowledge model on ligand–target links. This tool aids the unsupervised identification of cell-to-cell interactions based on their transcriptomic profile. The HB-EGF target genes provided by NicheNet based on the dataset by Wheeler et al. (PMID: 32051591) were then subjected to GO analysis. We have described these steps in the methods section and figure legend. For a more detailed explanation on how target genes are identified by NicheNet based on single-cell sequencing data, we kindly refer to the original publication of the methodology (PMID: 3181926).

33. Fig 4e is missing in the figure legend.

We apologize for this oversight and have added the information in the revised version of the manuscript.

Figures:

34. Lines 72-75, ext data fig 1d: this is an interesting case report, but difficult for the reader to interpret without knowing what the timepoints correspond to

We agree with the Reviewer that a singular case can only provide a limited amount of information. However, serial CSF sampling of RRMS patients is not commonly performed due to the invasive nature of the procedure. To address the Reviewers comment, we have added the time between sampling timepoints in the respective figure legend.

35. Fig 1j: this panel does not feel informational.

We thank the Reviewer for the opportunity to clarify this. Indeed, the pie chart depicts the protein class of genes significantly downregulated in LSW astrocytes compared to peak EAE. Among these, Hbegf and various other trophic and anti-inflammatory factors were downregulated in the CSF samples analyzed here. This highlights the notion that the loss of protective functions in astrocytes during late stage autoimmune CNS inflammation is not limited to Hbegf, but also observed for other soluble mediators that have previously been associated to tissue-protective functions (e.g. PMID: 36893254, PMID: 24860191). In order to further clarify this notion, we have provided additional reasoning in the discussion part of the revised manuscript.

36. Fig 2c: it would be helpful to add numbers to the legend. Are these absolute counts? What is the range? These data would likely be better accessible in a bar chart.

Indeed, the figure depicts absolute counts of HB-EGF+ cells. In response to the Reviewers comments, we have included a numerical range in the revised version of the manuscript and further provided additional information in the figure legend.

37. Fig 2d-n: the figure title implies that all of these experiments correspond to cells collected at the end stage of EAE (day 25). Is this correct? Please clarify.

We thank the Reviewer for the opportunity to clarify this. Indeed, all data were collected during late stage EAE (day 25). To clarify this, we added this information in the figure legends in the revised version of the manuscript.

38. Fig 2d-n: if as above these data are after EAE, what of steady-state? Are the changes shown dependent on induced neuroinflammation? Does Hbegf inactivation in homeostasis have similar effects? This may be a useful addition if the data exist.

We thank the Reviewer for this insightful suggestion. In this study, we have focused on the role of astrocyte-derived HB-EGF in the context of autoimmune CNS inflammation. Indeed, multiple roles of HB-EGF during CNS development and under steady state have been described and it is reasonable to assume that CRISPR/Cas9-mediated abrogation of HB-EGF expression by astrocytes under steady state may affect CNS-resident cells (e.g. PMID: 23006514). However, we would also like to point out that the expression of HB-EGF by astrocytes relies on their activation, as demonstrated in Figure 2. Thus, we believe that focusing on the effects of HB-EGF during autoimmune inflammatory conditions such as in MS highlights the importance of HB-EGF especially in this context, which is the focus of this study.

39. Fig 2j: it is difficult to discern which rows correspond to which labelled cell type. Adding horizontal lines between cell types may improve this, or greater spacing between labels.

We thank the Reviewer for this helpful suggestion and have added horizontal lines between cell types in the revised version of the manuscript.

40. Fig 3c: the statistical tests used are not described in the legend. Are the comparisons each made relative to time 0? Were corrections for multiplicity made? Please clarify.

We apologize for the oversight and have added the respective information in the revised version of the manuscript. Indeed, for all timecourse experiments, all comparisons were made to a baseline control. All statistical analyses are described in the figure legends and method section. For timecourse experiments, One-way ANOVA with Dunnett's multiple comparisons test was performed if not otherwise indicated.

41. Fig 3j: indicating x-axis units in the figure proper may improve clarity

We thank the Reviewer for this suggestion. In the revised version of the manuscript, we have excluded these data and instead provide a more comprehensive analysis on the effect of CoCl₂ stimulation on HBEGF promoter activation (Fig. 4j).

42. Fig 4f and g: the description of these experiments is somewhat difficult to follow. The schematic implies that donor cells are stimulated with TNF+IL1b and/or HB-EGF, but the x-axis of the graph implies that these compounds are added to ACM afterwards. Better clarity here may be helpful.

To address the Reviewers comment, we have added an improved schematic outlining the experimental design and timepoints. Furthermore, we have added detailed descriptions about the stimulation periods and procedure in the figure legend. Indeed, HB-EGF was added simultaneously with TNFa/IL1b, while the ACM/MGCM only contained the factors produced by the respective producer cell.

43. Fig 5e-f: what is the rationale for doing this experiment in ex vivo optic nerve as opposed to quantification using the same systems established in prior experiments?

In addition to in vitro experiments using primary cell culture, we aimed to validate our observations in an ex vivo system that is closer to physiological conditions. By using ex vivo optic nerve cultures, we were able to investigate the trophic effects of HB-EGF in a complex setting based on a multicellular system, therefore strengthening prior observations in isolated cell culture.

44. Fig 7a: Presumably "MRT" should be "MRI"? The acronym is not referred to elsewhere

We thank the Reviewer for pointing this out and have corrected the acronym in the revised version of the manuscript.

Discussion:

45. Line 368-369: Given that Ext Data Fig 6e suggests that intranasal 5-Aza had no or worsened disease scores in mice, it seems inappropriate to suggest it as a potential human therapeutic.

We agree with the Reviewer that the current data from our EAE experiments do not suggest a therapeutic value of 5-Aza in the context of neuroinflammation. Indeed, in these experiments, we used 5-Aza as a means to modulate hypermethylation rather than as a therapeutic strategy. One potential reason for the lack of clinical benefit may be the dosage, as previous reports suggest that low-dose 5-Aza treatment effectively ameliorates EAE (PMID: 24869907). While additional dose-response studies using 5-Aza will be necessary to claim its therapeutic validity, we nevertheless believe that our data offers novel therapeutic avenues for MS: for example, targeted demethylation of specific loci may circumvent potentially toxic effects of applying hypomethylating agents like 5-Aza throughout the CNS. In the updated version of the manuscript, we now discuss these limitations and implications in more detail (lines 622-633) and have toned down the statement of potential therapeutic applicability.

Reviewer #2

(Remarks to the Author)

In summary, the authors identify a factor, HB-EGF, that is increased in the CSF of patients with CIS vs RRMS. HB-EGF is also increased in astrocytes in the acute phase of EAE but not in the late, chronic stage. The authors go on to demonstrate that HB-EGF attenuates proinflammatory signaling and exerts trophic effects on OPCs and neurons. The authors also show that HB-EGF expression is driven by HIF1 α and that methylation of HIF1 α binding sites is driven by inflammatory stimulation. Finally, intranasal HB-EGF is administered both at the beginning of symptom onset (acute EAE) and at disease peak (affecting chronic EAE) reduced disease and improved recovery. The role of astroglial HB-EGF has been previously explored in MS but there are multiple new aspects, including the therapeutic effect of HB-EGF in both acute and chronic EAE. Moreover, the study is rigorously conducted and employs a number of synergistic models and datasets. The data is of high quality and presented in a lucid fashion.

We thank the Reviewer for the positive assessment of our manuscript.

- 1. A major critique of this study is that it is unclear whether CSF from RRMS patients was sampled during a relapse or during remission. The description is somewhat ambiguous: the extended data Fig1/b suggest that the RRMS patients were sampled during a relapse while a supplementary table lists RRMS patients as being in remission. This should be clarified.**

We thank the Reviewer for the opportunity to clarify this and apologize for the ambiguous wording. Indeed, all CIS and RRMS patients analyzed in Figure 1 were sampled during relapse, highlighting the importance of our finding – the fundamental difference between a first and subsequent relapses in MS. In the updated version of the manuscript, we have provided additional information in the main text, figure legends, and methods section.

- 2. If the latter is true, the upregulation of HB-EGF in CIS may only indicate the presence of acute lesions. The reduction of HB-EGF over time may then only reflect the decrease of acute demyelination in chronic MS patients rather than an evolving failure of tissue-protective mechanisms that leads to progression.**

As described in the prior comment, all patients were sampled during relapse, representing a timepoint of acute inflammation. This is of fundamental importance and biological relevance, which we have highlighted in the revised version of the manuscript.

Other shortcomings are:

- 3. It is striking that progressive MS patients are not included in the CSF study, given that the authors refer to MS progression throughout the introduction and discussion.**

We thank the Reviewer for the opportunity to clarify this point. Indeed, effective therapies for progressive MS are limited and finding novel therapeutic approaches is of utmost relevance. However, CSF of patients with progressive disease is only sampled on rare occasions and the underlying pathophysiology is highly heterogenous, representing technical challenges for the analysis of large quantities of patients. Nevertheless, in order to address the Reviewers comment, we have analyzed the abundance of HB-EGF in the CSF of secondary MS and primary progressive MS by single-plex ELISA (Extended Data Fig. 1g). Furthermore, we have performed methylation analyses in CNS specimens and whole blood derived from acute and progressive MS patients (Fig. 9), overall suggesting that epigenetically mediated suppression of HB-EGF expression may also play a role in progressive disease stages of MS.

- 4. The authors suggest that increased promoter methylation at Hifa/ARNT binding sites might be responsible for reduced HB-EGF production in astrocytes over time. There is only one time point (disease peak) presented vs naïve mice. Promoter methylation e.g. at disease onset and during the late disease stage would make a more convincing argument for increasing promoter methylation during the disease course.**

We thank the Reviewer for this great suggestion. In response, we have performed additional EAE experiments and sampled astrocytes from naïve, and peak, as well as recovery stages of EAE. Targeted methylation analysis of HIF1a binding sites in the HBEGF promoter by high-resolution melt analysis following bisulfite conversion revealed that promoter hypermethylation persists throughout recovery stages (Fig. 7d). This is complemented by the analysis of HBEGF promoter methylation in RRMS patients during relapse and remission, as well as progressive MS patients (Fig. 9), demonstrating that hypermethylation may mediate the long-lasting loss of protective signaling.

- 5. The initial observation has been done in CIS/MS patients, and the remainder of the study is conducted in mice and cell cultures. Looping back to human tissue and demonstrating e.g. increased promoter methylation at Hifa/ARNT binding sites in astrocytes within chronic active vs acute lesions, would translate the in vitro and in vivo observations back into MS.**

In response to this excellent point brought forward by the Reviewer, we have leveraged two published epigenome datasets of CNS specimens derived from MS patients, which revealed glia specific hypermethylation in HBEGF promoter segments that harbored HIF1a binding sites (Fig. 9). Additionally, we have validated our initial observations of HB-EGF upregulation by astrocytes upon acute inflammatory insult in stereotactic biopsies from a patient with ADEM, collectively supporting the validity of our findings, looping back into MS tissue and showing the validity of our observations in MS patients.

Minor:

- 6. Blocking of HB-EGF in LPS-activated microglia incubated with ACM derived from hypoxic astrocytes did produce significant differences in Il1b or Csf2 (Fig 4h), therefore, this finding does not support “the idea that astrocyte-derived HB-EGF directly controls the inflammatory potential of microglia”.**

To address the Reviewers comment, we have performed additional experiments, now demonstrating the effects of astrocyte-derived HB-EGF on the expression of inflammatory genes associated with pro- or anti-inflammatory functions in microglia (Fig. 5j).

- 7. Line 333-334. “The negative correlation between HB-EGF levels in the CSF with the number of CNS lesions in CIS patients, “. The legend to Fig. 1f describes a linear regression analysis of HB-EGF concentration in the CSF of RRMS patients and lesion number.**

We apologize for this oversight. Indeed, the correlation describes a linear regression analysis of the HB-EGF concentration in the CSF of CIS patients and the number of CNS lesions. We have corrected this error in the updated version of the manuscript.

Decision Letter, first revision:

13th Nov 2023

Dear Dr. Rothhammer,

Thank you for submitting your revised manuscript "Astrocyte-produced HB-EGF limits autoimmune CNS pathology" (NI-A35533A). It has now been seen by the original referees and their comments are below. The reviewers find that the paper has improved in revision, and therefore we'll be happy in principle to publish it in Nature Immunology, pending minor revisions to satisfy the referees' final requests and to comply with our editorial and formatting guidelines.

We will now perform detailed checks on your paper and will send you a checklist detailing our editorial and formatting requirements in about a week. Please do not upload the final materials and make any revisions until you receive this additional information from us.

If you had not uploaded a Word file for the current version of the manuscript, we will need one before beginning the editing process; please email that to immunology@us.nature.com at your earliest convenience.

Thank you again for your interest in Nature Immunology. Please do not hesitate to contact me if you have any questions.

Sincerely,

Stephanie Houston, PhD
Senior Editor
Nature Immunology

Reviewer #1 (Remarks to the Author):

We thank the authors for diligently addressing our concerns with the original version of this manuscript. The manuscript is significantly improved.

We would add only the following minor alteration as being needed prior to publication in NI:

1. The new data showing that there is no evidence for off-target effects should be added as a supplemental figure. This is valuable information that readers will appreciate.

2. A few remaining issues in Fig. 8:

- While this reviewer has no quibble with the EAE data in Fig. 8a – they appear robust, I remain perplexed at the choice of regression analysis for analysis of the data. What test is producing a p-value? What model is used in the regression? Either reviewers should explain further or use an appropriate test to capture difference at each timepoint or as AUC. I am fine with either, but as it stands it is difficult for the reader to understand what was done.

- There is no difference in the brain (Fig. 8c). Line 483 says there are – thus the text needs to be modified.
- There is a text overlap mistake in Fig. 8d (lower y-axis) that needs to be corrected.
- Data in Fig. 8e are qualitative rather than quantitative. Thus, the statement in lines 494-496 need to be adjusted.

Author Rebuttal, first revision:

See inserted PDF

Reviewer #1 (Remarks to the Author):

1. We thank the authors for diligently addressing our concerns with the original version of this manuscript. The manuscript is significantly improved.

We would add only the following minor alteration as being needed prior to publication in NI:

We thank the Reviewer for this positive assessment and appreciate all the helpful comments that significantly improved the quality of our manuscript.

2. The new data showing that there is no evidence for off-target effects should be added as a supplemental figure. This is valuable information that readers will appreciate.

In response to the Reviewers comment, we have added the additional data on off-target effects in Extended Data Figure 3b-e.

3. While this reviewer has no quibble with the EAE data in Fig. 8a – they appear robust, I remain perplexed at the choice of regression analysis for analysis of the data. What test is producing a p-value? What model is used in the regression? Either reviewers should explain further or use an appropriate test to capture difference at each timepoint or as AUC. I am fine with either, but as it stands it is difficult for the reader to understand what was done.

We thank the Reviewer for the opportunity to clarify this point. The *t*-test of the linear regression model used for the analysis of EAE experiments compares the slope of the curves. A *P* value (two-tailed) is calculated by testing the null hypothesis that the slopes are all identical (the lines are parallel).

4. There is no difference in the brain (Fig. 8c). Line 483 says there are – thus the text needs to be modified.

In response to the Reviewers comment, we have made the respective changes in the main text and apologize for this oversight.

5. There is a text overlap mistake in Fig. 8d (lower y-axis) that needs to be corrected.

We thank the Reviewer for noting this mistake. We have corrected it in the updated version of the Manuscript.

6. Data in Fig. 8e are qualitative rather than quantitative. Thus, the statement in lines 494-496 need to be adjusted.

In response to the Reviewers comment, we now provide quantitative data in Extended Data Figure 8.

Final Decision Letter:

Dear Dr. Rothhammer,

I am delighted to accept your manuscript entitled "Astrocyte-produced HB-EGF limits autoimmune CNS pathology" for publication in an upcoming issue of Nature Immunology.

Over the next few weeks, your paper will be copyedited to ensure that it conforms to Nature Immunology style. Once your paper is typeset, you will receive an email with a link to choose the appropriate publishing options for your paper and our Author Services team will be in touch regarding any additional information that may be required.

Please note that *Nature Immunology* is a Transformative Journal (TJ). Authors may publish their research with us through the traditional subscription access route or make their paper immediately open access through payment of an article-processing charge (APC). Authors will not be required to make a final decision about access to their article until it has been accepted. [Find out more about Transformative Journals](https://www.springernature.com/gp/open-research/transformative-journals).

If you have any questions about our publishing options, costs, Open Access requirements, or our legal

forms, please contact ASJournals@springernature.com

Your paper will be published online soon after we receive your corrections and will appear in print in the next available issue.

Also, if you have any spectacular or outstanding figures or graphics associated with your manuscript - though not necessarily included with your submission - we'd be delighted to consider them as candidates for our cover. Simply send an electronic version (accompanied by a hard copy) to us with a possible cover caption enclosed.

Please note that we encourage the authors to self-archive their manuscript (the accepted version before copy editing) in their institutional repository, and in their funders' archives, six months after publication. Nature Portfolio recognizes the efforts of funding bodies to increase access of the research they fund, and strongly encourages authors to participate in such efforts. For information about our editorial policy, including license agreement and author copyright, please visit www.nature.com/ni/about/ed_policies/index.html

An online order form for reprints of your paper is available at ><https://www.nature.com/reprints/author-reprints.html>. Please let your coauthors and your institutions' public affairs office know that they are also welcome to order reprints by this method.

Sincerely,

Stephanie Houston, PhD
Senior Editor
Nature Immunology